# Do we still need reflectance? From radiance to snow properties in mountainous terrain: a case study with EMIT

Niklas Bohn[1], Edward H. Bair[2], Philip G. Brodrick[1], Nimrod Carmon[1], Robert O. Green[1], Thomas H. Painter[3], and David R. Thompson[1]

[1]Jet Propulsion Laboratory, California Institute of Technology, Pasadena, CA, USA
[2]Civil Group, Leidos, Inc., Reston, VA, USA
[3]Joint Institute for Regional Earth System Science and Engineering, UCLA, Los Angeles, CA, USA

**Correspondence:** Niklas Bohn (urs.n.bohn@jpl.nasa.gov)

**Abstract.** Global patterns of snow darkening and melting, induced by grain metamorphism and the accumulation of small light-absorbing particles (LAPs), such as mineral dust, black carbon, volcanic ash, or algae cells, lead to an intensified radiative forcing and retreat of Earth's snow cover. Mapping and quantifying snow grain size and LAPs on both temporal and spatial scales is needed to improve the prediction of melt rates and their impacts on climate change. High-resolution visible-to-shortwave-infrared (VSWIR) imaging spectrometers herald a new era of passive spaceborne remote sensing, which will help to fulfill this objective. This technology provides measurements of reflected solar radiation in continuous spectral channels throughout the solar spectrum, allowing to detect narrow ice and LAP absorption bands. One of these instruments is NASA's Earth Surface Mineral Dust Source Investigation (EMIT) that was launched to the International Space Station (ISS) in July 2022. EMIT observations include snow cover in low to mid-latitude mountainous regions, such as the Western US, the Andes in South America, or high-mountain Asia. Accurate retrievals of snow surface properties, including grain size, liquid water content, as well as concentration of mineral dust and algae, require a precise, ideally joint accounting for atmospheric, topographic, and anisotropic effects in the reflected radiance. However, some methods still either neglect physical effects of the surface or utilize the surface reflectance as an intermediate non-physical quantity, in part without proper error propagation from the atmospheric modeling and obtained from statistical modeling. Moreover, the term 'surface reflectance' is often used with ambiguity in the literature, which instantly raises the question if we still need this quantity as a retrieval product. In this contribution, we present a novel forward model that couples the MODTRAN atmosphere radiative transfer code with a physics-based snow reflectance model that utilizes the multistream DISORT program. Our model allows to estimate snow surface and atmosphere properties directly from measured radiance. We apply the approach to EMIT images from Patagonia, South America, and compare our results to the EMIT L2A products that retrieve surface reflectance as a free parameter. We find discrepancies in snow grain size of up to 200 $\mu$m and in dust mass mixing ratio of up to 75 $\mu$g g$^{-1}$. Furthermore, we demonstrate differences in instantaneous LAP radiative forcing of up to 400 W m$^{-2}$. We conclude that we still need reflectance, but only if clearly defined and preferably as a modeled quantity within the forward model. These findings will be essential for the conception of retrieval algorithms for future orbital imaging spectroscopy missions, such as NASA's Surface Biology and Geology (SBG).

## 1 Introduction

Snow surfaces play a key role in Earth's radiation budget as their high albedo reflects most of the incoming solar radiation, steering important feedback mechanisms in climate change (Lemke et al., 2007). Changes in global snow cover are very sensitive to small variations in both air temperatures and the amount of absorbed solar radiation (Di Mauro et al., 2015). Moreover, these variations significantly impact the Earth's hydrological cycle and energy balance at different spatial and temporal scales (Flanner and Zender, 2005; Flanner et al., 2007; Bond et al., 2013; Painter et al., 2013a). Reduction of snow cover and albedo significantly contributes to climate warming, and therefore, bears a major risk to human society (Di Mauro et al., 2015).

One of the main drivers of the decrease in snow cover and albedo is the presence of small light-absorbing particles (LAPs) on snow and ice surfaces (Di Mauro, 2020). These particles are mainly absorptive in the visible part of the solar spectrum where the Sun's irradiance is highest, and when present lead to a considerable amount of additionally absorbed radiation. This process elevates surface temperatures, causing enhanced snow melt and finally, a decline of snow cover. By modifying the chemical and radiative properties of snow, LAP absorption is directly linked to snow hydrology as it impacts the timing of snowmelt and the retreat of glaciers and ice sheets (Clarke and Noone, 1985; Flanner et al., 2007, 2009; Oerlemans et al., 2009; Rhoades et al., 2010; Hadley et al., 2010; Wientjes et al., 2011; Painter et al., 2012b, 2013a; Sterle et al., 2013; Li et al., 2013; Kaspari et al., 2015). LAPs include both inorganic particles such as mineral dust, black carbon, or volcanic ash, as well as biological material such as various algal species (Flanner et al., 2007; Skiles et al., 2018; Di Mauro, 2020). While algae grow in nutrient-rich melting snow (Williamson et al., 2018), inorganic LAPs are mainly deposited from the atmosphere after sometimes being transported thousands of kilometers (Schepanski, 2018). They originate from different sources, depending on the particle type. Black carbon and ash are emitted from forest fires, volcanic eruptions, or fossil fuel and biofuel combustion (Bond et al., 2013). Mineral dust, however, emanates from arid, non-vegetated regions that are exposed to wind erosion (Prospero, 2002). Characteristic dust storms can transport the particles to remote areas including high mountain snow cover, glaciers, and ice sheets (Mahowald et al., 2014). Impacts of dust deposition on snowpack dynamics have been reported for different geographical regions, including the western US, the Himalayas, Greenland, and the European Alps (Painter et al., 2012b, a; Dumont et al., 2014; Di Mauro et al., 2015). However, it is still challenging to accurately quantify global and regional deposition rates of mineral dust due to its highly variable optical properties, which depend on the source region and can change during atmospheric transport (Miller et al., 2004).

Several studies have quantified the albedo reduction and enhanced radiative forcing caused by LAP absorption using remote sensing data, with a particular focus on mineral dust (Painter et al., 2012a; Gautam et al., 2013; Li et al., 2013; Dumont et al., 2014). Generally based on MODIS data, they applied spectral indices and modeling of snow albedo. However, a quantitative retrieval of inorganic LAP concentration from spaceborne measurements is for the most part still missing in the literature. Measurements from the recently launched orbital imaging spectroscopy missions EnMAP and PRISMA have been utilized

to conduct preliminary sensitivity analyses and first attempts to estimate dust concentration on snow and ice (Bohn et al., 2021, 2022; Kokhanovsky et al., 2022). These studies concluded though that low amounts of inorganic LAP deposition cannot be detected with remote sensing measurements, which is in line with findings from other studies (e.g., Warren (2013)). One of the main challenges in these cases is that variations in snow albedo caused by weak LAP absorption are of the same magnitude as instrument noise.

Future orbital imaging spectroscopy missions, such as NASA's Surface Biology and Geology (SBG) (National Academies of Sciences, Engineering, and Medicine, 2018) and ESA's Copernicus Hyperspectral Imaging Mission for the Environment (CHIME) (Rast et al., 2019) will address this problem by providing high signal-to-noise ratios (SNR) of more than 400 in the visible-to-near-infrared (VNIR) and more than 250 in the shortwave-infrared (SWIR), as well as high spectral and spatial resolution. This will enable the detection and quantification of LAPs by resolving their subtle spectral absorption features even for low concentrations. One of the precursor missions for SBG and CHIME is NASA's Earth Surface Mineral Dust Source Investigation (EMIT) that was launched to the International Space Station (ISS) in July 2022 (Green et al., 2020). EMIT is a high performance VNIR-SWIR imaging spectrometer whose prime mission focus is to deliver maps of surface mineralogy and relative abundance of different mineral types from arid dust source regions. These maps will provide improved input to Earth System Models of atmospheric transport and radiative forcing by constraining the composition of regional dust emissions (Connelly et al., 2021). Extending over a wavelength range of 380-2500 nm with a spectral resolution of approximately 7.5 nm, EMIT images provide a pixel size of around 60 m on a 74 km wide swath. The temporal revisit time is variable depending on the orbital cycle of the ISS, and ranges between one day and more than a week (Thompson et al., 2024). After more than a year in operation, EMIT provides data products from many different regions of the Earth, including snow covered high mountains, and experiments have confirmed a remarkably high SNR of more than 500 on average and above 750 in the VNIR wavelengths (Thompson et al., 2024). SBG and CHIME will supply an even larger data volume by giving approximately weekly combined observations of reflected solar radiation from almost all locations on Earth's surface. This new era of spaceborne imaging spectroscopy calls for a more advanced, globally applicable atmosphere and surface modeling that maximizes the incorporation of physically-based retrievals (Cawse-Nicholson et al., 2021).

Recent work has demonstrated that a simultaneous inversion of atmosphere and surface state using optimal estimation (OE) shows promising potential to quantify even low concentrations of LAPs on a global scale from spaceborne imaging spectroscopy observations (Bohn et al., 2021, 2022). However, the approach utilizes the surface reflectance as an intermediate non-physical retrieval quantity assuming Lambertian behavior. It is obtained from statistical modeling using constrained priors, impeding a proper consideration of surface topography and anisotropy. This could lead to significant biases in downstream estimates of LAP concentration, and propagate to erroneous calculations of LAP radiative forcing as these physical effects influence both magnitude and shape of measured spectral radiance as a function of local view and solar geometry (Carmon et al., 2022, 2023). Specifically, Picard et al. (2020) demonstrated the sensitivity of snow albedo measurements to surface slope based on spectral data taken in the field, and proposed a correction approach to retrieve the intrinsic albedo. Using a digital elevation model (DEM), this local geometry, including surface slope and aspect, can be calculated and incorporated in atmospheric modeling schemes in order to correct for spectral distortions in the retrieved surface reflectance (Richter and

Schläpfer, 2017). However, given the complex terrain of mountainous regions, and the current unavailability of coincident radar/lidar and imaging spectroscopy data in orbit, reliance on fixed DEMs may introduce additional retrieval errors, not only due to variability in local snow depth, but also because of uncertainties in the DEM product itself (Dozier et al., 2022).

For instance, Donahue et al. (2023) showed that topographic correction with coarse and non-coincident DEMs introduces significant errors in estimated snow albedo from air- or spaceborne imaging spectroscopy of up to 20%. Overall, there can be a handful of factors biasing snow reflectance shape and magnitude, especially in the blue wavelengths, as demonstrated in a complementing study by Bair et al. (2024). Only recently, Wilder et al. (2024) introduced a more mature and comprehensive modeling of topography for spaceborne imaging spectroscopy data over mountain snow, but most studies apply a limited

post-hoc correction at the airborne scale (Painter et al., 2013b; Seidel et al., 2016).

To improve the downstream estimation of biogeophysical quantities, we need to align the surface and atmospheric forward modeling assumptions. In particular, the retrieval of properties on highly anisotropic surfaces such as snow and ice will benefit from capturing local topographic conditions through physical modeling as directional effects are minimized. We present an updated version of the algorithm that was originally introduced by Thompson et al. (2018) and modified by Bohn et al.

(2021). It simultaneously retrieves atmosphere and surface properties from imaging spectrometer measurements by inverting a wavelength-dependent top-of-atmosphere (TOA) radiance model. In this work, we introduce a full physics-based characterization of atmosphere and surface by coupling the MODTRAN atmosphere radiative transfer code with the multistream DISORT program. The latter is utilized to simulate directional snow reflectance as a function of biogeophysical properties as well as view and illumination geometry. This facilitates the consideration of local surface anisotropy and topography in the forward

model and removes dependency from external DEMs (Carmon et al., 2023; Wilder et al., 2024). Aim is to utilize this best in class physical and atmospheric modeling simultaneously to present estimations of snow surface properties directly from measured radiance. Initial results for a single EMIT scene from Patagonia in South America are shown in Bohn et al. (2023) and indicate that retrieval errors of mineral dust concentration and LAP radiative forcing increase when a physics-based modeling of the surface is omitted. In this contribution, we substantiate previous findings by adding a comprehensive sensitivity analysis

of both our forward model and individual snow surface parameters, and provide more robust numbers by utilizing another EMIT image from a different region in Patagonia.

## 2 Background

### 2.1 Snow and ice anisotropy

Snow and ice are among the most variable surface types on Earth. Their albedo ranges from almost 1 for fresh snow to less than

120 0.3 for dark ice (Flanner and Zender, 2006; Painter et al., 2012b). Moreover, the hemispherical-directional reflectance factor (HDRF) of snow can even reach values well beyond 1 due to strong forward scattering effects (Painter and Dozier, 2004b). In this study, we follow the definition of HDRF as given by Schaepman-Strub et al. (2006), which is based on the nomenclature of Nicodemus et al. (1977) but incorporates the adaptations to the remote sensing case from Martonchik et al. (2000). It is defined as the ratio of reflected spectral radiance $L_r$ at a particular solar and view geometry to the radiant flux $L_{id}$ that would

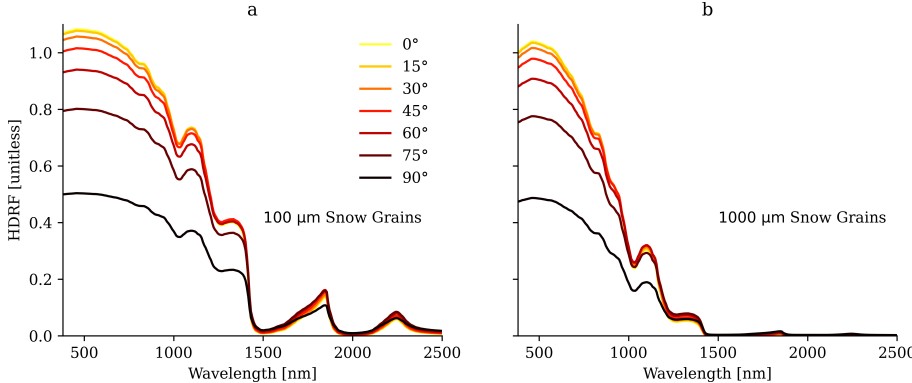

**Figure 1.** Snow HDRF simulated with DISORT as a function of incident angle $\theta_i$ for snow grain radii of 100 $\mu$m (a) and 1000 $\mu$m (b). The spectra are simulated with $\theta_r = 0°$ $\phi_r = 0°$, and 80 % direct irradiance, and resampled to EMIT instrument wavelengths.

be reflected from an ideal Lambertian surface, illuminated and observed under the same conditions:

$$HDRF = R(\theta_i, \phi_i, 2\pi; \theta_r, \phi_r) = \frac{L_r(\theta_i, \phi_i, 2\pi; \theta_r, \phi_r)}{L_{id}(\theta_i, \phi_i, 2\pi; \theta_r, \phi_r)}, \tag{1}$$

where $\theta$ and $\phi$ define zenith and azimuth, and the subscripts $i$ and $r$ denote incident and reflection angles, respectively. The HDRF scenario is composed of hemispheric illumination, i.e., both direct and diffuse irradiance, but only direct reflection. Reflectance products from air- and spaceborne imaging spectrometers generally yield the HDRF based on the assumption that it is constant over the relatively small cone angle of the instantaneous field-of-view (IFOV) of the instrument (Schaepman-Strub
et al., 2006).

Observed HDRF of snow surfaces significantly vary depending on the incident angle of solar illumination and the view angle of the sensor, resulting in a highly anisotropic surface (Painter and Dozier, 2004a). Figure 1 shows HDRF simulated with DISORT as a function of $\theta_i$ for snow grain radii of 100 $\mu$m (panel a) and 1000 $\mu$m (panel b). $\theta_r$, $\phi_r$, and fraction of direct solar irradiance are kept constant for this example. For both grain sizes, the HDRF exhibits a remarkable decrease in magnitude at
larger solar zenith angles due to increased forward scattering (Warren, 1982). On the other hand, we observe values $> 1$ in the visible (VIS) wavelength range when $\theta_i = \theta_r = \phi_r = 0°$. This excess of 100 % reflectance slightly diminishes for larger grains since they feature less angular structure in the scattering phase function and a smaller single-scattering albedo (Warren, 1982). At the same time, this causes the HDRF of dry snow with small grain sizes to be more sensitive to solar illumination angles in the near-infrared (NIR) and shortwave-infrared (SWIR) portions of the spectrum, resulting in higher uncertainties of retrieved
grain sizes when not accounting for the correct $\theta_i$.

Since the HDRF includes diffuse illumination from the entire hemisphere, its shape and magnitude are susceptible to variations in the ratio of direct to diffuse incoming flux (Strub et al., 2003). Figure 2 highlights the sensitivity of HDRF to different fractions of direct solar irradiance and a complementing isotropic, hemispheric diffuse illumination for solar zenith angles of $0°$ (panel a) and $80°$ (panel b). Snow surfaces are properly forward scattering, i.e., have a significantly higher reflectance factor
in the forward direction than in the backward direction, though only for direct irradiance fractions of $> 80$ % (Schaepman-

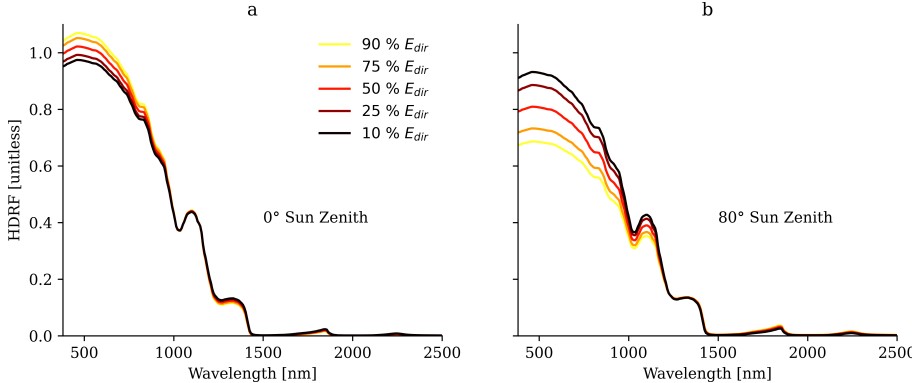

**Figure 2.** Snow HDRF simulated with DISORT as a function of direct solar irradiance fraction $E_{dir}$ for $\theta_i = 0°$ (a) and $\theta_i = 80°$ (b). The spectra are simulated with $\theta_r = 0°$, $\phi_r = 0°$, and $r_s = 500\ \mu$m, and resampled to EMIT instrument wavelengths.

Strub et al., 2006). As a consequence, we observe an inverse behavior of HDRF magnitude for the two illumination conditions, particularly in the VIS wavelengths. With smaller amounts of direct irradiance, HDRF decreases at $\theta_i = 0°$ due to the angular intersection of the forward scattering phase function with the surface (Schaepman-Strub et al., 2006), whereas its values significantly increase at $\theta_i = 80°$. As illumination conditions approach an entirely diffuse scenario, the HDRF becomes almost independent from $\theta_i$. This can be explained by the notably smaller magnitude of the forward reflectance distribution in these cases (Schaepman-Strub et al., 2006).

All these characteristics induce a dependency of HDRF on atmospheric conditions, scattering from surrounding objects, and local solar and view geometry. The latter is mainly determined by slope and aspect of the surface. Snow surfaces thus commonly exhibit significant directional effects, given the combination of high intrinsic anisotropy and occurrence in topographically challenging mountainous terrain. Variance in these directional effects is therefore primarily controlled by variations in surface topography (Painter and Dozier, 2004a). Hence, consideration of these effects during forward modeling is essential to retrieve the intrinsic shape and magnitude of snow reflectance, which only enables an accurate retrieval of downstream products, such as snow grain size and LAP concentration.

## 2.2 Study area: Patagonia

While targeting arid dust source regions of the Earth (Green et al., 2020), EMIT acquired many images over snow-covered low to mid-latitude mountain ranges, such as the Himalayas, the Sierra Nevada, and the Andes. We focused our study on Patagonia in South America since the largest ice fields and glaciers of the Southern Hemisphere outside Antarctica can be found there. A combination of high rates of precipitation at the Western slopes of the Andes and relatively low sea-surface temperatures of the coastal Pacific allows cold and humid air masses to rise and form an icy macroclimate (McEwan et al., 1997).

We selected two EMIT images from the early acquisition stage in September 2022, covering the Andes and Patagonia in late Southern Hemisphere winter. Figure 3 shows true-color images of the two scenes in the upper panels, accompanied by

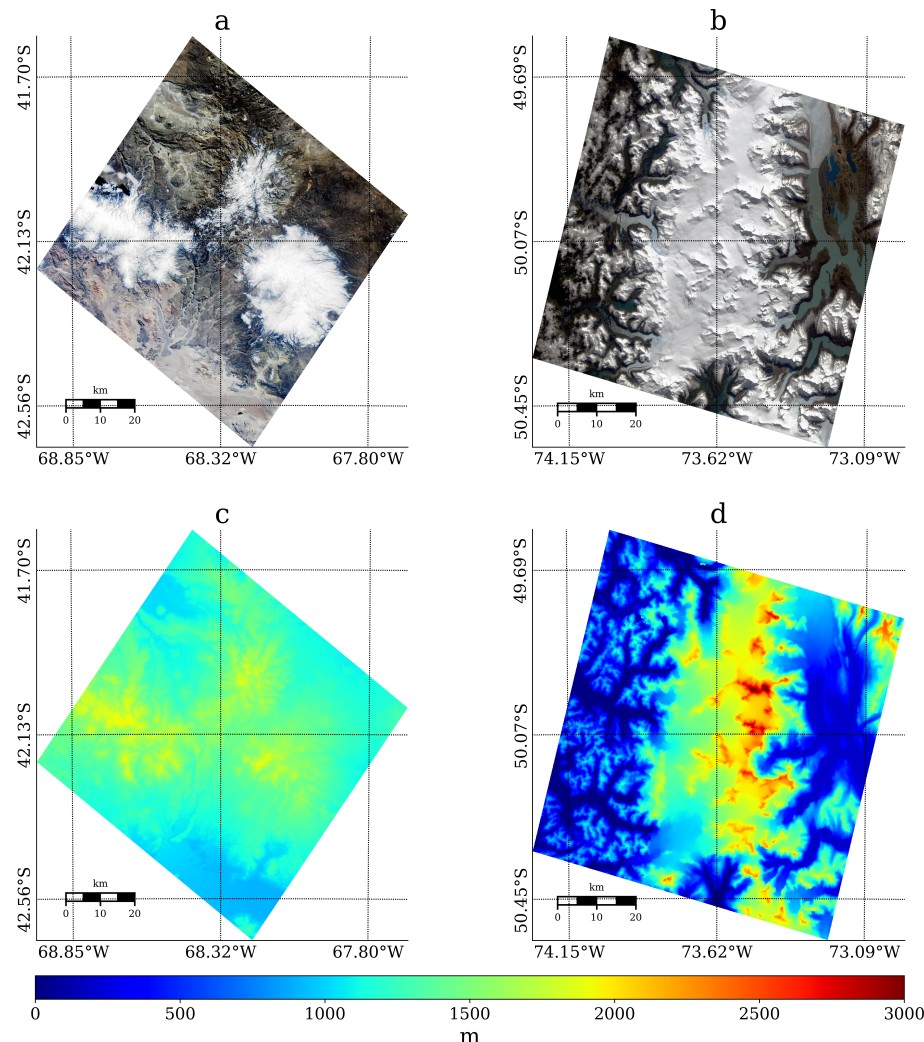

**Figure 3.** Selected EMIT acquisitions over South America. Panels (a) and (b) are true-color images, panels (c) and (d) show surface elevation for the given image extent calculated from the global SRTM DEM. Left panels show an area from the Argentine plain, right panels from the Chilean ice field. (Panel (b) adapted from Bohn et al. (2023). Copyright [2023] by IEEE. Adapted with permission.)

surface elevation maps calculated from the SRTM global DEM in the lower panels. Figures 3a and 3c display a scene collected over the Argentine plain with center coordinates $42.13°\,S$, $68.32°\,W$, whereas Figs. 3b and 3d depict an image acquired over the Chilean ice field in the southwest of Patagonia at $50.07°\,S$, $73.62°\,W$. The Argentine plain features less variation in surface elevation, but provides an interesting study area by having patchy snow fields mixed with bare soil and rock pixels. Furthermore, almost the entire southern part of Argentina is covered by the Patagonian desert, which is an arid source region of mineral dust. This dust is likely to be deposited in adjacent regions on snow-covered surfaces (Gasso and Torres, 2019).

The Chilean ice field belongs to the southernmost part of the Andes in close proximity to the Pacific ocean. The area features a highly variable topography with surface elevation ranging between sea level and 3000 m (see Fig. 3d). The mountain range shows the characteristic shape of the American Cordillera, having a rather smooth increase in elevation at the ocean-facing side, and a steep slope at the eastward part. It is almost entirely covered by the Southern Patagonian Ice Field, which is the world's second largest contiguous extrapolar ice field and feeds dozens of glaciers. The selected EMIT image covers glacier tongues and outlets as well as glacial lakes and fjords, offering a variety of different ice and snow surface types.

## 3 Methods

### 3.1 Coupled atmosphere-surface radiative transfer model

Our approach is based on previously published work that simultaneously retrieves atmosphere and surface properties from imaging spectrometer measurements (Thompson et al., 2018; Bohn et al., 2021, 2022). It inverts a forward model $\boldsymbol{F}(\boldsymbol{x})$ that calculates wavelength-dependent TOA radiance $\boldsymbol{l_{toa}}$ as a function of the state $\boldsymbol{x}$. Following Thompson et al. (2018), we combine an atmosphere and surface state in $\boldsymbol{x}$, so that $\boldsymbol{x} = [\boldsymbol{x_{ATM}}, \boldsymbol{x_{SURF}}]^T$, and invert each pixel of the image, i.e., each measured radiance spectrum, independently. While inversions of remotely sensed data commonly rely on a simplified version of $\boldsymbol{F}(\boldsymbol{x})$, we slightly increase complexity by adding a different scaling of incoming direct and diffuse illumination to account for surface topography, which we call a multi-transmittance approach following Carmon et al. (2022). While acknowledging their impact on reflectance of larger slopes, we currently exclude adjacency effects of neighboring pixels as their modeling is complex and requires assumptions on the neighboring terrain which may introduce additional uncertainty themselves (Picard et al., 2020). With bold letters indicating $m$-dimensional vectors, and $m$ being the number of instrument channels, $\boldsymbol{F}(\boldsymbol{x})$ denotes as:

$$\boldsymbol{l_{toa}} = \boldsymbol{l_0} + \frac{\boldsymbol{e_o}\pi^{-1}(cos(\theta_i)\boldsymbol{t_{dir}^{\downarrow}} + cos(\theta_0)\boldsymbol{t_{dif}^{\downarrow}})}{1 - s\rho_s}\rho_s \boldsymbol{t^{\uparrow}} \quad . \tag{2}$$

Here, $\theta_0$ and $\theta_i$ represent solar zenith angles at top-of-atmosphere and to the surface normal, respectively. The latter is also called local solar zenith angle. $\boldsymbol{e_o}$ is the incoming solar irradiance at top-of-atmosphere. $\boldsymbol{l_0}$ is the atmospheric path radiance, i.e., the number of solar photons that are scattered by the atmosphere into the line-of-sight of the sensor without interaction with the surface. $\boldsymbol{t_{dir}^{\downarrow}}, \boldsymbol{t_{dif}^{\downarrow}}$, and $\boldsymbol{t^{\uparrow}}$ are direct downwelling, diffuse downwelling, and total upwelling transmittance of the atmosphere. Scaled by two different angles, $\boldsymbol{t_{dir}^{\downarrow}}$ and $\boldsymbol{t_{dif}^{\downarrow}}$ represent the partition into direct and diffuse irradiance at the surface. Finally, $s$ is the spherical albedo of the atmosphere, which describes the multiple scattering of photons between the target pixel and the surrounding atmosphere before they enter the line-of-sight of the sensor. All these functions depend on $\boldsymbol{x_{ATM}}$, which includes water vapor, aerosol optical depth (AOD), and pressure elevation, and are obtained from the MODTRAN6 radiative transfer code (Berk and Hawes, 2017). $\rho_s$ is the HDRF and a function of $\boldsymbol{x_{SURF}}$ that holds snow grain size, liquid water content, algae concentration, and dust mass mixing ratio, as well as of the partition into direct and diffuse irradiance. Aligning with Wilder et al. (2024), we remove dependency from digital elevation models by adding $\theta_i$ to $\boldsymbol{x_{SURF}}$, which ensures "physical consistency, temporal coincidence, and spatial alignment" (Carmon et al., 2023). Our approach breaks from previous

**Table 1.** Overview of state vector parameters.

| Parameter | Unit | State Vector |
|---|---|---|
| Water Vapor | g cm$^{-2}$ | $\boldsymbol{x_{ATM}}$ |
| AOD | unitless | $\boldsymbol{x_{ATM}}$ |
| Pressure Elevation | m | $\boldsymbol{x_{ATM}}$ |
| Local Solar Zenith Angle ($\theta_i$) | degree | $\boldsymbol{x_{SURF}}$ |
| Snow Grain Size | $\mu$m | $\boldsymbol{x_{SURF}}$ |
| Liquid Water Fraction | % | $\boldsymbol{x_{SURF}}$ |
| Mineral Dust Mass Mixing Ratio | $\mu$g g$^{-1}$ | $\boldsymbol{x_{SURF}}$ |
| Snow Algae Concentration | cells mL$^{-1}$ | $\boldsymbol{x_{SURF}}$ |

implementations as we calculate all wavelength-dependent values of $\boldsymbol{\rho_s}$ by running a combination of Mie scattering theory and
the multistream DISORT program (Stamnes et al., 1988) instead of having them as free parameters in the state surface vector.
DISORT has traditionally been applied to model directional snow reflectance (Painter and Dozier, 2004a, b; Gardner and Sharp,
2010; Painter et al., 2013b; Skiles et al., 2017). By obtaining $\boldsymbol{\rho_s}$ from a surface radiative transfer model as a function of only
a handful parameters, our modification significantly reduces the dimension of $\boldsymbol{x}$. While $\boldsymbol{x_{SURF}}$ held the reflectance values
of each of the 285 EMIT channels in the previous implementation, our approach reduces this number to only 5 parameters
(Table 1).

Figure 4 outlines the basic steps of the retrieval framework. Both atmosphere and surface models are functions of $\theta_i$ since it
scales the incoming direct illumination on one hand, and on the other, it influences shape and magnitude of the surface HDRF.
Having $\theta_i$ as an input parameter both to the atmosphere and the surface model, couples its influence on the cost function and
provides leverage for the retrieval. To enhance processing speed, we use a multilinear interpolation of pre-calculated look-up-
tables (LUTs) of both MODTRAN and DISORT simulations in the inversion of the forward model.

We run DISORT with 16 streams and use three horizontal layers for modeling snow HDRF: a small near-surface layer that
contains impurities, and two semi-infinite LAP-free snow layers. Primarily, DISORT requires wavelength dependent optical
properties (OPs) of the medium of interest as input. OPs include the single-scattering albedo, the extinction coefficient, and the
asymmetry parameter. We assume the snow grains to be shaped as a collection of spheres characterized by a specified radius
that equals three times the volume-to-area ratio of the real non-spherical snowpack (Warren, 2019), and apply traditional Mie
Theory to obtain their OPs (Grenfell and Warren, 1999). Next, we calculate the single-scattering phase functions by decom-
posing the Henyey–Greenstein phase function, which better captures the actual phase function of snow than the phase function
for spheres, into Legendre coefficients for 20 moments (Aoki et al., 2000; Painter and Dozier, 2004a). We add the influence
of liquid water by modeling OPs of coated spheres for the case of wet snow by adding the width of a circular layer of liquid
water to the grain radius (Green et al., 2002). Defining OPs of LAPs is more arbitrary since many assumptions are theoretical
and based on numerical models, and lack the validation with field or laboratory measurements (Cook et al., 2017). However,

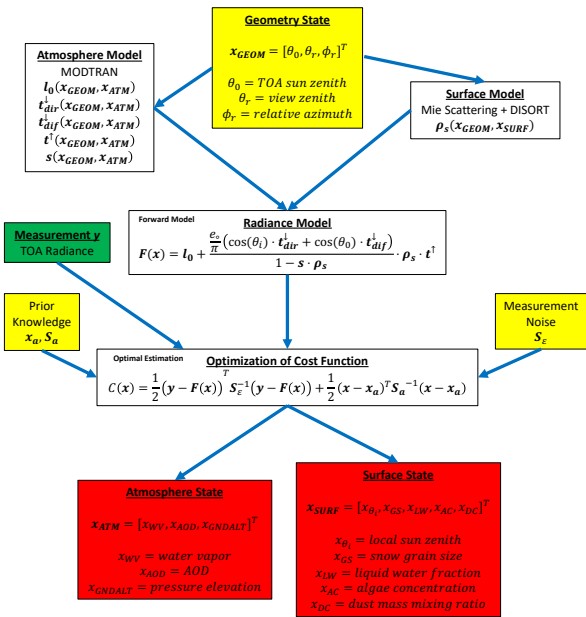

**Figure 4.** Schematic of algorithm workflow. The green box denotes the algorithm input, while red boxes indicate the final output. Yellow boxes contain known parameters needed by either the forward model or the Optimal Estimation framework. They include observation geometry as well as prior knowledge and instrument noise.

a recent study by Chevrollier et al. (2022) came up with a set of empirically measured OPs of both snow and glacier algae based on field samples collected on the Greenland Ice Sheet. We use their dataset to represent algal absorption in our DISORT simulations. Despite being sampled at a different geographic location, we believe that their characteristics are also applicable
to algal cells found on the Patagonian Ice Sheet. For mineral dust, we selected OPs as measured and described in Skiles et al. (2017). Again, it is an approximation as these properties are obtained from the San Juan Mountains in southwestern Colorado, but we are confident that they capture the significant impacts of dust particles on snow reflectance also in the Patagonian Andes. A thorough discussion about our choice of LAP OPs can be found in Sect. 5.2. We model the LAP-contaminated snow as a linear mixture of all OPs (Cook et al., 2020). Figure 5 visualizes the impacts of the different snow surface parameters on
shape and magnitude of HDRF. The samples are taken from the DISORT LUT for a fixed geometry of $\theta_0 = 40°$, $\theta_r = 0°$, and $\phi_r = 90°$. Larger snow grain size induces increasing absorption primarily throughout the entire NIR and SWIR wavelengths (Fig. 5a), and the effect of liquid water is expressed by a shift of NIR ice absorption features towards shorter wavelengths (Fig. 5b). In contrast, mineral dust particles reduce reflectance in the VIS by a smooth absorption behavior, similar to that of atmospheric aerosols (Fig. 5c), whereas snow algae show a distinct chlorophyll feature at $680$ nm and a broader carotenoid
absorption in the blue and green wavelengths (Fig. 5d) (Painter et al., 2001). Note that algae absorb solar radiation only in the visible part of the spectrum where the energy of the irradiance is highest, to maximize their photosynthesis (Chevrollier et al.,

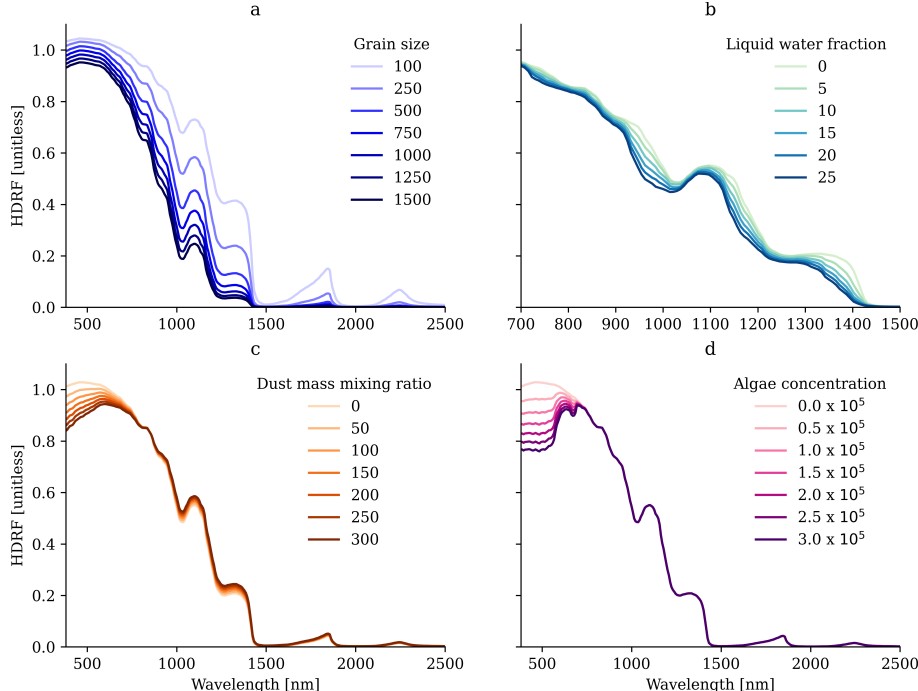

**Figure 5.** Shape and magnitude of snow HDRF as a function of (a) grain size [$\mu$m], (b) liquid water fraction [%], (c) dust mass mixing ratio [$\mu$g g$^{-1}$], and (d) algae concentration [cells mL$^{-1}$]. The spectra are simulated with $\theta_0 = 40°$, $\theta_r = 0°$, and $\phi_r = 90°$, and resampled to EMIT instrument wavelengths. Panels (b) to (d) assume a grain radius of 300 $\mu$m.

2022). Moreover, absorption by the surrounding ice and snow is overly strong in the near- and shortwave-infrared wavelengths, so that almost no solar energy would remain for algae pigments.

It has to be noted that our approach is only applicable to snow- and ice-covered pixels. We identified these areas by following the procedure of Dozier (1989). We utilize TOA reflectance $\rho_{TOA}$ of EMIT bands at 485 nm (blue), 567 nm (green), and 1648 nm (SWIR), and considered a pixel as snow-covered when $\rho_{TOA,485} > 0.16$, $\rho_{TOA,1648} < 0.25$, and NDSI > 0.4. The NDSI is the normalized-difference snow index calculated using $\rho_{TOA,567}$ and $\rho_{TOA,1648}$ (Dozier, 1989).

### 3.2 Inversion

We utilize Optimal Estimation (OE) to invert Equation 2 following the notation of Rodgers (2000). OE applies Bayes' theorem of joint probabilities and assumes Gaussian distribution of both measurement and state vector elements and their uncertainties. The inverse problem is then solved by locally linearizing the forward model and minimizing the value of the cost function $\mathcal{C}(\boldsymbol{x})$:

$$\mathcal{C}(\boldsymbol{x}) = (\boldsymbol{y} - \boldsymbol{F}(\boldsymbol{x}))^T \boldsymbol{S_\epsilon}^{-1} (\boldsymbol{y} - \boldsymbol{F}(\boldsymbol{x})) + (\boldsymbol{x} - \boldsymbol{x_a})^T \boldsymbol{S_a}^{-1} (\boldsymbol{x} - \boldsymbol{x_a}), \tag{3}$$

where $S_\epsilon$ is the measurement covariance matrix expressing instrument noise as well as forward model uncertainties; $x_a$ contains the prior mean for each state vector element; and $S_a$ is the associated prior covariance matrix. Since instrument noise is commonly assumed to be uncorrelated between channels (Thompson et al., 2018), $S_\epsilon$ takes a diagonal form, and its inverse weights the residual between modeled and measured radiance $l_{toa}$. On the other hand, the difference between prior mean and solution state is scaled by the inverse of $S_a$, which initially includes off-diagonal elements to allow correlation across state vector parameters (Thompson et al., 2018). However, we use only light constraints on the parameters in $x_{ATM}$, and uninformative, flat priors for $x_{SURF}$ by adding large values to the diagonal of $S_a$. Due to the reduced number of only eight state vector parameters and 285 independent measurements from EMIT's spectral channels, the problem is well-posed, in contrast to using previous surface models (Thompson et al., 2018; Bohn et al., 2021, 2022). Omitting the use of constrained priors is further supported by the high SNR of EMIT measurements. We enforce physical boundary conditions by utilizing DISORT to simulate surface reflectance during the inversion. Hence, the optimized snow and ice parameters are constrained by the physical shape of the reflectance as a function of grain size, liquid water content, and LAP concentration. We then apply the Truncated Newton Conjugate-Gradient Descent method to minimize the value of $\mathcal{C}(x)$ in Equation 3. Upon convergence, the reported state $x$ is the most probable solution to fit the measurement using the forward model in Equation 2. Our selected optimization scheme implies the risk of ending up in a local minimum, reporting a non-physical output. However, we use a traditional sequential atmospheric correction approach to initialize our inversion, which provides a first guess close to the probabilistic solution from OE. This promotes stability and fast convergence (Thompson et al., 2018). It usually finds a solution after five iterations on average with an overall processing speed-up of about four orders of magnitude compared to the traditional approach as presented in Thompson et al. (2018).

## 3.3 Radiative forcing

The retrieved snow surface properties can then be used to calculate additional downstream products, such as spectral and broadband snow albedo as well as LAP radiative forcing. The latter is a traditional way to assess the impacts of LAPs on snow melt and describes the enhancement of absorbed solar radiation induced by the presence of these small particles (Painter et al., 2013b).

We take $\rho_s$ at the solution state (see Table 1) and apply the approach from Painter et al. (2013b) to calculate LAP radiative forcing. The first step is to convert $\rho_s$ to spectral albedo $\alpha_s$. Since imaging spectrometers do not provide angular, hemispheric measurements, we convert $\rho_s$ to $\alpha_s$ by applying the spectral anisotropy factor $c$:

$$\alpha_s = \rho_s c, \tag{4}$$

where $c$ is a function of observation and illumination geometry as well as snow grain size, and is calculated as the ratio of spectral albedo to HDRF. We follow the approach of Painter et al. (2013b) and use a LUT of pre-calculated c coefficients based on modeled spectral albedo for different grain sizes as well as HDRF for different view and illumination geometries and varying grain sizes obtained from DISORT simulations.

We then calculate LAP radiative forcing by taking the spectral integral of the difference between a modeled clean snow albedo $\alpha_m$ and $\alpha_s$, convolved with the total downward irradiance arriving at the surface, $e$:

$$RF_{lap} = \sum_{\lambda=380nm}^{2500nm} e(\lambda)(\alpha_m(\lambda) - \alpha_s(\lambda))\Delta\lambda. \tag{5}$$

Here, $\Delta\lambda$ is the full width at half maximum (FWHM) of each EMIT channel. $\alpha_m$ is calculated by running DISORT with the retrieved surface state as input, but excluding any LAP concentration. $e$ is optimized during the inversion as a function of atmospheric state and $\theta_i$ (see Equation 2):

$$e = e_o\pi^{-1}(cos(\theta_i)t_{dir}^{\downarrow} + cos(\theta_0)t_{dif}^{\downarrow}). \tag{6}$$

$RF_{lap}$ is reported in W m$^{-2}$ and describes the amount of solar energy that is absorbed by the surface due to the presence of LAPs.

## 4 Results

### 4.1 Snow properties of Patagonia

We first present retrieval maps for the EMIT Patagonia scenes in order to highlight regional spatial patterns of selected surface parameters. We focus on the local solar zenith angle, snow grain size, and dust mass mixing ratio to outline the connection between topography, physical snow properties, and light-absorbing particles. Figure 6 shows estimated per-pixel $\theta_i$, grain size, and dust for the Argentine plain (Figs. 6a to 6c) and the Chilean ice field (Figs. 6d to 6f). Note the different colorbar ranges between the upper and lower panels, which we applied for clarity. P1 and P2 mark the locations of two example HDRF spectra, which we further investigate in Sect. 4.2.1. P1 points to the upper panels, P2 to the lower panels of Fig. 7.

The retrieved $\theta_i$ angles follow apparent slope and aspect in both images by featuring smaller values of less than $30°$ on northern, sun-facing slopes, and larger values of more than $75°$ on pixel plains oriented towards the south, away from direct illumination. Overall, the maps of estimated $\theta_i$ confirm the remarkable variation of surface orientation in mountainous terrain and the indispensable need for considering topography when modeling reflected radiance from snow and ice. A comparison between retrieved solar zenith angles and those calculated from the SRTM DEM is presented in Sect. 4.2. The SRTM DEM has a spatial resolution of 30 m, so that we averaged four pixels to get surface elevation, slope, and aspect values that match EMIT's ground sampling distance of 60 m.

Physical characteristics of the surface can influence the size of snow grains. Air temperatures can decrease with increasing elevation, which abates melt processes, causing drier snow with smaller grain size. In contrast, the higher amount of direct illumination on sun facing slopes could induce melt processes, leading to wetter snow with larger grain sizes than those which are observed on shaded flanks of mountain ranges. The retrieval maps in Figs. 6b and 6e outline this pattern by showing larger values towards the edges of the snow pack in Argentina, as well as at the lower elevations of the fjord slopes, mostly at the eastern side of the Chilean ice field. These late southern hemisphere winter scenes show that the snow is relatively dry, featuring

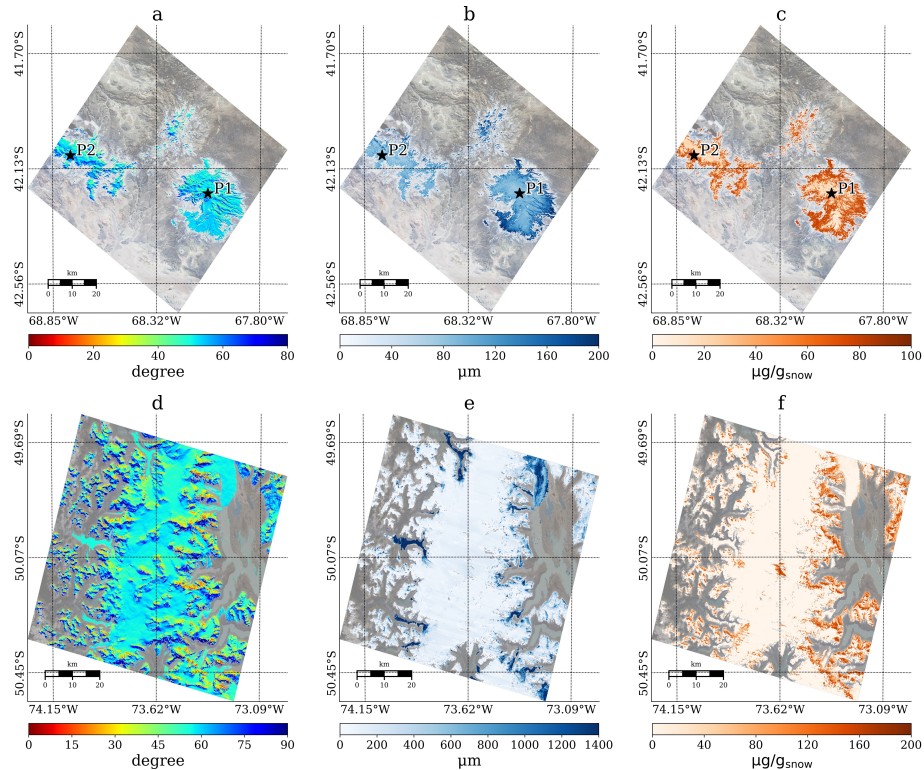

**Figure 6.** Retrieved $\theta_i$, snow grain size, and mineral dust mass mixing ratio from the Argentine plain (panels (a) to (c)), and from the Chilean ice field (panels (d) to (f)). Non-snow covered pixels are excluded from the retrieval. P1 and P2 in panels (a) to (c) mark the locations of the selected spectra in Fig. 7. P1 points to the upper panels, P2 to the lower panels of Fig. 7. Note the different colorbar ranges between panels (a) to (c) and (d) to (f). (Panels (d) to (f) adapted from Bohn et al. (2023). Copyright [2023] by IEEE. Adapted with permission.)

grain sizes of less than 200 $\mu$m at the Argentine plain and at higher elevations of the ice field in Chile. Here, areas closer to glacial outflows at the edges of the snow-covered surface show up to twice as large values, with grain sizes finally reaching up to 1500 $\mu$m for bare ice pixels.

The reported dust mass mixing ratios on snow surfaces follow a similar distribution on the Argentine plain, as shown in Fig. 6c. Areas with smaller snow grains feature less contamination than the edges of the snow covered patches. Likewise, the elevated regions of the Chilean ice field are almost free of dust deposition (Fig. 6f). An interesting pattern can be observed here as the eastern tongues of the ice field feature significantly higher deposition rates of dust particles than the western parts. Besides being less elevated, the eastern slopes of the Patagonian Ice Sheet are also more exposed to depositions of mineral

dust that has been transported through the atmosphere most likely from the Patagonian desert, which basically covers the entire southern part of Argentina.

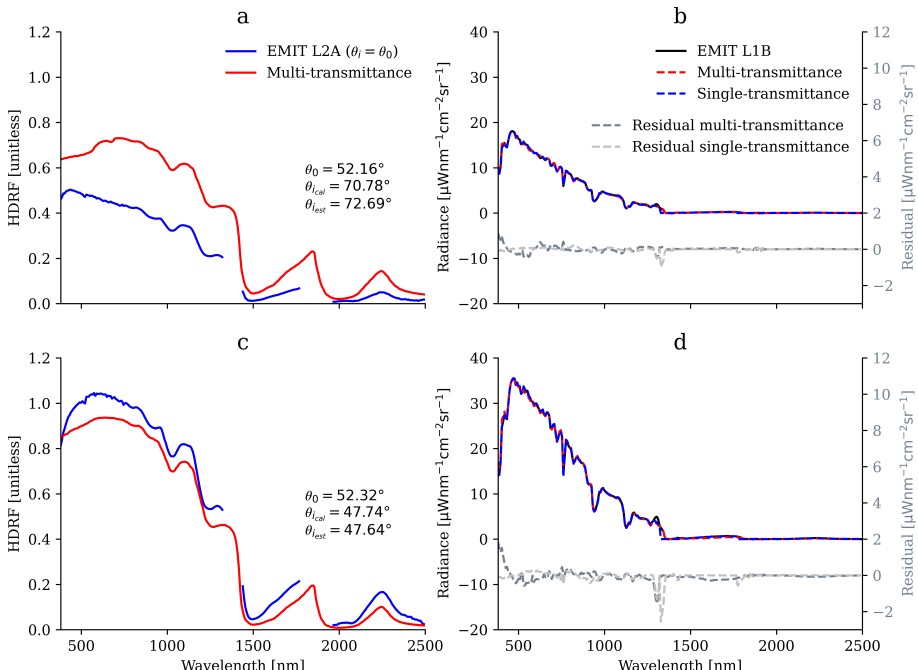

**Figure 7.** Estimated HDRF as well as radiance fits for two selected pixels from the Argentine plain. For comparison, panels (a) and (c) are complemented by results from the EMIT L2A product, which assumes $\theta_i = \theta_0$, i.e., uses a single downward transmittance term (direct + diffuse) in the forward model (called single-transmittance model hereinafter). Additionally, panels (a) and (c) indicate retrieved $\theta_{i_{est}}$ and the DEM-calculated $\theta_{i_{cal}}$. Panels (b) and (d) visualize modeled radiance at convergence and spectral residuals from the different forward models (single- and multi-transmittance) compared to the EMIT L1B measurement.

## 4.2 Sensitivity of snow physics

In this section we outline the major results from our sensitivity analysis by first looking at differences in estimated HDRF and corresponding radiance fits. Second, we highlight potential correlation between differences in assumed solar zenith angles and
325 uncertainties in retrieved surface parameters. Finally, we present induced impacts of errors in mineral dust concentration on the downstream calculation of LAP radiative forcing by showing a selection of prominent examples from the Chilean ice field.

### 4.2.1 Directional reflectance

Figure 7 picks two examples from the snow cover at the Argentine plain. Figures 7a and 7b correspond to location P1 in Figs. 6a-c, and Figs. 7c and 7d correspond to location P2. In addition, Fig. 7 lists both retrieved and DEM-calculated $\theta_{i_{est}}$
and $\theta_{i_{cal}}$, respectively, as well as the TOA $\theta_0$. We also show corresponding spectra from the EMIT L2A product. They were retrieved by applying the same OE technique, but without considering topography in the forward model, and by obtaining HDRF from statistical modeling using constrained priors instead of utilizing a snow surface radiative transfer model. We selected these particular pixels from the EMIT image since they are representative examples of both over- and underestimated

HDRF, given the respective difference of local to TOA solar zenith angle. The upper panels of Fig. 7 showcase a surface tilted away from the sun, i.e., with $\theta_i > \theta_0$. When ignoring this difference in the forward model, the amount of direct solar irradiance arriving at the surface is overestimated, causing a lower magnitude of retrieved reflectance due to a relatively small count of photons measured by the sensor. Likewise, the assumed ratio of direct to diffuse irradiance is too large, which compensates for present LAP absorption in the VIS wavelengths. Overall, this leads to a blue-shift in HDRF and is expressed by an upward hook in the shortest wavelengths. The multi-transmittance model is able to incorporate these effects, confirmed by the accurately retrieved $\theta_{i_{est}}$, which is around $20°$ larger than at TOA. The deviation from the DEM-calculated $\theta_{i_{cal}}$ is only $1.91°$. The low residuals of much less than $1\ \mu W\ nm^{-1}\ cm^{-2}\ sr^{-1}$ for all wavelengths in Fig. 7b indicate the validity of the multi-transmittance model, similar to the performance of the less complex single-transmittance model.

The lower panels of Fig. 7 show the opposite case where $\theta_i < \theta_0$ on a sun-facing slope. The difference is only about $5°$, but causes a clearly observable overestimation of HDRF when ignoring the local topography. The assumed direct illumination and direct to diffuse ratio are consequently too small, leading to higher reflectance due to more photons reaching the instrument, and causing a red-shift in HDRF and the formation of a downward hook, which is identified as a strong decrease in HDRF in the VIS blue wavelengths below 500 nm. This behavior is critical as it could imply the presence of LAP absorption on surfaces which are actually composed of clean snow. In particular, inorganic LAPs such as mineral dust are overestimated in most of these cases. Applying the multi-transmittance model and adding $\theta_i$ to the state vector of free parameters again improves the result. $\theta_{i_{est}}$ deviates from $\theta_{i_{cal}}$ by only $0.1°$ in this example. The radiance residuals in Fig. 7d show still acceptable low values relative to the absolute magnitude of the radiance spectrum.

### 4.2.2 Snow properties

The previous example highlighted the sensitivity of snow HDRF, but how is the retrieval of biogeophysical properties affected by the different handling of physics in the forward model? In addition to our multi-transmittance approach, we also ran the traditional retrieval with the assumption of a flat and horizontal surface, i.e., $\theta_0 = \theta_i$. Figure 8 shows scatter plots for the difference in retrieved snow properties as a function of the difference between $\theta_0$ and $\theta_i$. The upper panels a and b illustrate snow grain size and liquid water content, whereas the lower panels c and d look at the distribution of inorganic and biological LAPs, represented by mineral dust and snow algae.

It is obvious that the difference in retrieved values of liquid water and algae is uncorrelated with the difference in assumed local solar zenith angle. Both scatter plots feature an $r^2$ of around $0.0$ and almost no slope of the regression line. It seems that omitting topography and anisotropy in the forward model has no significant influence on the retrieval of those properties that have subtle absorption features and only marginally form the reflectance magnitude. This is confirmed by the simulated HDRF in Fig. 5. Liquid water in snow only causes a slight shift towards shorter wavelengths in the ice absorption features. Snow algae indeed decrease overall VIS reflectance due to the presence of different carotenoids, but they also commonly form a distinct chlorophyll absorption feature around $680\ nm$, which retains the identifiability even in challenging mountainous terrain.

In contrast, Figs. 8a and 8c indicate a dependency of derived snow grain size and mineral dust on the assumed illumination angle. Errors in mineral dust estimation are obvious as these particles cause a smooth absorption in the VIS spectrum, which

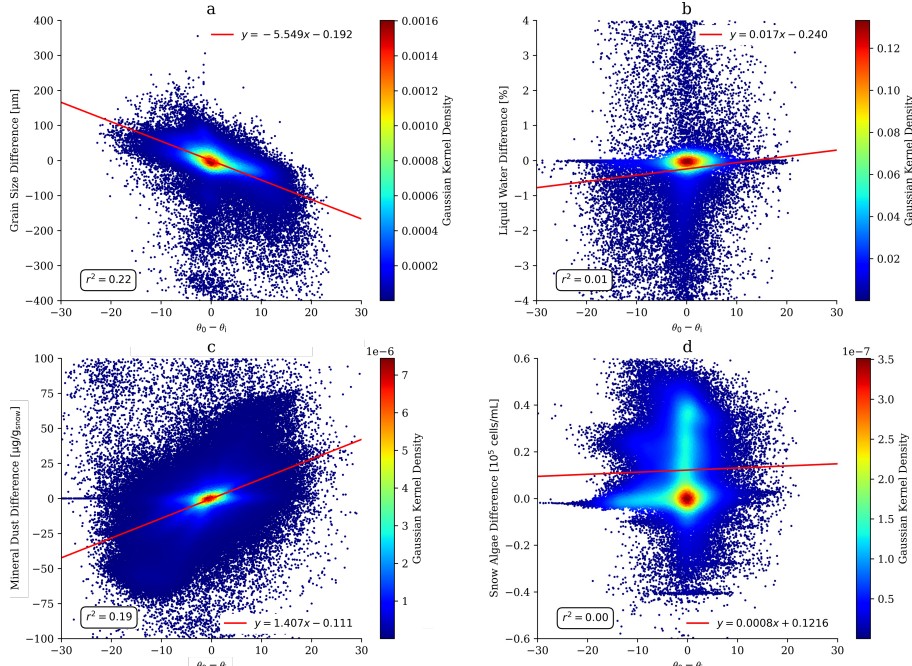

**Figure 8.** Difference in retrieved snow surface properties from the Argentine plain as a function of difference between $\theta_0$ and $\theta_i$. (a) Grain size, (b) liquid water fraction, (c) dust mass mixing ratio, and (d) algae concentration.

increases towards the blue wavelengths. This resembles the impacts of applying an erroneous ratio of direct to diffuse irradiance and therefore, increases the ambiguity between the two factors. When ignoring that $\theta_i < \theta_0$ in cases of sun-facing slopes, i.e.,

$\theta_0 - \theta_i > 0$, dust mass mixing ratio tends to be overestimated to compensate for an increased absorptive behavior of VIS blue reflectance. On surfaces tilted away from direct illumination, i.e., $\theta_0 - \theta_i < 0$, the retrieval is impacted in the opposite direction. Present snow contamination with dust particles is then underestimated. Our results show a mean increase of mineral dust difference of $1.4~\mu\text{g g}^{-1}$ per $1°$ gain of angular difference.

     Interestingly, the effects on the snow grain size retrieval are of a similar magnitude as on the dust estimations. An impul-

sive assumption would be that grain size is less affected due to the distinct ice absorption features in the NIR wavelengths. However, Fig. 8a confirms an $r^2$ of 0.22 and a mean decrease of retrieved snow grain size of $5.5~\mu\text{m}$ per $1°$ gain of angular difference. Consulting Figs. 1, 2, and 5, these findings become more traceable. The anisotropy of snow causes changing shape and magnitude of HDRF as a function of increasing grain size all along the solar spectrum. Hence, varying ratios of direct to diffuse illumination as a consequence of erroneously assumed local solar zenith angles lead to errors in derived snow grain

size. In particular, grain size is overestimated when $\theta_0 - \theta_i < 0$, and underestimated when $\theta_0 - \theta_i > 0$. Figure 7 supports this finding by showing respective differences in estimated NIR and SWIR reflectance.

     Another way of assessing the sensitivity of retrieved parameters is to look at their error correlation. OE provides a measure of retrieval uncertainty for each state vector element in terms of a posterior covariance matrix, which can be normalized to an error

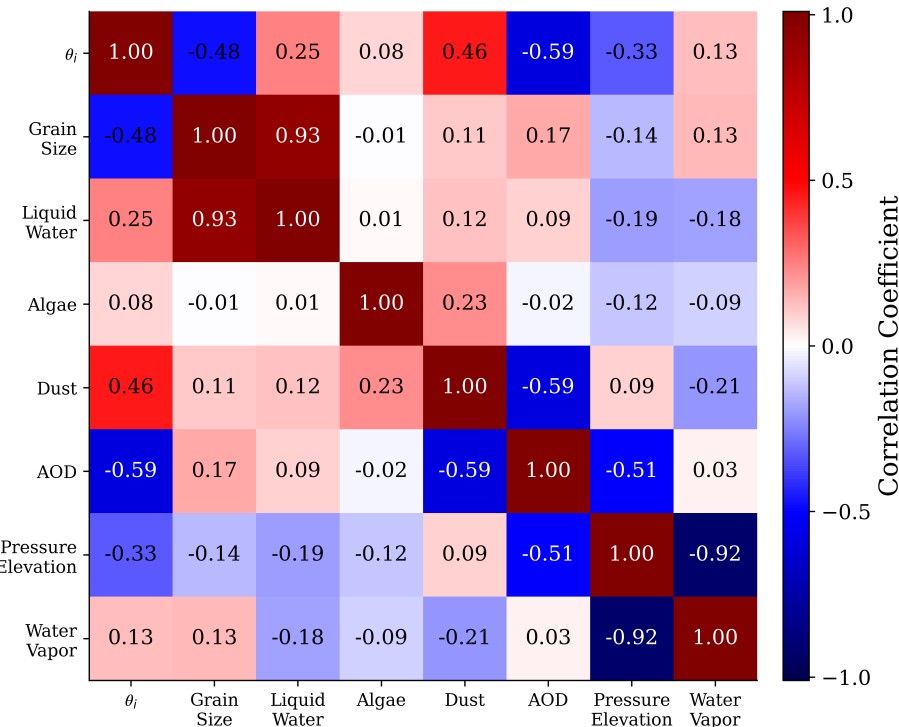

**Figure 9.** Posterior error correlation matrix calculated for the retrieved surface and atmosphere parameters from the image of the Chilean ice field.

correlation matrix (Rodgers, 2000; Govaerts et al., 2010). We calculated this matrix from the retrieval maps for the Chilean ice field for both atmosphere and surface properties (Fig. 9). The coefficients confirm the findings from Fig. 8 by showing a slight negative correlation between $\theta_i$ and grain size, a slight positive correlation between $\theta_i$ and dust, and no correlations between the assumed incident angle and both liquid water and algae. We observe expected anticorrelation between pressure elevation and both AOD and water vapor, and identify an overall disentanglement between surface and atmosphere parameters. Only exceptions are $\theta_i$, being an input to both atmosphere and surface model, and AOD, which features a negative correlation with dust LAPs. $\theta_i$, or more precisely, the assumed ratio of direct to diffuse irradiance, dust, and atmospheric aerosols cause a very similar shape of reflectance in the visible wavelengths, leading to potential ambiguities between them.

### 4.2.3 Radiative forcing

Given the findings from Sect. 4.2.2, ignoring topography and anisotropy can affect the calculation of LAP radiative forcing. We took a closer look at the Chilean ice field, manually selected six representative surface types, and calculated LAP radiative forcing for each of them. The selection process was based on visual interpretation of the true-color image and our retrieval results. The left panel of Fig. 10 shows the locations of these pixels with appropriate labels, and Table 2 presents the entire derived state from all six examples. Location S1 marks a spectrum from the plateau of the ice sheet at an elevation of 1827 m

**Table 2.** $\theta_0$, $\theta_{i_{cal}}$, and $\theta_{i_{est}}$, as well as retrieved atmospheric state and biogeophysical snow properties for the six locations as displayed in Fig. 10. In addition, we show posterior uncertainties for each retrieved parameter in standard deviation as reported by the optimal estimation framework. S1: dry snow, S2: melting snow, S3: dusty snow, S4: algal snow, I1: blue ice, I2: dark ice.

| | $\theta_0$ | $\theta_{i_{cal}}$ | $\theta_{i_{est}}$ | Pressure Elevation (DEM) [m] | Pressure Elevation ($est$) [m] | AOD [unitless] | Water Vapor [g cm$^{-2}$] | Grain Size [$\mu m$] | Liquid Water [%] | Dust [$\mu$g g$^{-1}$] | Algae [$10^5$ cells mL$^{-1}$] |
|---|---|---|---|---|---|---|---|---|---|---|---|
| S1 | 55.95 | 56.63 | 57.11 ± 4.41 | 1827 | 1902 ± 17 | 0.01 ± 0.01 | 0.22 ± 0.05 | 77 ± 3 | 0 ± 0.38 | 0 ± 3.1 | 0 ± 0.03 |
| S2 | 55.72 | 54.67 | 54.55 ± 3.72 | 1078 | 1023 ± 37 | 0.05 ± 0.07 | 0.28 ± 0.05 | 652 ± 15 | 4 ± 0.50 | 10 ± 4.4 | 0 ± 0.01 |
| S3 | 55.74 | 37.07 | 40.49 ± 7.29 | 1611 | 1645 ± 34 | 0.08 ± 0.10 | 0.24 ± 0.05 | 372 ± 14 | 1 ± 0.44 | 76 ± 6.8 | 0 ± 0.02 |
| S4 | 56.07 | 70.02 | 71.56 ± 2.37 | 815 | 902 ± 90 | 0.08 ± 0.06 | 0.33 ± 0.06 | 232 ± 12 | 9 ± 0.63 | 45 ± 8.5 | 0.62 ± 0.02 |
| I1 | 56.13 | 50.17 | 52.41 ± 3.85 | 39 | 13 ± 4 | 0.14 ± 0.07 | 0.69 ± 0.13 | 1500 ± 26 | 25 ± 3.62 | 0 ± 0.5 | 0 ± 0.01 |
| I2 | 56.19 | 72.08 | 76.41 ± 2.40 | 416 | 639 ± 35 | 0.40 ± 0.05 | 0.43 ± 0.06 | 1500 ± 30 | 25 ± 7.10 | 80 ± 10 | 0.1 ± 0.04 |

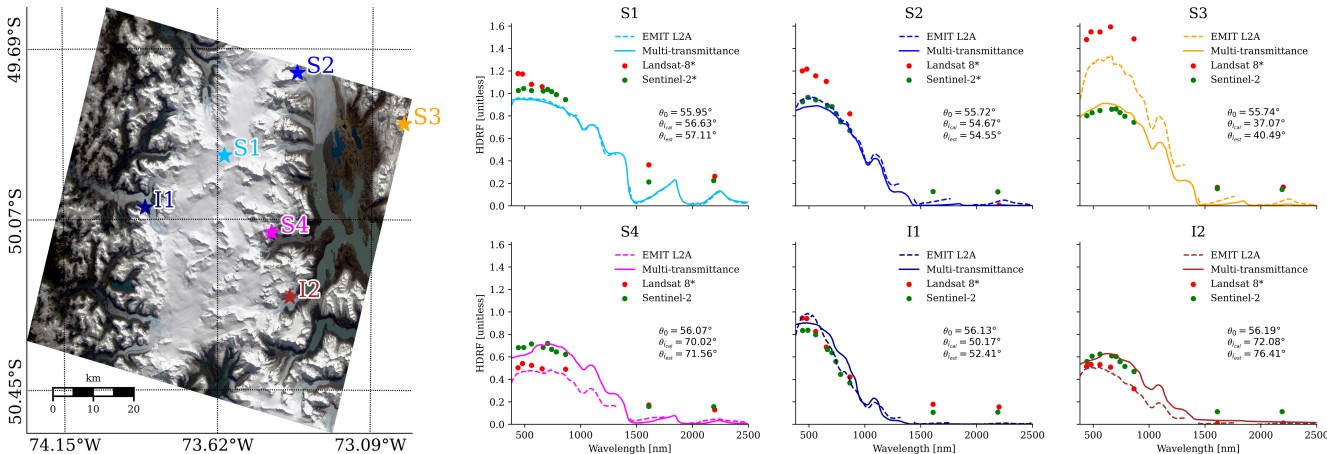

**Figure 10.** Selected HDRF for different surface types from the Chilean ice field. Solid lines represent results from the multi-transmittance approach, while dashed lines depict spectra from the EMIT L2A product for comparison. In addition, coincident Landsat 8 and Sentinel-2 reflectance values for each location are shown by the red and green points, respectively. Asterisks at their labels indicate pixels without topography correction applied. S1: dry snow, S2: melting snow, S3: dusty snow, S4: algal snow, I1: blue ice, I2: dark ice. All panels are complemented by $\theta_0$, $\theta_{i_{cal}}$, and $\theta_{i_{est}}$. Locations are shown in the true-color image in the left panel. (Adapted from Bohn et al. (2023). Copyright [2023] by IEEE. Adapted with permission.)

featuring dry snow with only 77 $\mu$m grain size, no liquid water, and no present impurities. Locations S2 - S4 represent melting snow with larger grain sizes between 200 and 700 $\mu$m, and more or less LAP contamination on the Eastern slope of the ice
field. Location I1 flags blue, clean ice at the Western part, and location I2 designates a dark glacier outflow containing both dust particles and algal cells on the Eastern face. The derived atmospheric state of AOD and water vapor accords well with pressure elevation for all examples.

    Table 2 also shows posterior uncertainties for each retrieved parameter in standard deviation as reported by the OE framework. Errors are generally low, in particular for quantities exhibiting distinct absorption features in the reflectance spectrum,
such as water vapor, grain size, or algae. Higher uncertainties are reported especially for AOD, but also for retrieved dust concentration. It has to be noted though that the linearized posterior error predictions by OE are likely optimistic. OE presumes a linearized version of the forward model, with a local multivariate Gaussian error prediction. This can not only lead to error distributions extending into the negative value range, but also ignores other local minima solutions, if they exist, and even the local estimate may under predict errors (Hobbs et al., 2017; Cressie, 2018). However, the uncertainty estimates provide at least
a useful hint whether the variance of retrieved values is reasonable or not.

    The retrieved HDRF spectra show characteristic shapes and magnitudes depending on grain size and LAP concentration, as shown in Fig. 10. The selected distinction between snow and ice is supported by the near zero reflectance in the SWIR wavelengths for locations I1 and I2, which indicates bare ice surfaces with large grain sizes. The subplots represent results from both the EMIT L2A product and our multi-transmittance framework for each of the six locations. Higher abundances

of LAPs lead to clearly enhanced absorption in the VIS wavelengths, as shown in locations S3, S4, and I1. Additionally, all panels are complemented by $\theta_0$ as well as estimated $\theta_{i_{est}}$ and calculated DEM-based $\theta_{i_{cal}}$. We observe a good agreement in our six examples between $\theta_{i_{est}}$ and $\theta_{i_{cal}}$ with only marginal deviations of up to $4°$. This is confirmed by looking at the regression analysis of all 995,372 snow covered pixels in the image, which shows an $R^2$ of 0.64 and an RMSE of $3.58°$ between $\theta_{i_{est}}$ and $\theta_{i_{cal}}$. We also notice that the larger the difference between $\theta_0$ and $\theta_i$, the more significant the change in HDRF magnitude is for the multi-transmittance approach in comparison to EMIT L2A spectra. However, the shape of the retrieved HDRF also responds to different assumed angles of direct illumination, particularly in the VIS between 380 and 700 nm. S3 is the most significant example showing remarkably different reflectance slopes in the blue wavelengths. Assuming $\theta_i = \theta_0$ leads the retrieval to model both much higher HDRF and more absorption in the short VIS spectral range, because $\theta_i$ is actually $15 - 20°$ smaller than $\theta_0$, i.e., representing a bright, sun-facing slope of the ice field. An interesting note in S4 is that both spectra retain the fine algal chlorophyll absorption feature at $680$ nm, despite having different magnitudes and shapes in the VIS blue. This confirms our findings from Sect. 4.2.2 that the retrieval of biological impurities is less affected by surface topography and anisotropy. In general, locations S1, S2, and I1 attest the validity of the multi-transmittance approach since retrieved HDRF match the EMIT L2A spectra for nearly horizontal surfaces, where $\theta_i \approx \theta_0$.

To further validate the performance of the multi-transmittance approach, Fig. 10 is complemented by reflectance values derived from Landsat 8 and Sentinel-2 images, acquired over the six selected locations on September 14 and 12, 2022, respectively. Landsat 8 observed the region two days later than EMIT, but we believe that it is still valuable for a comparison. Note that standard L2 reflectance products from Landsat 8 are not corrected for topography (Yin et al., 2022), while those from Sentinel-2 include a terrain correction accounting for local slope and aspect (Louis et al., 2021; Santini and Palombo, 2022). This comparison provides an additional indication for the accuracy of the multi-transmittance derived reflectance spectra, given by their good agreement with Sentinel-2 results, particularly for locations S3, S4, and I2, which exhibit surface tilts of $10-20°$. In contrast, the non-corrected Landsat 8 data rather follow the results from the EMIT L2A product.

Next, we calculated LAP radiative forcing in W m$^{-2}$ for each of the six locations for both the EMIT L2A spectra and the multi-transmittance HDRF (Table 3). The result for S3 highlights that LAP radiative forcing is more than $400$ Wm$^{-2}$ higher on sun-facing slopes if snow surface physics are neglected in the forward model. Not accounting for $\theta_i < \theta_0$ causes a steeper slope in the estimated blue reflectance, which resembles LAP absorption. On pixels with $\theta_i > \theta_0$, e.g., locations S4 and I2, radiative forcing estimated from EMIT L2A spectra is generally smaller. The assumed underestimation ranges between $140$ and $207$ W m$^{-2}$ in our examples. As expected, the differences are smaller for locations S1, S2, and I1, where $\theta_i \approx \theta_0$.

### 4.2.4 Blue wavelengths

Due to their absorptivity, LAPs cause a downturn in reflectance in the VIS, particularly in the blue wavelengths. However, this effect can be intensified on bright, sun-facing slopes when omitting topography and anisotropy in the retrieval. A prominent example is location S3 in Fig. 10. We conducted a preliminary sensitivity analysis for this location to emphasize the influence of an incorrectly assumed $\theta_i$ compared to atmospheric AOD, which impacts similar wavelengths and could also cause low reflectance values in the blue when erroneously retrieved. Our selection of AOD for the comparison is supported by the

**Table 3.** LAP radiative forcing in W m$^{-2}$ calculated from the HDRF as displayed in Fig. 10, both for the multi-transmittance and the EMIT L2A results. The third column highlights the difference between the two approaches. S1: dry snow, S2: melting snow, S3: dusty snow, S4: algal snow, I1: blue ice, I2: dark ice. (Reprinted from Bohn et al. (2023). Copyright [2023] by IEEE. Reprinted with permission.)

|  | Multi-transmittance | EMIT L2A | Difference |
|---|---|---|---|
| S1 | 0 | 13 | $-13$ |
| S2 | 32 | 60 | $-28$ |
| S3 | 161 | 601 | $-440$ |
| S4 | 303 | 96 | $+207$ |
| I1 | 0 | 39 | $-39$ |
| I2 | 166 | 26 | $+140$ |

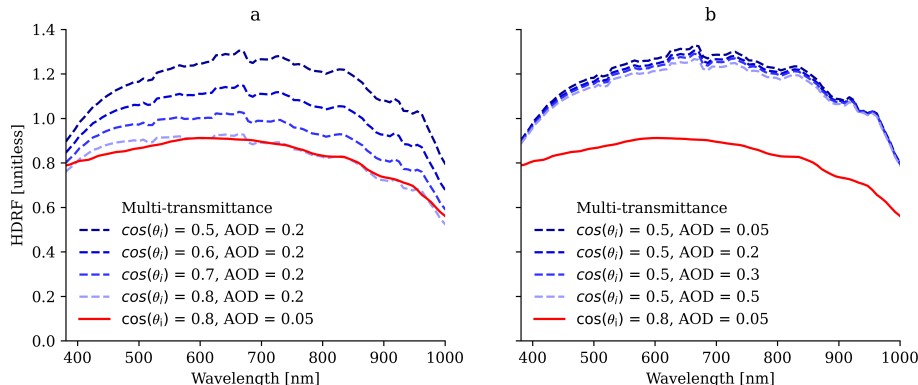

**Figure 11.** Sensitivity of retrieved snow HDRF to erroneous estimations of AOD and $\theta_i$ based on location S3 in Fig. 10. Panel (a) shows results for setting a false AOD, but varying $\theta_i$. Panel (b) displays HDRF retrievals assuming a false $\theta_i$, but varying AOD. The red line depicts the HDRF obtained from running the multi-transmittance model. Note that only the spectral range from 380 to 1000 nm is shown for clarity.

findings from Fig. 9, which show a negative correlation between AOD and both $\theta_i$ and dust. In other words, the retrieval could bare the risk of compensating for a biased AOD by altering the other two parameters and vice versa. Figure 11 shows derived VNIR snow HDRF for different scenarios based on location S3. The solid red line depicts the spectrum retrieved from the multi-transmittance approach with an optimized scaling of direct solar irradiance of $cos(\theta_i) = 0.8$, and $AOD = 0.05$. The dashed blue lines represent HDRF retrievals when setting one of the two parameters to a wrong fixed value, but varying the second one. Figure 11a displays results for assuming a biased AOD of $0.2$, while Fig. 11b contains the outcome when setting $cos(\theta_i)$ to a fixed value of $0.5$. We observe that adjusting $cos(\theta_i)$ to the correct value can almost fully compensate for errors in AOD estimation. The result for $cos(\theta_i) = 0.8$ and $AOD = 0.2$ almost matches the joint solution. In addition, the decrease in blue reflectance is less prominent the closer we approximate $\theta_i$ at the joint solution. In contrast, variations in AOD can not compensate for erroneously assumed $\theta_i$, which supports the conclusion that the anticorrelation between the two properties is

rather one-sided, and a correct assumption or retrieval of the incident angle is more important than a proper characterization of atmospheric aerosols. The latter has traditionally been a particular challenge since most algorithms use strong assumptions about specific scene content that may be absent at coarse ground sampling distance.

## 5 Discussion

The presented retrieval of snow surface properties directly from measured EMIT radiance confirms the reliability of considering a more comprehensive suite of physical effects both in the forward model and in the surface state, found by previous studies (Carmon et al., 2022, 2023). In particular, it facilitates the correction of over- or underestimated HDRF, having the largest impact in the VIS wavelengths. Initial results have been highlighted in Bohn et al. (2023) showing that subsequently estimated LAP radiative forcing likewise improves due to more accurately retrieved dust concentrations, which are less confounded by physical characteristics of the surface when applying the new approach. We significantly extended the sensitivity analysis in this work by confirming accurate performance of our forward model, quantifying the sensitivity of various snow parameters adding snow grain size, algae concentration, and liquid water content, and presenting findings regarding the blue wavelengths based on an investigation of AOD retrieval sensitivity. In the following subsections, we discuss a few important aspects of remote sensing retrievals, which are still not considered yet or only partly solved with the multi-transmittance forward model.

### 5.1 Atmospheric aerosols

Thompson et al. (2018) suggest that jointly estimating atmosphere and surface state can facilitate a more accurate retrieval of AOD. However, it is still heavily dependent on the aerosol optical properties as defined within the utilized atmosphere radiative transfer model. Being able to accurately retrieve the local incident angle directly from the measured radiance though, as shown by Carmon et al. (2023) and Wilder et al. (2024), and confirmed by our manuscript, adds one constraint to the shape of the reflectance in the blue wavelengths, facilitating a more reliable retrieval of AOD. At the same time, we confirm conclusions from previous publications (e.g., Picard et al. (2020)) that considering the local topography, i.e., observation and illumination geometry, enables a more accurate modeling of the blue VIS wavelengths in remotely sensed snow HDRF, when retrieved over challenging mountainous terrain. Applying a topography-aware forward model that incorporates a physical modeling of snow HDRF takes us much closer to the intrinsic reflectance of the surface, and simultaneously, improves the estimation of atmospheric AOD and inorganic LAP in snow. However, the blue VIS wavelengths in retrieved snow HDRF remain a major source of uncertainty as several other confounding factors beyond topography could lead to the formation of the so-called "blue hook", which manifests in an artificial downward or upward trend in the shortest wavelengths of the reflectance spectrum (Painter and Dozier, 2004b; Naegeli et al., 2015; Di Mauro et al., 2017; Picard et al., 2020; Kokhanovsky et al., 2022). A detailed investigation of the hooking behavior, including examples from different sensors, is outlined in Bair et al. (2024), and future work still needs to target an improved understanding of the cross-correlation between atmospheric aerosols, dust particles on the surface, and the blue wavelengths in snow HDRF.

## 5.2 Snow and LAP optical properties

The initial step in our snow surface model is to obtain the optical properties (OPs) of the snow-LAP mixture. We apply Mie scattering theory to obtain the ice OPs, and follow the approach of Grenfell and Warren (1999) to model non-spherical snow particles by a collection of independent spheres that has the same volume-to-surface-area ratio as the non-spherical particles. This assumption might be inappropriate, but studies have shown that this approach provides an accurate representation of extinction efficiency and single-scattering albedo, while only the scattering asymmetry factor is usually overestimated (Grenfell and Warren, 1999; Neshyba et al., 2003; Grenfell et al., 2005; Warren, 2019). Its effect on bulk optical properties can be compensated though by reducing the grain size of the model (Dang et al., 2016). Nevertheless, more accurate representations of the optical shape of snow as well as more realistic snow BRDF datasets certainly exist (e.g., see Malinka (2014); Dumont et al. (2021); Malinka (2023)), but we would like to defer the testing of various representations of snow and ice optical properties in our retrieval framework to a subsequent study. We consider the current manuscript rather as a concept study for the simultaneous retrieval of atmosphere and physical surface properties directly from at-sensor radiance. However, we have to be aware of potential misrepresentations in many cases. For instance, the glacier outflows from the Chilean ice field are likely not well represented by applying Mie theory. The grains on bare ice surfaces typically appear to be arbitrarily shaped with irregular dimensions, so that, e.g., a Geometric Optics approach based on ray-tracing would be more appropriate to model their OPs (Kokhanovsky and Zege, 2004; Cook et al., 2020; Bohn et al., 2022). In future work, we need to compare retrieval results from assuming different grain shapes, including spheres, spheroids, hexagonal plates, and Koch snowflakes (He et al., 2017; Hao et al., 2023). We will also apply different approaches to model ice layers, e.g., with enclosed air bubbles and a Fresnel layer between the ice and a thin snow cover (Whicker et al., 2022).

To model biological LAPs, we utilize a set of algae OPs for the species *Ancylonema* (glacier algae) as well as *Sanguina nivaloides* and *Chloromonas nivalis* (snow algae), derived from samples collected on the Greenland Ice Sheet (Chevrollier et al., 2022). Despite being characterized at a different geographic location far away from our study site, we assume that these OPs adequately represent algae cells found on ice sheets, glaciers, and snow worldwide. This is corroborated by previous studies that identified those three species as being responsible for the darkening of snow and ice surfaces in various regions, including the Greenland Ice Sheet, Svalbard, the European Alps, and the Sierra Nevada in California (Yallop et al., 2012; Remias et al., 2012; Di Mauro et al., 2020; Painter et al., 2001). Moreover, Takeuchi and Kohshima (2004) and Kohshima et al. (2007) identified *Ancylonema* and *Chloromonas* algae as among the most frequently encountered species on the Patagonian Ice Sheet.

The use of dust OPs poses a different challenge, as they strongly depend on mineralogy and source area (Di Biagio et al., 2019). Several sets of dust OPs from different geographic regions, derived using diverse techniques and data, are publicly available. They have been obtained from any combination of field samples, spectral measurements, and linear mixing modeling, with Sahara, Colorado, Greenland, and Mars being the most prominent regional types (Polashenski et al., 2015; Skiles et al., 2017; Balkanski et al., 2007; Singh and Flanner, 2016). However, only a few studies have considered specific dust minerals when assessing their impact on snow melt (Lawrence et al., 2010; Kaspari et al., 2014; Reynolds et al., 2014). For our study,

we selected only one type of dust OPs representing rather large particles, measured from samples that were collected in the San Juan Mountains of southwestern Colorado (Skiles et al., 2017). In the lack of dust OP characterization in South America, we believe that the Colorado type is closest to the dust type found in Patagonia. This is especially supported by the finding that very large dust particles are often present in patchy snow of arid environments (Skiles et al., 2017). Moreover, studies of the geochemical composition and mineralogy suggest that both the San Juan and the Patagonian dust are significantly dominated by quartz with 30-50% of the total mineral mass (Lawrence et al., 2010; Demasy et al., 2024). Such analyses will be facilitated on even larger geographical scales by the EMIT mission objective, which is providing an improved understanding of the mineralogy of dust particle source regions, and enabling an enhanced identification and classification of dust OPs and their distribution around the Earth's snow-covered areas (Connelly et al., 2021; Gonçalves Ageitos et al., 2023).

## 5.3 Mixed pixels

The proposed method neglects the issue of mixed pixels, which is particularly of interest in mountainous regions with subpixel mixtures of snow, vegetation, and rock (Painter et al., 2003). Apparent rock or soil fractions within a pixel very likely bias the retrieval of dust concentration and could lead to erroneously high values. Figures 6b and c show small grain sizes of $100 - 200 \ \mu m$ for dust concentrations of $60 - 100 \ \mu g \ g^{-1}$ at the edges of the snow cover. For comparison, Skiles and Painter (2017) measured snow grain sizes ranging around $400 \ \mu m$ under those dust concentrations. On the other hand, they also observed that high dust loadings could weaken the ice absorption feature at $1030 \ nm$ leading to a decreasing trend in remotely sensed grain size (see also Fig. 5c). However, field measurements suggest that this effect is caused by at-surface dust particles, but not supported by high-dust content in the subsurface of the snow pack (Skiles and Painter, 2017). We currently apply an NDSI threshold of $0.4$ to determine snow-covered pixels. This threshold includes only partially snow-covered pixels as well though, and should optimally be more adjusted to the instrument characteristics. For instance, Seidel et al. (2016) used $NDSI >= 0.90 - 0.93$, but for airborne AVIRIS measurements with higher spectral and spatial resolution than EMIT. A more sophisticated alternative would be to calculate snow fractional cover by including it in the state vector, and use a respective minimum value as constraint. For multispectral sensors, Bair et al. (2021) developed the Snow Property Inversion from Remote Sensing (SPIReS) that only makes dust estimates for pixels with snow fractional cover $>= 0.90$ and then spatially interpolate to pixels with surface type mixtures. To benefit from the high spectral resolution of imaging spectrometers such as EMIT, a required improvement of our approach would be the combination with a surface mixture model. This would provide reliable dust estimates even at lower fractional cover or NDSI values. On the other hand, this might also increase the processing time significantly.

## 5.4 Field validation

One essential part of remote sensing retrievals is still missing for the presented multi-transmittance approach, which is the validation with field measurements. This can be explained by the remoteness of our study sites in South America, but also by the fact that this work has rather been designed as a theoretical proof of concept. A good indication for the accuracy of our proposed inversion method is already given by the comparison to Landsat 8 and Sentinel-2 data in Sect. 4.2.3. However, to

further compensate for the lack of validation and to put our results into the right context, we provide a comparison to findings from previous studies.

We look at the estimated error in radiative forcing when topography and anisotropy are not considered in the forward model. Previous work in the Chilean and Argentinian Andes reports daily or annual averages of LAP radiative forcing, or even the mean over a period of multiple years (Rowe et al., 2019; Cordero et al., 2022; Figueroa-Villanueva et al., 2023). Their values range between 0 and 10 W m$^{-2}$, mainly investigating the influence of black carbon, which was not the focus of our study. In contrast, our method provides the instantaneous radiative forcing due to LAP, which allows a reasonable comparison to similar

work conducted in the Sierra Nevada, CA, or the Rocky Mountains, CO (Painter et al., 2013b; Seidel et al., 2016). Even though estimated in a different geographical location, their values of up to 400 W m$^{-2}$ agree well with the range of LAP radiative forcing retrieved from the multi-transmittance approach (see Table 3).

## 5.5   Do we still need reflectance?

Remote sensing retrievals that utilize measurements of reflected sunlight will always need a reflectance term within the forward

model as this is the essential quantity to summarize the radiative transfer of photons through a material. However, our findings confirm that this term needs to be treated carefully and should either refer to the intrinsic reflectivity of the target material or even more accurately, be defined as a bi-directional reflectance distribution function (BRDF) providing coupled, angle- and illumination-dependent reflectance terms (Vermote et al., 1997; Schaepman-Strub et al., 2006; Verhoef and Bach, 2007; Guanter et al., 2009). Furthermore, our study shows that using surface reflectance as a modeled quantity within the forward

model rather than retrieving it as a free parameter improves the accuracy of estimated biophysical snow properties. In the absence of well-parametrized surface radiative transfer models, this can be balanced by a clear definition and consistent use of the surface reflectance term.

## 6   Conclusions

We introduce a new retrieval algorithm that estimates snow surface properties directly from at-sensor radiance measured by

the spaceborne EMIT imaging spectrometer. We utilize a coupled full physics snow and atmosphere model and apply Optimal Estimation to solve for the most probable surface state. On one hand, this allows to reduce the number of retrieved state vector elements to only a handful of snow surface and atmosphere properties, including AOD, water vapor, snow grain size, liquid water content, LAP concentration, and local solar zenith angle. On the other, it facilitates a more thorough consideration of physical surface characteristics such as anisotropy and topography. We utilize two representative EMIT images acquired over

the Argentine plain and the Chilean ice field in Patagonia, South America, to conduct a sensitivity analysis of our proposed forward model. We demonstrate that the retrieval of snow liquid water fraction and snow algae concentration is insensitive to topographic and directional effects. In contrast, estimations of snow grain size and mineral dust mass mixing ratio can be biased under these scenarios, which directly propagates into incorrect calculations of LAP radiative forcing. A validation with field measurements is still missing for the presented approach, but comparisons to Landsat 8 and Sentinel-2 reflectance values

as well as to estimates of radiative forcing of previous studies indicate that surface reflectance as an intermediate quantity can be omitted in the retrieval framework in favor of inferring surface properties directly from radiance. Surface reflectance is still needed as a modeled quantity within the forward model but must be consistent and clearly defined. Finally, we evidence that erroneous assumptions about surface topography are one of the -though not the only- causes for the formation of the "blue hook" in remotely sensed retrievals of snow reflectance. Future work must include a thorough validation effort, and needs to

address mixed pixels and the modeling of both ice and LAP OPs to account for local geographical characteristics. Nevertheless, our findings are critical for for the conception of retrieval algorithms for future orbital imaging spectroscopy missions, such as NASA's SBG. These missions provide the framework to develop and enhance processing schemes and retrieval algorithms on a global scale.

*Code and data availability.* Software code used in this study is available in the ISOFIT GitHub repository at https://github.com/isofit/isofit.

Data sets from the EMIT imaging spectrometer used in this study are available at https://earth.jpl.nasa.gov/emit/data/data-portal/coverage-and-forecasts/.

*Author contributions.* N. Bohn: Conceptualization, Methodology, Investigation, Data curation, Writing – original draft preparation. E. H. Bair: Conceptualization, Methodology, Investigation, Writing - review and editing. P. G. Brodrick: Conceptualization, Methodology, Data curation, Funding acquisition, Writing - review and editing. N. Carmon: Methodology, Writing - review and editing. R. O. Green: Funding

acquisition, Writing - review and editing. T. H. Painter: Methodology, Writing - review and editing. D. R. Thompson: Methodology, Funding acquisition, Writing - review and editing.

*Competing interests.* The authors declare that they have no known competing financial interests or personal relationships that could have appeared to influence the work reported in this paper.

*Acknowledgements.* The High Performance Computing resources used in this investigation were provided by funding from the JPL Infor-

610 mation and Technology Solutions Directorate. Portions of this work were funded by the Earth Surface Mineral Dust Source Investigation (EMIT), a NASA Earth Ventures-Instrument (EVI-4) Mission, as well as through a NASA Research Announcement NNH20ZDA001N for Research Opportunities in Space and Earth Science grant (20-OSTFL20-0002) for the development of the ISOFIT code base. The research was carried out at the Jet Propulsion Laboratory, California Institute of Technology, under a contract with the National Aeronautics and Space Administration (80NM0018D0004).

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
