# Peer review of "Do we still need reflectance? From radiance to snow properties in mountainous terrain: a case study with EMIT"

_EGUsphere, 2024_

## Referee Comment (RC3)

Review of « The pitfalls of ignoring topography in snow retrievals: a case study with EMIT», by Niklas Bohn et al. Review by Quentin Libois.

**General comments**

This paper presents a novel retrieval algorithm based on optimal estimation to retrieve snow properties (grain size, dust and algae contents, liquid water content) from hyperspectral satellite images in the solar spectrum. The main originality of the algorithm is to account for local topography by including the slope angle in the retrieved parameters. The algorithm is applied to two images acquired above Patagonia by the spaceborne instrument EMIT. Accounting for the local slope has a significant impact of the dust and grain size retrievals, but is less critical for algae and liquid water content which are characterized by localized spectral features. These retrievals are also used to compute the instantaneous radiative forcing of light absorbing particles, which is also very sensitive to slope effects. The spatial gradients of the retrieved quantities are discussed on a physical basis, supporting the reliability of the retrievals. A preliminary sensitivity study is performed, which highlights how the aerosol optical depth in the atmosphere and the slope angle impact the snow reflectance. The apparent drop in the blue range of snow spectra is discussed in details to highlight that topography is largely responsible for this common feature of snow reflectance as measured from space. Some limitations of the current algorithm and suggestions for improvements are also provided.

The paper is overall well written, although the abstract and introduction could probably be greatly improved, in particular to better remind the existing strategies already used to account for topography in snow retrievals from space and the link between the present work and previous studies from the same authors. More technical information about EMIT would be appreciated and the relevance of the retrieval algorithm to other spaceborne instruments could be elaborated. Some technical details lack in the presentation of the algorithm but could easily be provided. My main concern is about the representation of snow by a collection of spheres to simulate the directional reflectance. This issue is only very briefly mentioned in the discussion, without any quantification of the potential impacts. It certainly deserves much more attention. Also, the accuracy of the retrievals is not discussed, while the theoretical framework used would make it easy to investigate, and would strengthen the conclusion that retrievals of LAP are possible with EMIT while they were apparently challenging with previous spaceborne instruments. For these reasons I recommend that these points be carefully treated before the paper can be considered for publication.

**Specific comments**

1) The abstract is probably too long. It could more efficiently start with the relevance of monitoring LAP and grain size. Also the objective is not clearly stated, neither the main novelty compared to previous work. In general it is not very clear and would deserve some general rewriting (see some suggestions in the technical comments).

2) The introduction as well could notably be improved. What is mainly missing is information about algorithms already used for LAP retrieval, and to account for topography (e.g. Picard et al., 2020). EMIT is selected because it supposedly has a larger SNR so what would happen if existing algorithms were applied to EMIT? Why such a motivation to build a new retrieval algorithm? This should be better motivated. Also, how does this study complement recent previous work from the same authors?

3) EMIT is central in the present study. However it is nowhere described in details. In particular information on the spatial resolution (which is very critical) is lacking. Likewise, information about

its spectral resolution and radiometric accuracy would be very useful. An important question being: why using EMIT and not any other spaceborne hyperspectral (or multi-spectral if it would be enough for the purpose of the study) sensor.

4) The authors represent snow as a collection of spheres, although it has been known for a long time that this is not appropriate, in particular to describe the anisotropy of snow reflectance. The quantitative impact of such an assumption on the retrievals is not investigated, which is detrimental to the overall quality and impact of the study. I'd encourage the authors to test the retrievals with other datasets of snow directional reflectance (from observations or models depending on the availability of the data, see some suggestions in the technical comments). Also, adding a figure to illustrate the snow HDRF used in the forward model would probably help the interpretation of the spectra and their sensitivity to illumination and viewing geometry

5) I feel like some technical details in the algorithm are missing. First, the equation for the forward model would deserve more physical explanations, beyond a reference to a paper. Then, snow grain size is not properly defined. Is it a radius, a diameter, averaged or effective over a prescribed size distribution, or on the contrary a monodisperse collection? As a consequence, the way liquid water content is accounted for is not sufficiently clear. Likewise it is not clear if the treatment of LAP relies on mass absorption coefficients or any other optical quantity. The instrumental noise is not detailed either while some quantitative information to highlight the high SNR would be appreciated.

6) One originality of the retrieval algorithm is to use the optimal estimation in combination with a forward model to retrieve the parameters of a state vector (instead of retrieving for instance reflectances). However only the most probable solutions of the problems are presented, without any reference to the associated uncertainties. Given that the objective of the study is to demonstrate that EMIT can be used to retrieve snow properties that are not accessible with other instruments, mentioning uncertainties is key to convince the reader that the retrieved quantities are reliable, in particular given that there is no ground truth. For instance it is quite questioning that the retrieved dust quantities can be strictly zero (Table 2). Beyond the uncertainties I'd encourage the authors to further investigate the correlations between the retrieved parameters, which could help understand how compensation between variables can affect the quality of the retrievals.

**Technical comments**

l.2: solar radiation would be better than illumination

l.3: at negative temperature melting is not critical, but metamorphism is

l.6: maybe specify LAPs on/in snow?

l.11: not clear what dust properties EMIT measures

l.12: what is a "target mask"?

l.15: anisotropy of what?

l.16: why "forward scattering"? Not clear

l.18-20: quite difficult to understand in an abstract

l.21: it would have been helpful to detail earlier (e.g. l.15) what are the snow properties to be retrieved

l.22: use µg g$^{-1}$ instead, as well as for all units

l.23: such a forcing seems huge! It's because it's instantaneous.

l.25: is the "blue hook" something sufficiently well known (it is not to me) to appear as is in an abstract?

l.25-26: the link with runoff and climate models is definitely not obvious. Either to be removed, or expanded

l.30: I don't see why having the highest albedo of all natural surfaces is a reason for playing a key role...

l.31: "cooling effect" is a bit surprising to read. Snow does not cool the Earth, or at least it depends with respect to what? Ok if you say "more snow-covered surfaces will tend to cool the Earth"

l.37: LAPs are not the only reason for snow darkening (or at least albedo decrease). Metamorphism has a similar effect

l.39: Sun's energy is unclear → where the solar spectrum peaks? Where the sun irradiance is maximum?

l.59: not clear what "not tied to physical units" means

l.61, 62: what are EnMAP, PRISMA?

l.68: not clear what these references correspond to

l.69: absorption spectral features?

l.73: not clear why mapping arid surfaces informs about transport and radiative forcing. EMIT should be more clearly introduced

l.83: the transition from the Lambertian assumption issue to the topography issue is too fast. Is there a link between both?

l.89: what does "rapidly shifting terrain" mean?

l.91: at first order and satellite footprint scale the mountain topography probably dominates snow depth variability, nope?

l.95: could you explain what does this algorithm

l.96-97: not clear what is the atmospheric radiative code and the snow one. Also it suggests that 3D effects (reillumination by neighboring slopes) are not accounted for by such a model. Do you confirm?

l.123: what tool was used to compute these HDRF?

l.138: it's not obvious to me why the backward reflectance decreases with less direct irradiance (it means comparing backward and side scattering). Some explanation detailing the equivalent incidence angle of diffuse illumination would be helpful

l.141: could you clarify whether "scattering by surrounding objects" can actually be modeled.

l.144: I regret that EMIT has not been introduced before in more details. In particular its spatial resolution seems to be a critical quantity if it is meant to see independently distinct mountain slopes instead of a mixture of various slopes.

l.163: I think units (here and elsewhere) should not be italic

l.172: can you clarify whether you invert independently the individual pixels, or not.

Eq. (2): I think it could be better explained in terms of the various contributions. Also, transmittance is a physical property (of the atmosphere for instance). Here it seems that it includes the partition between direct and diffuse irradiance. I'd recommend to explicitly mention the direct/diffuse partition. Also, I don't know what "atmospheric path radiance" is. Is it related to the spherical albedo of the atmosphere? By the way spherical albedo has not been defined before.

l.185: I'd expect the HDRF to depend also on the direct/diffuse partition. Regarding the incidence angle what is the motivation to have it in the state vector instead of using a DEM? How would the results with fixed vs retrieved incidence angle compare?

l.186-187: Not very clear. Do you mean that previously the HDRF was in the surface state vector? Also I'm afraid to read that you assume spherical particles for snow (confirmed l. 198), which are very inappropriate, in particular when it comes to computing HDRF. Database exist for more realistic snow BRDF data (from either measurements of models). If not detailed elsewhere, could you clarify how many snow layers you use in the model.

l.187: how many streams are used for the DISORT simulations?

l.190: do you mean that the dimension was larger previously due to the multispectral dimension?

l.197: not only the asymmetry parameter matters, but also the detailed phase function for that kind of applications

l.199: how is then defined the snow grain size? Including the liquid water coating? What about the size distribution of snow particles?

l.204: would you have any reference to support that algae are similar in Greenland and Patagonia? Otherwise why would you believe this? Also could you clarify what optical property is defined. Only a mass absorption coefficient? The same question holds for dust.

l.213: what do you mean by "atmospheric aerosols"? Those assumed in MODTRAN?

l.222: it should be clear what measurements are included here. Multi spectral or also multi-pixels?

l.226: then why using a prior at all if in the end it does not constrain the cost function?

l.227: for the model to be well-posed it should be proved that measurements at distinct wavelengths are actually independent, and the number of spectral channels should be mentioned (to be compared to the number of parameters to be retrieved).

Eq. (4): here again this factor is very dependent on the actual phase function of snow, which is likely to be different than that of spheres.

l.230: the main advantage of optimal estimation is to provide an estimation of the posterior error, which is not discussed at all. It would be worth adding this uncertainty range for the retrieved parameters of interest.

l.248: I think the units should be like $W\ m^{-2}$.

l.264: how do EMIT spatial resolution and SRTM match (or not)? Is SRTM averaged somehow to find incidence angles comparable to EMIT retrievals?

l.268: the correlation between snow grain size and slope is tricky. You could either argue that the retrieval is homogeneous for snow grain size in a mountainous terrain with various slopes, suggesting that accounting for slope corrects for an apparent heterogeneity of snow grain size when assuming flat terrain. Or you give a physical reason why snow grain size can differ depending on the slope… Looking at the correlations between retrieved parameters may help clarify this point. A too strong correlation may indicate compensation between both variables.

l.271: I would not necessarily say that snow grains of 200 microns (at least if it is the radius) are small.

Fig.7: it should be clear somewhere that EMIT L2A is the standard EMIT product ignoring topography

l.299: it is not clear what the single-transmittance model is (what wavelength?).

l.303: I think this "hook" behavior should be better identified in the figure. Is it the too strong decrease in the blue visible in the HDRF? I believe the direct/diffuse partition, that greatly changes in this spectral range, if not properly accounted for can also contribute to this hook.

l.311: I don't see in this paragraph the sensitivity to assuming a Lambertian snow surface. Unless both impacts are combined altogether. In this case it would be worth separating both to disentangle the impacts, and point what assumptions is most critical.

l.325: only here is the direct/diffuse partition explicitly mentioned, while I think it would be valuable to clarify its treatment and impact earlier on.

l.333: the 3 digits may be a lot for such an estimation.

Table 2: any comment on the fact that algae can be zero somewhere, and present elsewhere?

l.375: "small" is awkward. Preliminary?

l.376: this point suggests that their could be correlations between the retrieved parameters. You could look at these correlations to inform about the independence (or not) of the retrieved parameters, which is trivial with optimal estimation. The underlying question being for instance:

can the retrieval algorithm return stronger AOD and lower LAP in snow, which may result in more or less the same apparent radiance at TOA?

l.382: this suggests that AOD cannot be accurately retrieved, unless it is the blue end of the spectrum that puts most constraint on AOD (rather than the longer wavelengths). Could you expand on that?

l.395: AOT or AOD?

l.399: what would be the impact of not considering blue wavelenghts in the retrievals? What variables would be most affected, and to which extent?

l.401: this physical explanation for the blue hook could have been given earlier on, and a bit more detailed.

l.402: can you expand on these laboratory measurements?

l.411: where does this assumption come from?

l.412:  you could also refer to Picard et al. (2016) who suggest absorption is in between Warren and Brandt (2008) and Warren (1984).

l.423:  on which basis do you argue that the spherical assumption is the best general shape? As a suggestion, Malinka (2014, 2023) has developed a general mixture model that works very well to estimate the "optical shape" of snow. Maybe it's relevant as well for snow BRDF. See also Dumont et al. (2021).

l.426: much larger than what? Why couldn't it be large spherical particles?

l.427: It's definitely a good idea, and I would strongly suggest to further investigate this in the present paper.

l.430: I guess one of the EMIT objectives is to map this variability in dust optical properties, so it might be worth referring to this and directly related studies.

l.442: as the spatial resolution has never been discussed it's hard to guess how critical are these mixed pixels.

l.453: how would you calculate snow fractional cover? By including it in the state vector?

l.471: the link between this work and melt runoff and climate model input are not clear, but this might be clarified if it sounds important to the authors.

**References**

Dumont, M., Flin, F., Malinka, A., Brissaud, O., Hagenmuller, P., Lapalus, P., ... & Ando, E. (2021). Experimental and model-based investigation of the links between snow bidirectional reflectance and snow microstructure. *The Cryosphere, 15*(8), 3921-3948.

Malinka, A. V. (2014). Light scattering in porous materials: Geometrical optics and stereological approach. *Journal of Quantitative Spectroscopy and Radiative Transfer, 141*, 14-23.

Malinka, A. (2023). Stereological approach to radiative transfer in porous materials. Application to the optics of snow. *Journal of Quantitative Spectroscopy and Radiative Transfer, 295*, 108410.

Picard, G., Libois, Q., & Arnaud, L. (2016). Refinement of the ice absorption spectrum in the visible using radiance profile measurements in Antarctic snow. *The Cryosphere, 10*(6), 2655-2672.

Picard, G., Dumont, M., Lamare, M., Tuzet, F., Larue, F., Pirazzini, R., & Arnaud, L. (2020). Spectral albedo measurements over snow-covered slopes: theory and slope effect corrections. *The Cryosphere, 14*(5), 1497-1517.

---

## Author Comment (AC1)

**RC1**: Anonymous Referee #1

The manuscript from Bohn et al. deals with a relevant problem of imaging spectroscopy of snow and ice: topography. The authors developed a new methodology to correct imaging spectroscopy data acquired from the EMIT satellite mission for the effect of topography and they conclude that high error in LAPs-induced radiative forcing estimates are possible if the topographic effect is neglected. The study is based on two EMIT scenes in Chile and Argentina without any field validation of the retrieval. This issue partly weakens the outcomes of this study, and I suggest to refine the conclusions in accordance. Several hyperspectral satellite mission will be launched in the future (e.g. SBG, CHIME) and new data will be available for retrieval of surface parameters of snow and ice. The results of this manuscript raise important questions regarding the uncorrected topographic effect in the context of parameter retrieval. I think that the manuscript is interesting both for the cryospheric and remote sensing community, and it can be accepted only after minor comments listed below are taken into account.

We thank the referee for the positive feedback and the constructive review. We updated our conclusion by adding the following statements:

*"A validation with field measurements is still missing for the presented approach, but comparisons to Landsat 9 and Sentinel-2 reflectance values as well as to estimates of radiative forcing of previous studies indicate an accurate performance of the retrieval framework."*

And:

*"Future work must include a thorough validation effort, and …"*

Below, we address the line-by-line comments by providing respective responses and by indicating the changes to the manuscript.

lines 69-70: What are the expected signal-to-noise ratio for those two missions? Please provide some numbers.

The expected signal-to-noise performance for both SBG and CHIME is $\geq 400$ in the VNIR and $\geq 250$ in the SWIR. We added these numbers:

*"Future orbital imaging spectroscopy missions, such as NASA's Surface Biology and Geology (SBG) (National Academies of Sciences, Engineering, and Medicine, 2018) and ESA's Copernicus Hyperspectral Imaging Mission for the Environment (CHIME) (Rast et al., 2019) will address this problem by providing high signal-to-noise ratios (SNR) of more than 400 in the visible-to-near-infrared (VNIR) and more than 250 in the shortwave-infrared (SWIR), as well as high spectral and spatial resolution."*

In the introduction, I suggest to add a brief discussion on the attempts that have been made to model snow albedo in complex topography (e.g. Picard et al. 2020). In fact, those studies already show the "blue hook" that is described later in your manuscript.

Thanks for this suggestion! We agree and extended the discussion about modeling snow albedo in complex terrain by adding references to Picard et al. (2020) and Donahue et al. (2023):

*"Specifically, Picard et al. (2020) demonstrated the sensitivity of snow albedo measurements to surface slope based on spectral data taken in the field, and proposed a correction approach to retrieve the intrinsic albedo. Using a digital elevation model (DEM), this local geometry, including surface slope and aspect, can be calculated and incorporated in atmospheric modeling schemes in order to correct for spectral distortions in the retrieved surface reflectance (Richter and Schläpfer, 2017). However, given the complex terrain of mountainous regions, and the current unavailability of coincident radar/lidar and imaging spectroscopy data in orbit, reliance on fixed DEMs may introduce additional retrieval errors, not only due to variability in local snow depth, but also because of uncertainties in the DEM product itself (Dozier et al., 2022). For instance, Donahue et al. (2023) showed that topographic correction with coarse and non-coincident DEMs introduces significant errors in estimated snow albedo from air- or spaceborne imaging spectroscopy of up to 20%. Overall, a mature and comprehensive modeling of topography for spaceborne imaging spectroscopy data over mountain snow has not yet been demonstrated, and only a few studies have applied a limited post-hoc correction at the airborne scale (Painter et al., 2013a; Seidel et al., 2016)."*

Figure 1: I suggest to add the grain size value (100 um and 1000 um) also in the plot. HDRF should be also displayed in the label, in order to be consistent with the main text.

We updated Fig. 1 accordingly.

line 154: the spatial, spectral and temporal resolutions of EMIT data should be provided here. Furthermore, I suggest to add a scale bare to Figure 3.

We added information about spatial, spectral, and temporal resolution of EMIT images to the introduction:

*"Extending over a wavelength range of 380-2500 nm with a spectral resolution of approximately 7.5 nm, EMIT images provide a pixel size of around 60 m on a 74 km wide swath. The temporal revisit time is variable depending on the orbital cycle of the ISS, and ranges between one day and more than a week (Thompson et al., 2024)."*

We also added a scale bar to Fig. 3.

line 256: this info should be provided in the methods. Which bands have been used to calculate ndsi? You used Ndsi<0.0: this is strange, please verify which threshold that you applied to identify snow/ice areas.

We agree and moved the description of our snow mask to the methods in Sect. 3.1. In addition, we updated our approach of identifying snow-covered pixels by now following the procedure of Dozier (1989). We now use top-of-atmosphere (TOA) reflectance $\rho_{TOA}$ of EMIT bands at 485 nm (blue), 567 nm (green), and 1648 nm (SWIR), and marked a pixel as snow-covered when $\rho_{TOA,485} > 0.16$, $\rho_{TOA,1648} < 0.25$, and NDSI $> 0.4$. The latter is calculated using the bands at $\rho_{TOA,567}$ and $\rho_{TOA,1648}$. We added this information to the text:

*"It has to be noted that our approach is only applicable to snow- and ice-covered pixels. We identified these areas by following the procedure of Dozier (1989). We utilize TOA reflectance rho_TOA of EMIT bands at 485 nm (blue), 567 nm (green), and 1648 nm (SWIR), and considered a pixel as snow-covered when rho_TOA,485 > 0.16, rho_TOA,1648 < 0.25, and*

*NDSI > 0.4. The NDSI is the normalized-difference snow index calculated using rho_TOA,567 and rho_TOA,1648 (Dozier, 1989)."*

Also, we updated Fig. 6 accordingly and added the following:

*"We currently apply an NDSI threshold of 0.4 to determine snow-covered pixels."*

line 265: Geophysically? I never read/heard this term..

We agree that this term sounds strange and removed it:

*"Topography can influence the size of snow grains."*

line 266-268: this is true only during a period of time. When air temperature is low, this may not hold true.

Yes, that is a good point. We added the word 'can' to express that it is a potential process but not always the case:

*"Air temperatures can decrease with increasing elevation, which abates melt processes, causing drier snow with smaller grain size."*

line 282-284: how you can be so sure without any field validation?

That is a fair argument. We removed the respective sentence.

line 333: the variance explained by this regression is very low. The reasoning should be more conservative.

We agree that our statement is disproportionate in this context and reverted to a more conservative reasoning as suggested:

*"Interestingly, the effects on the snow grain size retrieval are of a similar magnitude as on the dust estimations."*

Figure 9: this figure is impactful. I would be very curious to see at least reflectance data from one multispectral mission (Landsat 8-9 or Sentinel 2) acquired in the same period over the same spots. This would confirm that the multi-transmittance approach provides a sound correction for HDRF.

Thanks for the comment, this is a great idea! We pulled the respective L2 surface reflectance images from Landsat 8 and Sentinel-2, which captured our study area under clear sky conditions on September 14 and 12, 2022, respectively. Landsat 8 observed the region two days later than EMIT, but we believe that it is still valuable for a comparison. The comparison to both of these multispectral instruments is indeed of particular interest because standard L2 reflectance products from Landsat 8 are not corrected for topography (Yin et al., 2022), while those from Sentinel-2 are (Louis et al., 2021; Santini & Palombo, 2022). We updated Fig. 9 by adding the Landsat 8 and Sentinel-2 reflectance values for the six selected locations, adjusted the figure caption, and added the following paragraph to Sect. 4.2.3:

*"To further validate the performance of the multi-transmittance approach, Fig. 9 is complemented by reflectance values derived from Landsat 8 and Sentinel-2 images, acquired over the six selected locations on September 14 and 12, 2022, respectively. Landsat 8 observed the region two days later than EMIT, but we believe that it is still valuable for a comparison. Note that standard L2 reflectance products from Landsat 8 are not corrected for topography (Yin et al., 2022), while those from Sentinel-2 include a terrain correction accounting for local slope and aspect (Louis et al., 2021; Santini & Palombo, 2022). This comparison provides an additional indication for the accuracy of the multi-transmittance derived reflectance spectra, given by their good agreement with Sentinel-2 results, particularly for locations S3, S4, and I2, which exhibit surface tilts of 10-20°. In contrast, the non-corrected Landsat 8 data rather follow the results from the EMIT L2A product."*

Section 4.2.3: in general, I like this narrative but I think that the error in Rf estimates should be put in the right context since no field validation data are provided in this study. I suggest at least to compare your estimates with previous results in the same area (e.g. Rowe et al. 2019 and references therein).

In order to address this comment, we added subsection 5.4 'Field validation' to the discussion section with the following content:

*"One essential part of remote sensing retrievals is still missing for the presented multi-transmittance approach, which is the validation with field measurements. This can be explained by the remoteness of our study sites in South America, but also by the fact that this work has rather been designed as a theoretical proof of concept. A good indication for the accuracy of our proposed inversion method is already given by the comparison to Landsat 8 and Sentinel-2 data in Sect. 4.2.3. However, to further compensate for the lack of validation and to put our results into the right context, we provide a comparison to findings from previous studies.*

*We look at the estimated error in radiative forcing when topography and anisotropy are not considered in the forward model. Previous work in the Chilean and Argentinian Andes reports daily or annual averages of LAP radiative forcing, or even the mean over a period of multiple years (Rowe et al., 2019; Cordero et al., 2022; Figueroa-Villanueva et al., 2023). Their values range between 0 and 10 W m$^{-2}$, mainly investigating the influence of black carbon, which was not the focus of our study. In contrast, our method provides the instantaneous radiative forcing due to LAP, which allows a reasonable comparison to similar work conducted in the Sierra Nevada, CA, or the Rocky Mountains, CO (Painter et al., 2013a; Seidel et al., 2016). Even though estimated in a different geographical location, their values of up to 400 W m$^{-2}$ agree well with the range of LAP radiative forcing retrieved from the multi-transmittance approach (see Table 3)."*

Section 4.2.4. Here your results should be put in context with other modeling results (Picard et al. 2020)

We added acknowledgment of previous findings about the 'blue hook' in the discussion section by referencing the work from Picard et al. (2020):

*"As one of the major results of our study, we confirm conclusions from previous publications (e.g., Picard et al., (2020)) that remotely sensed snow HDRF can form an artificial downward*

*hook in the blue wavelengths, when retrieved over challenging mountainous terrain without considering the local topography, i.e., observation and illumination geometry."*

Section 5.1: I encourage the authors to briefly review other studies where the "blue hook" is visible (e.g. Naegeli et al. 2015; Di Mauro et al. 2017; Kokhanovsky et al. 2022).

Thanks for the encouragement. We added a brief review of other studies to Sect. 5.1:

*"The downward hook has been observed in various measurements from different types of instruments, including endmember spectra of ice surface materials derived from airborne APEX data (Naegeli et al., 2015), spaceborne Hyperion observations of bare ice (Di Mauro et al., 2017), or reflectance of Antarctic snow derived from the orbital PRISMA sensor (Kokhanovsky et al., 2022)."*

Line 412: I agree with this point. In fact, a bending in the blue band is displayed also in the imaginary part of the refractive index of ice. More discussion should be added on this point in the manuscript.

We agree that more discussion on this point would be valuable. However, we decided to move a detailed investigation of the blue hook to a subsequent manuscript, which has recently been submitted as a brief communication to The Cryosphere as well. We would like to point the referee to this resource, which can be found at
https://egusphere.copernicus.org/preprints/2024/egusphere-2024-1681/.

I think it's important to mention that often field spectroscopy data display a "upwarding" hook (e,g. Painter & Dozier 2004), that has been also modeled by Picard et al. 2020. This can be also found in imaging spectroscopy data for snow in particular slope/aspect conditions.

Thanks for mentioning this important point. In fact, we already describe this opposite behavior in lines 403-407: "Similarly, we can also observe an upward hook in cases where the local solar zenith angle is significantly larger than at top of atmosphere. Again, it is caused by an erroneous ratio of direct to diffuse irradiance, this time under the assumption of much more direct illumination than actually present, i.e., a red-shift in the solar irradiance. Opposite to sun-facing slopes, the blue reflectance is now much more overestimated than green or red reflectance since it has to compensate for larger radiance deviations in the blue spectral range when running the forward model." To include the referee's suggestion, we added the following sentences to the second paragraph of Sect. 5.1:

*"Previous simulation-based studies have also modeled the upward hook (e.g., Picard et al., 2020)."*

And:

*"This behavior has likewise been observed in field spectroscopy data, although being rather induced by saturation or linearity effects in the solar energy rich blue wavelengths (Painter & Dozier, 2004)."*

lines 428-429: this would be interesting.

While being very important, we believe that an investigation of different grain shape representations and snow and ice layer models would go beyond the scope of this manuscript. However, we would like to inform the referee that we are currently working on another study that investigates alternative approaches to model ice layers. Amongst others, it includes an approach to combine DISORT simulations with the model proposed by Whicker et al. (2022), which encloses air bubbles and a Fresnel layer between the ice and a thin snow cover.

line 430 and on: This is crucial because the OPs of dus are strongly dependent on its mineralogy and source area (Di Biagio et al. 2019). Using optical properties from Colorado is clearly a strong approximation here. I suggest to go in more detail regarding the possible differences in dust mineralogy between those two regions.

We fully agree with this comment and improved the discussion of dust optical properties in Sect. 5.2:

*"The use of dust OPs poses a different challenge, as they strongly depend on mineralogy and source area (Di Biagio et al., 2019). Several sets of dust OPs from different geographic regions, derived using diverse techniques and data, are publicly available. They have been obtained from any combination of field samples, spectral measurements, and linear mixing modeling, with Sahara, Colorado, Greenland, and Mars being the most prominent regional types (Polashenski et al., 2015; Skiles et al., 2017b; Balkanski et al., 2007, Singh et al., 2016). However, only a few studies have considered specific dust minerals when assessing their impact on snow melt (Lawrence et al., 2010; Kaspari et al., 2014; Reynolds et al., 2014). For our study, we selected only one type of dust OPs representing rather large particles, measured from samples that were collected in the San Juan Mountains of southwestern Colorado (Skiles et al., 2017b). In the lack of dust OP characterization in South America, we believe that the Colorado type is closest to the dust type found in Patagonia. This is especially supported by the finding that very large dust particles are often present in patchy snow of arid environments (Skiles et al., 2017b). Moreover, studies of the geochemical composition and mineralogy suggest that both the San Juan and the Patagonian dust are significantly dominated by quartz with 30-50 % of the total mineral mass (Lawrence et al., 2010; Demasy et al., 2024). Such analyses will be facilitated on even larger geographical scales by the EMIT mission objective, which is providing an improved understanding of the mineralogy of dust particle source regions, and enabling an enhanced identification and classification of dust OPs and their distribution around the Earth's snow-covered areas."*

line 450: I have the feeling that this threshold is quite low. I suggest to justify in detail this choice also showing frequency histogram of NDSI over the study area. Other possible classification methods can be applied to get snow cover from hyperspectral data (e.g. maximum likelihood, support vector machine etc.). Did the authors tested other methods?

No, we did not test other methods, but we agree that an NDSI threshold of 0.0 was indeed quite low. We updated the derivation of our snow mask by now following the procedure of Dozier (1989). We now use top-of-atmosphere (TOA) reflectance $\rho_{TOA}$ of EMIT bands at 485 nm (blue), 567 nm (green), and 1648 nm (SWIR), and marked a pixel as snow-covered when $\rho_{TOA,485} > 0.16$, $\rho_{TOA,1648} < 0.25$, and NDSI > 0.4. The latter is calculated using the bands at $\rho_{TOA,567}$ and $\rho_{TOA,1648}$. We added this information to the text:

*"It has to be noted that our approach is only applicable to snow- and ice-covered pixels. We identified these areas by following the procedure of Dozier (1989). We utilize TOA reflectance*

*rho_TOA of EMIT bands at 485 nm (blue), 567 nm (green), and 1648 nm (SWIR), and considered a pixel as snow-covered when rho_TOA,485 > 0.16, rho_TOA,1648 < 0.25, and NDSI > 0.4. The NDSI is the normalized-difference snow index calculated using rho_TOA,567 and rho_TOA,1648 (Dozier, 1989)."*

Also, we updated Fig. 6 accordingly and added the following:

*"We currently apply an NDSI threshold of 0.4 to determine snow-covered pixels."*

References:

Cordero, R. R., Sepúlveda, E., Feron, S., Wang, C., Damiani, A., Fernandoy, F., Neshyba, S., Rowe, P. M., Asencio, V., Carrasco, J., Alfonso, J. A., MacDonell, S., Seckmeyer, G., Carrera, J. M., Jorquera, J., Llanillo, P., Dana, J., Khan, A. L., and Casassa, G.: Black carbon in the Southern Andean snowpack, Environmental Research Letters, 17, 044 042, https://doi.org/10.1088/1748-9326/ac5df0, 2022.

Demasy, C., Boye, M., Lai, B., Burckel, P., Feng, Y., Losno, R., Borensztajn, S., and Besson, P.: Iron dissolution from Patagonian dust in the Southern Ocean: under present and future conditions, Frontiers in Marine Science, 11, https://doi.org/10.3389/fmars.2024.1363088, 2024.

Donahue, C. P., Menounos, B., Viner, N., Skiles, S. M., Beffort, S., Denouden, T., Arriola, S. G., White, R., and Heathfield, D.: Bridging the gap between airborne and spaceborne imaging spectroscopy for mountain glacier surface property retrievals, Remote Sensing of Environment, 299, https://doi.org/10.1016/j.rse.2023.113849, 2023.

Figueroa-Villanueva, L., Castro, L., Bolaño-Ortiz, T. R., Flores, R. P., Pacheco-Ferrada, D., and Cereceda-Balic, F.: Changes in Snow Sur- face Albedo and Radiative Forcing in the Chilean Central Andes Measured by In Situ and Remote Sensing Data, Water, 15, 3198, https://doi.org/10.3390/w15183198, 2023.

Kaspari, S., Painter, T. H., Gysel, M., Skiles, S. M., and Schwikowski, M.: Seasonal and elevational variations of black carbon and dust in snow and ice in the Solu-Khumbu, Nepal and estimated radiative forcings, Atmospheric Chemistry and Physics, 14, 8089–8103, https://doi.org/10.5194/acp-14-8089-2014, 2014.

Lawrence, C. R., Painter, T. H., Landry, C. C., and Neff, J. C.: Contemporary geochemical composition and flux of aeolian dust to the San Juan Mountains, Colorado, United States, Journal of Geophysical Research: Biogeosciences, 115, https://doi.org/10.1029/2009JG001077, 2010.

Louis, J., Devignot, O., and Pessiot, L.: S2 MPC - Level-2A Algorithm Theoretical Basis Document, Tech. Rep. S2-PDGS-MPC-ATBD-L2A - 2.10, European Space Agency (ESA), Keplerlaan 1, 2201 AZ Noordwijk, The Netherlands, 2021.

Reynolds, R. L., Goldstein, H. L., Bryant, B. M. M. A. C., Skiles, S. M., Kokaly, R. F., Flagg, C. B., Yauk, K., Berquó, T., Breit, G., Ketterer, M., Fernandez, D., Miller, M. E., and Painter, T. H.: Composition of dust deposited to snow cover in the Wasatch Range (Utah, USA):

Controls on radiative properties of snow cover and comparison to some dust-source sediments, Aeolian Research, 15, 73–90, https://doi.org/10.1016/j.aeolia.2013.08.001, 2014.

Santini, F. and Palombo, A.: Impact of Topographic Correction on PRISMA, Sentinel 2, and Landsat 8 Images, Remote Sensing, 14, 3903, https://doi.org/10.3390/rs14163903, 2022.

Yin, H., Tan, B., Frantz, D., and Radeloff, V. C.: Integrated topographic corrections improve forest mapping using Landsat imagery, Interna- tional Journal of Applied Earth Observations and Geoinformation, 108, https://doi.org/10.1016/j.jag.2022.102716, 2022.

---

## Author Comment (AC2)

**RC2**: Anonymous Referee #2

In this paper the authors - as of the title - want to investigate the pitfalls of topographic influence when analyzing snow signatures in mountaineous areas. However, the paper does not focus on this topic but rather gives an overview of how to perform a combined retrieval of snow parameters and atmospheric quantities and terrain influences. This work is of considerable importance and the paper's title should be changed accordingly. The pitfall of not considereing topography when analyzing hyperspectral data be it snow or other applications is well known and is of much less interest than the capability of retrieving the broad variety of paparameters from imagery in an optimization procedure simultaneously. However, for the latter the validation presented in the paper is not really sound and would need to be improved to make a convincing case about the accuracy of the such retrieved parameters. It is recommended to focus the paper on the parameter retrieval algorithm and describe the applied processing steps and the validation of the outputs more concisely.

We thank the referee for the feedback and the constructive review. Our general feeling of the significance of the work has always been in-between investigating the influence of topography and the combined retrieval of atmosphere and snow parameters. Therefore, we are in line with what the referee identified and changed the title of the manuscript to:

[revised manuscript text omitted]

We also changed the title of Sect. 4.2 to:

*"Sensitivity of snow physics"*

and slightly revised multiple phrases throughout Sect. 4.2 and the discussion section, mostly to move the focus from solely topography to physics in the forward model in general. Finally, we

substantially updated the conclusion to address the changes in the general narrative of the manuscript:

*"We introduce a new retrieval algorithm that estimates snow surface properties directly from at-sensor radiance measured by the spaceborne EMIT imaging spectrometer. We utilize a coupled full physics snow and atmosphere model and apply Optimal Estimation to solve for the most probable surface state. On one hand, this allows to reduce the number of retrieved state vector elements to only a handful of snow surface and atmosphere properties, including AOD, water vapor, snow grain size, liquid water content, LAP concentration, and local solar zenith angle. On the other, it facilitates a more thorough consideration of physical surface characteristics such as anisotropy and topography. We utilize two representative EMIT images acquired over the Argentine plain and the Chilean ice field in Patagonia, South America, to conduct a sensitivity analysis of our proposed forward model. We demonstrate that the retrieval of snow liquid water fraction and snow algae concentration is insensitive to topographic and directional effects. In contrast, estimations of snow grain size and mineral dust mass mixing ratio can be biased under these scenarios, which directly propagates into incorrect calculations of LAP radiative forcing. A validation with field measurements is still missing for the presented approach, but comparisons to Landsat 8 and Sentinel-2 reflectance values as well as to estimates of radiative forcing of previous studies indicate that surface reflectance as an intermediate quantity can be omitted in the retrieval framework in favor of inferring surface properties directly from radiance. Finally, we evidence that erroneous assumptions about surface topography are one of the -though not the only- major causes for the formation of the ``blue hook" in remotely sensed retrievals of snow reflectance. Future work must include a thorough validation effort, and needs to address mixed pixels and the modeling of both ice and LAP OPs to account for local geographical characteristics. Nevertheless, our findings are critical for updating melt runoff and climate model input, but also for the conception of retrieval algorithms for future orbital imaging spectroscopy missions, such as NASA's SBG. These missions provide the framework to develop and enhance processing schemes and retrieval algorithms on a global scale."*

Below, we address the additional line-by-line comments by providing respective responses and by indicating the changes to the manuscript.

Some detail comments:

- p3: l88: it is stated that the terrain may be rapidly shifting; this is indeed a problem for high spatial resolution imager - but at the resolution of EMIT such shifts are quite seldom and should not be a problem when using standard DSM products.

Thanks for this comment. That's certainly true. The topography is not rapidly shifting on a 60 m pixel resolution. We may have not fully correctly transferred the conclusion from Dozier et al. (2022). We rather wanted to express that reliance on auxiliary data products introduces an additional uncertainty component. We modified our statement to clarify our intention:

*"However, given the complex terrain of mountainous regions, and the current unavailability of coincident radar/lidar and imaging spectroscopy data in orbit, reliance on fixed DEMs may introduce additional retrieval errors, not only due to variability in local snow depth, but also because of uncertainties in the DEM product itself (Dozier et al., 2022)."*

*- p4, l97: it is claimed that a fully physics based model is employed when analyzing the data. On the other hand the optimal estimation is not based on physical parameter retrieval but rather on mathematical optimization what bears the risk of resulting in non-physical outputs at false minima. This limitation should be explained from the beginning.*

We agree that optimal estimation poses the risk of reporting local minima as the solution state. We now mention this limitation in more detail at the end of Sect. 3.2:

*"Our selected optimization scheme implies the risk of ending up in a local minimum, reporting a non-physical output. However, we use a traditional sequential atmospheric correction approach to initialize our inversion, which provides a first guess close to the probabilistic solution from OE. This promotes stability and fast convergence (Thompson et al., 2018)."*

*- p 4, l114: the term HDRF is used ambigously in this paper (and also in Literature). While Nicodemus defines HDRF as a physically well defined surface property with fully diffuse illumination and directional measurement, Schaepman-Strub 'redefined' the term as the real world hemispherical-directional situation with a anisotropic illumination field. That quantity would better be described as bottom-of-atmosphere directional reflectance rather than talking about 'HDRF". Please clarify in the paper how 'HDRF' is defined and uesd clearly. The same confusion is also geivn in line 120; integrating the 'Schaepman-Strup'-HDRF will not result in spectral albedo as long as the illumination field is not isotropic while integrating the Nicodemus-HDRF leads to a correct result.*

We agree that the use of the different reflectance terms is ambiguous and not consistent. Likewise, we understand that we compounded different terms and meanings ourselves. We actually follow the definition of HDRF as given by Schaepman-Strub et al. (2006), which is based on the nomenclature of Nicodemus et al. (1977) but incorporates the adaptations to the remote sensing case from Martonchik et al. (2000). We assume an anisotropic illumination field (direct + diffuse irradiance), including the dependency of the HDRF on atmospheric conditions and on the reflectance of the neighboring terrain. We revised the respective paragraph in Sect. 2.1 to clarify our usage of the term HDRF:

*"The HDRF is defined as the ratio of reflected spectral radiance $L_r$ at a particular solar and view geometry to the radiant flux $L_{id}$ that would be reflected from an ideal Lambertian surface, illuminated and observed under the same conditions (Schaepman-Strub et al., 2006):*
*...*
*The HDRF scenario is composed of hemispheric illumination, i.e., both direct and diffuse irradiance, but only direct reflection."*

*- p6, l141: again: the HDRF only depends on atmospheric conditions if the in-field bottom of atmosphere reflectance is confused with the real HDRF. A BRDF correction of the topographic effects would therefore be of high importance to analyze snow parameters in terrain.*

We hope that our answer to the previous comment clarified the usage of the term HDRF in our manuscript and that it follows the definition of Schaepman-Strub et al. (2006), including the dependency on atmospheric conditions and on the reflectance of the neighboring terrain. We also concur with the referee's comment that a BRDF correction would be the optimal way to get to accurate snow surface parameters. However, a BRDF correction is usually complex and less straightforward, and we believe that we address the consideration of topographic effects

to a large extent by using DISORT as our snow surface model and optimizing for the local solar zenith angle during the inversion.

- p8 eq (2): this equation does not include adjacency effects and terrain illumination on a pixel. In a snowy environment this assumption is a very rough approximation of the radiative interaction on a ground pixel.

Yes, that is correct. Our forward model does not include adjacency effects. While we agree that this factor is important, we also believe that it is less critical than modeling the ratio of direct to diffuse illumination correctly. We therefore decided to only include the latter in our study, as it is rather a proof of concept. However, subsequent work certainly needs to include additional physical effects of the terrain in the forward model. We added the following sentence to justify our choice:

*"We currently exclude the effects of adjacent pixels and slopes both to limit the complexity of our forward model and because their impact on modeled radiance is less critical than the separation of downward direct and diffuse transmittance (Guanter et al., 2009; Picard et al., 2020)."*

- p9 l204: the transferability of signatures between Greenland and Patagonia is a very rough assumptions. This should be corroborated by appropriate references or reasoning. The same applies to the transferability of dust signatures from Colorado to Patagonia.

That is a good point and certainly requires a thorough foundation. We implemented a detailed discussion about our choice of algae optical properties, and extended the justification for our selection of dust optical properties in Sect. 5.2:

*"To model biological LAPs, we utilize a set of algae OPs for the species Ancylonema (glacier algae) as well as Sanguina nivaloides and Chloromonas nivalis (snow algae), derived from samples collected on the Greenland Ice Sheet (Chevrollier et al., 2022). Despite being characterized at a different geographic location far away from our study site, we assume that these OPs adequately represent algae cells found on ice sheets, glaciers, and snow worldwide. This is corroborated by previous studies that identified those three species as being responsible for the darkening of snow and ice surfaces in various regions, including the Greenland Ice Sheet, Svalbard, the European Alps, and the Sierra Nevada in California (Yallop et al., 2012; Remias et al., 2012; Di Mauro et al., 2020a; Painter et al., 2001). Moreover, Takeuchi & Kohshima (2004) and Kohshima et al. (2007) identified Ancylonema and Chloromonas algae as among the most frequently encountered species on the Patagonian Ice Sheet.*

*The use of dust OPs poses a different challenge, as they strongly depend on mineralogy and source area (Di Biagio et al., 2019). Several sets of dust OPs from different geographic regions, derived using diverse techniques and data, are publicly available. They have been obtained from any combination of field samples, spectral measurements, and linear mixing modeling, with Sahara, Colorado, Greenland, and Mars being the most prominent regional types (Polashenski et al., 2015; Skiles et al., 2017b; Balkanski et al., 2007, Singh et al., 2016). However, only a few studies have considered specific dust minerals when assessing their impact on snow melt (Lawrence et al., 2010; Kaspari et al., 2014; Reynolds et al., 2014). For our study, we selected only one type of dust OPs representing rather large particles, measured from samples that were collected in the San Juan Mountains of southwestern Colorado (Skiles et al., 2017b). In the lack of dust OP characterization in South America, we believe that the*

*Colorado type is closest to the dust type found in Patagonia. This is especially supported by the finding that very large dust particles are often present in patchy snow of arid environments (Skiles et al., 2017b). Moreover, studies of the geochemical composition and mineralogy suggest that both the San Juan and the Patagonian dust are significantly dominated by quartz with 30-50 % of the total mineral mass (Lawrence et al., 2010; Demasy et al., 2024). Such analyses will be facilitated on even larger geographical scales by the EMIT mission objective, which is providing an improved understanding of the mineralogy of dust particle source regions, and enabling an enhanced identification and classification of dust OPs and their distribution around the Earth's snow-covered areas."*

We also added a sentence earlier in the manuscript that points the reader to the references and reasoning in the discussion section:

*"A thorough discussion about our choice of LAP OPs can be found in Sect. 5.2."*

- p10, fig5: just wondering: why are algae only influencing the visible part of the spectrum; what was the measurement database or could it be that the SWIR dat was simply not available?

Algae absorb solar radiation only in the visible part of the spectrum where the energy of the irradiance is highest, to maximize their photosynthesis (Chevrollier et al., 2022). Cell compounds such as carotenoid and chlorophyll lead to the characteristic absorption features (Painter et al., 2001). Moreover, absorption by the surrounding ice and snow is overly strong in the near- and shortwave-infrared wavelengths, so that almost no solar energy would remain for algae pigments. We added this clarification:

*"Note that algae absorb solar radiation only in the visible part of the spectrum where the energy of the irradiance is highest, to maximize their photosynthesis (Chevrollier et al., 2022). Moreover, absorption by the surrounding ice and snow is overly strong in the near- and shortwave-infrared wavelengths, so that almost no solar energy would remain for algae pigments."*

As mentioned in lines 202-204, we use the dataset from Chevrollier et al. (2022), which also includes the infrared part of the spectrum, to represent algal absorption in our DISORT simulations.

- p11 l226: it is stated that 'flat priors' are used in OE, however at the same time it is claimed that the method is fully physics based. How are physical boundary conditions enforced in the OE process then to avoid unphysical results?

We enforce the physical boundary conditions by utilizing a full-physics model to simulate snow reflectance during the inversion. Hence, the optimized snow and ice surface parameters are constrained by the physical shape of the reflectance as a function of grain size, liquid water content, and LAP concentration. Furthermore, and as mentioned in line 227, our problem is well-posed since we retrieve eight state vector parameters from 285 elements in the measurement vector. Under such conditions, the use of constrained priors in the OE setup can be obviated (Rodgers 2000). We added the following statement to Sect. 3.2:

*"We enforce physical boundary conditions by utilizing DISORT to simulate surface reflectance during the inversion. Hence, the optimized snow and ice parameters are constrained by the*

*physical shape of the reflectance as a function of grain size, liquid water content, and LAP concentration."*

- p11 eq. 4: how is the anisotropy factor c retrieved, a LUT is mentioned, what's in this LUT?

As mentioned in line 242, the LUT contains spectral anisotropy factors c for different geometries and grain sizes, where c is the ratio of spectral albedo to directional reflectance. We follow the approach of Painter et al. (2013b) and pre-calculated various c coefficients by utilizing modeled spectral albedo for different grain sizes as well as HDRF for different view and illumination geometries and varying grain sizes obtained from DISORT simulations (as stated in line 241). However, we revised to provide some clarification:

*"... where c is a function of observation and illumination geometry as well as snow grain size, and is calculated as the ratio of spectral albedo to HDRF. We follow the approach of Painter et al. (2013b) and use a LUT of pre-calculated c coefficients based on modeled spectral albedo for different grain sizes as well as HDRF for different view and illumination geometries and varying grain sizes obtained from DISORT simulations."*

- p16 l316: it does not seem obvious to me why liquid water and algae outputs should not be depending on terrain- please give some arguments.

Thanks for this comment. We realize that our phrasing in line 316 is misleading. We do not want to express that liquid water content and algae are independent of terrain. We rather wanted to make the point, that the difference in the retrieved values for those two quantities is uncorrelated with the difference in assumed local solar zenith angle. In other words, if the forward model does not consider topography and anisotropy, liquid water and algae estimates are not significantly biased. We revised our statement accordingly:

*"It is obvious that the difference in retrieved values of liquid water and algae is uncorrelated with the difference in assumed local solar zenith angle. Both scatter plots feature an $r^2$ of around 0.0 and almost no slope of the regression line. It seems that omitting topography and anisotropy in the forward model has no significant influence on the retrieval of those properties that have subtle absorption features and only marginally form the reflectance magnitude."*

- Figure 9: this is very small.. but the differences between L2A and OE is quite large; why?

Yes, we agree that the figure is quite small. We increased its size as much as possible. To clarify, both the L2A and the multi-transmittance spectra were retrieved using the same OE approach. The technical differences are 1. in the forward model, that considers topography in the multi-transmittance case; and 2. in the state vector, that comprises all 285 EMIT reflectance values in the L2A case, but only a handful of snow parameters in the multi-transmittance case. Since we show EMIT L2A spectra already in Sect. 4.2.1, we added the following phrases there:

*"We also show corresponding spectra from the EMIT L2A product. They were retrieved by applying the same OE technique, but without considering topography in the forward model, and by obtaining HDRF from statistical modeling using constrained priors instead of utilizing a snow surface radiative transfer model."*

The potential reasons for the difference between the EMIT L2A spectra and the results from the multi-transmittance approach are given in Sect. 4.2.1. Please see lines 288-307 in the initial version of the manuscript.

- P18 l356: 'a good agreement' of incidence angles is reported, how 'good' is it indeed, how large where the samples, and how about the statics on a per-pixel basis?

We agree that 'good agreement' is a very ambitious statement here. To substantiate our claim, we added a few statistics about the per-pixel comparison:

*"We observe a good agreement in our six examples between $\theta_{iest}$ and $\theta_{ical}$ with only marginal deviations of up to 4°. This is confirmed by looking at the regression analysis of all 995,372 snow covered pixels in the image, which shows an $R^2$ of 0.64 and an RMSE of 3.58° between $\theta_{iest}$ and $\theta_{ical}$."*

- Table3: differences in RF are quite large and one does not know the real value. So, how could you absolutely validate the results and why are you sure that the Multi-transmittance output is more reliable?

That is correct, we do not know the true value and cannot do a comprehensive validation. However, since LAP radiative forcing is obtained from the spectra shown in Fig. 9, we assume that the multi-transmittance output provides more reliable input to the RF calculation, simply because we see more reasonable reflectance shapes and magnitudes given the topographic characteristics of each of the six examples. We modified our wording to highlight that we only make assumptions here:

*"The result for S3 highlights that LAP radiative forcing is more than 400 W m$^{-2}$ higher on sun-facing slopes if snow surface physics are neglected in the forward model. Not accounting for $\theta_i < \theta_0$ causes a steeper slope in the estimated blue reflectance, which resembles LAP absorption. On pixels with $\theta_i > \theta_0$, e.g., locations S4 and I2, radiative forcing estimated from EMIT L2A spectra is generally smaller. The assumed underestimation ranges between 140 and 207 W m$^{-2}$ in our examples."*

We also added a new section to the discussion dealing with the missing field validation. In particular, we focus on the evaluation of estimated RF from the multi-transmittance approach:

*""One essential part of remote sensing retrievals is still missing for the presented multi-transmittance approach, which is the validation with field measurements. This can be explained by the remoteness of our study sites in South America, but also by the fact that this work has rather been designed as a theoretical proof of concept. A good indication for the accuracy of our proposed inversion method is already given by the comparison to Landsat 8 and Sentinel-2 data in Sect. 4.2.3. However, to further compensate for the lack of validation and to put our results into the right context, we provide a comparison to findings from previous studies.*

*We look at the estimated error in radiative forcing when topography and anisotropy are not considered in the forward model. Previous work in the Chilean and Argentinian Andes reports daily or annual averages of LAP radiative forcing, or even the mean over a period of multiple years (Rowe et al., 2019; Cordero et al., 2022; Figueroa-Villanueva et al., 2023). Their values range between 0 and 10 W m$^{-2}$, mainly investigating the influence of black carbon, which was*

*not the focus of our study. In contrast, our method provides the instantaneous radiative forcing due to LAP, which allows a reasonable comparison to similar work conducted in the Sierra Nevada, CA, or the Rocky Mountains, CO (Painter et al., 2013a; Seidel et al., 2016). Even though estimated in a different geographical location, their values of up to 400 W $m^{-2}$ agree well with the range of LAP radiative forcing retrieved from the multi-transmittance approach (see Table 3).”*

- p22: Conclusion: it is again stated that a 'full physics' approach was used, maybe I misunderstand the paper but as far as I can see this is not an inversion of full physics model but rather a statistical optimization with flat priors.

We agree that our wording might be misleading here. We invert a coupled full physics snow and atmosphere model that provides snow HDRF and atmospheric absorption and scattering properties by utilizing Optimal Estimation as inversion technique. So yes, the entire approach is not 'fully physics-based', but the inverted model is. We try to clarify this by revising:

[revised manuscript text omitted]

---

## Author Comment (AC3)

**RC3**: Quentin Libois

Review of « The pitfalls of ignoring topography in snow retrievals: a case study with EMIT», by Niklas Bohn et al. Review by Quentin Libois.

**General comments**

This paper presents a novel retrieval algorithm based on optimal estimation to retrieve snow properties (grain size, dust and algae contents, liquid water content) from hyperspectral satellite images in the solar spectrum. The main originality of the algorithm is to account for local topography by including the slope angle in the retrieved parameters. The algorithm is applied to two images acquired above Patagonia by the spaceborne instrument EMIT. Accounting for the local slope has a significant impact of the dust and grain size retrievals, but is less critical for algae and liquid water content which are characterized by localized spectral features. These retrievals are also used to compute the instantaneous radiative forcing of light absorbing particles, which is also very sensitive to slope effects. The spatial gradients of the retrieved quantities are discussed on a physical basis, supporting the reliability of the retrievals. A preliminary sensitivity study is performed, which highlights how the aerosol optical depth in the atmosphere and the slope angle impact the snow reflectance. The apparent drop in the blue range of snow spectra is discussed in details to highlight that topography is largely responsible for this common feature of snow reflectance as measured from space. Some limitations of the current algorithm and suggestions for improvements are also provided.

The paper is overall well written, although the abstract and introduction could probably be greatly improved, in particular to better remind the existing strategies already used to account for topography in snow retrievals from space and the link between the present work and previous studies from the same authors. More technical information about EMIT would be appreciated and the relevance of the retrieval algorithm to other spaceborne instruments could be elaborated. Some technical details lack in the presentation of the algorithm but could easily be provided. My main concern is about the representation of snow by a collection of spheres to simulate the directional reflectance. This issue is only very briefly mentioned in the discussion, without any quantification of the potential impacts. It certainly deserves much more attention. Also, the accuracy of the retrievals is not discussed, while the theoretical framework used would make it easy to investigate, and would strengthen the conclusion that retrievals of LAP are possible with EMIT while they were apparently challenging with previous spaceborne instruments. For these reasons I recommend that these points be carefully treated before the paper can be considered for publication.

We thank Dr. Libois for the positive feedback and the very constructive review. We significantly revised the manuscript by

1. Shortening and improving the abstract.
2. Clarifying objectives and novelty of the study as well as providing more context to existing literature in the introduction.
3. Providing more technical information about the EMIT instrument.
4. Enhancing the description of our retrieval algorithm.
5. Expanding the discussion about the modeling of snow grain shape and ice optical properties.
6. Adding posterior uncertainties and error correlation coefficients for the retrieved state vector parameters, obtained from the optimal estimation framework

Below, we address both specific and line-by-line comments by providing respective responses and by indicating the changes to the manuscript.

**Specific comments**

1) The abstract is probably too long. It could more efficiently start with the relevance of monitoring LAP and grain size. Also the objective is not clearly stated, neither the main novelty compared to previous work. In general it is not very clear and would deserve some general rewriting (see some suggestions in the technical comments).

Thanks for this comment! We shortened the abstract and provide a more efficient description of the relevance of monitoring LAP and grain size at the beginning:

*"Global patterns of snow darkening and melting, induced by grain metamorphism and the accumulation of small light-absorbing particles (LAPs), such as mineral dust, black carbon, volcanic ash, or algae cells, lead to an intensified radiative forcing and retreat of Earth's snow cover. Mapping and quantifying snow grain size and LAPs on both temporal and spatial scales is needed to improve the prediction of melt rates and their impacts on climate change."*

We also state both the objective and the novelty of our study more clearly now:

*"Accurate retrievals of snow surface properties, including grain size, liquid water content, as well as concentration of mineral dust and algae, require a precise, ideally joint accounting for atmospheric, topographic, and anisotropic effects in the reflected radiance. However, previous methods either neglect physical effects of the surface or utilize the surface reflectance as an intermediate non-physical quantity, in part without proper error propagation from the atmospheric modeling and obtained from statistical modeling. In this contribution, we present a novel surface-atmosphere radiative transfer model that couples the MODTRAN code with a physics-based snow surface reflectance model that utilizes the multistream DISORT program. Our model allows to omit the intermediate retrieval of surface reflectance, and to estimate snow surface and atmosphere properties directly from measured radiance."*

2) The introduction as well could notably be improved. What is mainly missing is information about algorithms already used for LAP retrieval, and to account for topography (e.g. Picard et al., 2020). EMIT is selected because it supposedly has a larger SNR so what would happen if existing algorithms were applied to EMIT? Why such a motivation to build a new retrieval algorithm? This should be better motivated. Also, how does this study complement recent previous work from the same authors?

Algorithms used for LAP retrieval are already mentioned in the introduction (Painter et al., 2013a; Seidel et al., 2016; Bohn et al., 2021, 2022), but we added references to already existing snow algorithms that account for topography (Picard et al., 2020; Donahue et al., 2023):

*"Specifically, Picard et al. (2020) demonstrated the sensitivity of snow albedo measurements to surface slope based on spectral data taken in the field, and proposed a correction approach to retrieve the intrinsic albedo. Using a digital elevation model (DEM), this local geometry, including surface slope and aspect, can be calculated and incorporated in atmospheric modeling schemes in order to correct for spectral distortions in the retrieved surface reflectance (Richter and Schläpfer, 2017). However, given the complex terrain of mountainous*

*regions, and the current unavailability of coincident radar/lidar and imaging spectroscopy data in orbit, reliance on fixed DEMs may introduce additional retrieval errors, not only due to variability in local snow depth, but also because of uncertainties in the DEM product itself (Dozier et al., 2022). For instance, Donahue et al. (2023) showed that topographic correction with coarse and non-coincident DEMs introduces significant errors in estimated snow albedo from air- or spaceborne imaging spectroscopy of up to 20 %. Overall, a mature and comprehensive modeling of topography for spaceborne imaging spectroscopy data over mountain snow has not yet been demonstrated, and only a few studies have applied a limited post-hoc correction at the airborne scale (Painter et al., 2013a; Seidel et al., 2016)."*

The motivation to build a new retrieval algorithm is not specifically driven by the selection of the EMIT instrument, although it indeed provides a very convincing SNR and data quality in general (see Thompson et al. (2024)). Our study is rather motivated by the need to include more comprehensive physics in the forward model and to circumvent the retrieval of surface reflectance as a non-physical, intermediate quantity. In fact, our proposed algorithm could be applied to any available spaceborne imaging spectrometer, e.g., to EnMAP and PRISMA as well. We tried to provide a better motivation and a more reasonable connection to our previous work:

*"Recent work has demonstrated that a simultaneous inversion of atmosphere and surface state using optimal estimation (OE) shows promising potential to quantify even low concentrations of LAPs on a global scale from spaceborne imaging spectroscopy observations (Bohn et al., 2021, 2022). However, the approach utilizes the surface reflectance as an intermediate non-physical retrieval quantity assuming Lambertian behavior. It is obtained from statistical modeling using constrained priors, impeding a proper consideration of surface topography and anisotropy. This could lead to significant biases in downstream estimates of LAP concentration, and propagate to erroneous calculations of LAP radiative forcing as these physical effects influence both magnitude and shape of measured spectral radiance as a function of local view and solar geometry (Carmon et al., 2022, 2023). [...] To improve the downstream estimation of biogeophysical quantities, we need to align the surface and atmospheric forward modeling assumptions. In particular, the retrieval of properties on highly anisotropic surfaces such as snow and ice will benefit from capturing local topographic conditions through physical modeling as directional effects are minimized. We present an updated version of the algorithm that was originally introduced by Thompson et al. (2018) and modified by Bohn et al. (2021). It simultaneously retrieves atmosphere and surface properties from imaging spectrometer measurements by inverting a wavelength-dependent top-of-atmosphere (TOA) radiance model. In this work, we introduce a full physics-based characterization of atmosphere and surface by coupling the MODTRAN atmosphere radiative transfer code with the multistream DISORT program. The latter is utilized to simulate directional snow reflectance as a function of biogeophysical properties as well as view and illumination geometry. This facilitates the consideration of local surface anisotropy and topography in the forward model and removes dependency from external DEMs (Carmon et al., 2023). Aim is to utilize this best in class physical and atmospheric modeling simultaneously to present estimations of snow surface properties directly from measured radiance."*

3) EMIT is central in the present study. However it is nowhere described in details. In particular information on the spatial resolution (which is very critical) is lacking. Likewise, information about its spectral resolution and radiometric accuracy would be very useful. An important question being: why using EMIT and not any other spaceborne hyperspectral (or multi-spectral if it would be enough for the purpose of the study) sensor.

Good point! A more detailed description of the EMIT instrument was indeed missing, so we revised:

*"EMIT is a high performance VNIR-SWIR imaging spectrometer whose prime mission focus is to deliver maps of surface mineralogy and relative abundance of different mineral types from arid dust source regions. These maps will provide improved input to Earth System Models of atmospheric transport and radiative forcing by constraining the composition of regional dust emissions (Connelly et al., 2021). Extending over a wavelength range of 380-2500 nm with a spectral resolution of approximately 7.5 nm, EMIT images provide a pixel size of around 60 m on a 74 km wide swath. The temporal revisit time is variable depending on the orbital cycle of the ISS, and ranges between one day and more than a week (Thompson et al., 2024). After more than a year in operation, EMIT provides data products from many different regions of the Earth, including snow covered high mountains, and experiments have confirmed a remarkably high SNR of more than 500 on average and above 750 in the VNIR wavelengths (Thompson et al., 2024)."*

The selection of EMIT for our study was primarily driven by its convincing performance with respect to SNR and overall data quality, but also simply because it was built at JPL and thus, represents an 'in-house' instrument. Again, our proposed algorithm could be applied to any other available spaceborne imaging spectrometer, for instance, to EnMAP or PRISMA.

4) The authors represent snow as a collection of spheres, although it has been known for a long time that this is not appropriate, in particular to describe the anisotropy of snow reflectance. The quantitative impact of such an assumption on the retrievals is not investigated, which is detrimental to the overall quality and impact of the study. I'd encourage the authors to test the retrievals with other datasets of snow directional reflectance (from observations or models depending on the availability of the data, see some suggestions in the technical comments). Also, adding a figure to illustrate the snow HDRF used in the forward model would probably help the interpretation of the spectra and their sensitivity to illumination and viewing geometry

We understand and agree that assuming spherical particles for snow might be inappropriate. We follow the approach of Grenfell and Warren (1999) and model non-spherical snow particles by a collection of independent spheres that has the same volume-to-surface-area ratio as the non-spherical particles. Several studies have shown that this approach provides an accurate representation of extinction efficiency and single-scattering albedo, while only the scattering asymmetry factor is usually overestimated (Grenfell and Warren, 1999; Neshyba et al., 2003; Grenfell et al., 2005; Warren, 2019). Its effect on bulk optical properties can be compensated though by reducing the grain size of the model (Dang et al., 2016). Nevertheless, more realistic snow directional reflectance datasets certainly exist, but we would like to defer the testing of various representations of snow and ice optical properties in our retrieval framework to a subsequent study. We consider the current manuscript rather as a concept study for the simultaneous retrieval of atmosphere and physical surface properties directly from at-sensor radiance. In fact, we changed and adjusted both title and narrative of the manuscript, following the suggestions of reviewer #2. In any case, we extended the discussion in Section 5.2:

*"We apply Mie scattering theory to obtain the ice OPs, and follow the approach of Grenfell and Warren (1999) to model non-spherical snow particles by a collection of independent spheres that has the same volume-to-surface-area ratio as the non-spherical particles. This assumption might be inappropriate, but studies have shown that this approach provides an*

*accurate representation of extinction efficiency and single-scattering albedo, while only the scattering asymmetry factor is usually overestimated (Grenfell and Warren, 1999; Neshyba et al., 2003; Grenfell et al., 2005; Warren, 2019). Its effect on bulk optical properties can be compensated though by reducing the grain size of the model (Dang et al., 2016). Nevertheless, more accurate representations of the optical shape of snow as well as more realistic snow BRDF datasets certainly exist (e.g., see Malinka (2014), Dumont et al. (2021), Malinka (2023)), but we would like to defer the testing of various representations of snow and ice optical properties in our retrieval framework to a subsequent study. We consider the current manuscript rather as a concept study for the simultaneous retrieval of atmosphere and physical surface properties directly from at-sensor radiance. However, we have to be aware of potential misrepresentations in many cases. For instance, the glacier outflows from the Chilean ice field are likely not well represented by applying Mie theory. The grains on bare ice surfaces typically appear to be arbitrarily shaped with irregular dimensions, so that, e.g., a Geometric Optics approach based on ray-tracing would be more appropriate to model their OPs (Kokhanovsky and Zege, 2004; Cook et al., 2020; Bohn et al., 2022). In future work, we need to compare retrieval results from assuming different grain shapes, including spheres, spheroids, hexagonal plates, and Koch snowflakes (He et al., 2017; Hao et al., 2023). We will also apply different approaches to model ice layers, e.g., with enclosed air bubbles and a Fresnel layer between the ice and a thin snow cover (Whicker et al., 2022)."*

We actually provide a figure of snow HDRF used in the forward model in the initial submission of our manuscript. Figure 1 highlights the sensitivity of snow HDRF to variations in solar zenith angles for different grain sizes. In addition, Figure 2 depicts changes in snow HDRF with varying ratios of direct to diffuse illumination.

5) I feel like some technical details in the algorithm are missing. First, the equation for the forward model would deserve more physical explanations, beyond a reference to a paper. Then, snow grain size is not properly defined. Is it a radius, a diameter, averaged or effective over a prescribed size distribution, or on the contrary a monodisperse collection? As a consequence, the way liquid water content is accounted for is not sufficiently clear. Likewise it is not clear if the treatment of LAP relies on mass absorption coefficients or any other optical quantity. The instrumental noise is not detailed either while some quantitative information to highlight the high SNR would be appreciated.

Thanks for this comment! We added a comprehensive explanation of the physical quantities in our forward model:

*"$I_0$ is the atmospheric path radiance, i.e., the number of solar photons that are scattered by the atmosphere into the line-of-sight of the sensor without interaction with the surface. $t^{\downarrow}_{dir}$, $t^{\downarrow}_{dif}$, and $t^{\uparrow}$ are direct downwelling, diffuse downwelling, and total upwelling transmittance of the atmosphere. Scaled by two different angles, $t^{\downarrow}_{dir}$ and $t^{\downarrow}_{dif}$ represent the partition into direct and diffuse irradiance at the surface. Finally, $s$ is the spherical albedo of the atmosphere, which describes the multiple scattering of photons between the target pixel and the surrounding atmosphere before they enter the line-of-sight of the sensor."*

Next, we revised the definition of snow grain size and liquid water content:

*"We assume the snow grains to be shaped as a collection of spheres characterized by a specified radius that equals three times the volume-to-area ratio of the real non-spherical snowpack (Warren, 2019), and apply traditional Mie Theory to obtain their OPs (Grenfell and*

*Warren, 1999). [...] We add the influence of liquid water by modeling OPs of coated spheres for the case of wet snow by adding the width of a circular layer of liquid water to the grain radius (Green et al., 2002)."*

We use single-scattering albedo, mass extinction coefficient, and asymmetry parameter as optical properties for LAPs. Essentially, they are the same as for the snow particles and we model the LAP-contaminated snow as a linear mixture of all OPs, as we've mentioned in lines 197-198 and 208 of the initial manuscript submission.

Finally, we added specific numbers for the high SNR of EMIT measurements:

*"After more than a year in operation, EMIT provides data products from many different regions of the Earth, including snow covered high mountains, and experiments have confirmed a remarkably high SNR of more than 500 on average and above 750 in the VNIR wavelengths (Thompson et al., 2024)."*

6) One originality of the retrieval algorithm is to use the optimal estimation in combination with a forward model to retrieve the parameters of a state vector (instead of retrieving for instance reflectances). However only the most probable solutions of the problems are presented, without any reference to the associated uncertainties. Given that the objective of the study is to demonstrate that EMIT can be used to retrieve snow properties that are not accessible with other instruments, mentioning uncertainties is key to convince the reader that the retrieved quantities are reliable, in particular given that there is no ground truth. For instance it is quite questioning that the retrieved dust quantities can be strictly zero (Table 2). Beyond the uncertainties I'd encourage the authors to further investigate the correlations between the retrieved parameters, which could help understand how compensation between variables can affect the quality of the retrievals.

That's a very good point! We absolutely agree that providing posterior uncertainties is one of the main advantages of optimal estimation. Our initial submission was more focused on the dependency of the retrieved parameters on topographic and anisotropic effects of the surface than on error estimates given by our OE inversion setup. However, we followed Dr. Libois' suggestion and added the posterior uncertainties, i.e., the square-root of the variance, to each retrieved parameter value in Table 2, and briefly discuss the numbers in a new paragraph:

*"Table 2 also shows posterior uncertainties for each retrieved parameter in standard deviation as reported by the OE framework. Errors are generally low, in particular for quantities exhibiting distinct absorption features in the reflectance spectrum, such as water vapor, grain size, or algae. Higher uncertainties are reported especially for AOD, but also for retrieved dust concentration. It has to be noted though that the linearized posterior error predictions by OE are likely optimistic. OE presumes a linearized version of the forward model, with a local multivariate Gaussian error prediction. This ignores other local minima solutions, if they exist, and even the local estimate may under predict errors (Hobbs et al., 2017; Cressie, 2018). However, the uncertainty estimates provide at least a useful hint whether the variance of retrieved values is reasonable or not."*

Likewise, we consent to Dr. Libois' comment that an investigation of the correlations between retrieved parameters is crucial. We calculated the error covariance matrix for the retrieved parameters from the image of the Chilean ice field following Govaerts et al. (2010), and added this result to Section 4.2.2 as Figure 9, including a new paragraph:

*"Another way of assessing the sensitivity of retrieved parameters is to look at their error correlation. OE provides a measure of retrieval uncertainty for each state vector element in terms of a posterior covariance matrix, which can be normalized to an error correlation matrix (Rodgers, 2000; Govaerts et al., 2010). We calculated this matrix from the retrieval maps for the Chilean ice field for both atmosphere and surface properties (Fig. 9). The coefficients confirm the findings from Fig. 8 by showing a slight negative correlation between $\theta_i$ and grain size, a slight positive correlation between $\theta_i$ and dust, and no correlations between the assumed incident angle and both liquid water and algae. We observe expected anticorrelation between pressure elevation and both AOD and water vapor, and identify an overall disentanglement between surface and atmosphere parameters. Only exceptions are $\theta_i$, being an input to both atmosphere and surface model, and AOD, which features a negative correlation with dust LAPs. $\theta_i$, or more precisely, the assumed ratio of direct to diffuse irradiance, dust, and atmospheric aerosols cause a very similar shape of reflectance in the visible wavelengths, leading to potential ambiguities between them."*

In addition, we added some of these findings to Section 4.2.4 about the blue wavelengths:

*"Our selection of AOD for the comparison is supported by the findings from Fig. 9, which show a negative correlation between AOD and both $\theta_i$ and dust. In other words, the retrieval could bare the risk of compensating for a biased AOD by altering the other two parameters and vice versa."*

And:

*"In contrast, variations in AOD can not compensate for erroneously assumed $\theta_i$, which supports the conclusion that the anticorrelation between the two properties is rather one-sided, and a correct assumption or retrieval of the incident angle is more important than a proper characterization of atmospheric aerosols."*

**Technical comments**

l.2: solar radiation would be better than illumination

To shorten our abstract, we removed the respective sentence.

l.3: at negative temperature melting is not critical, but metamorphism is

To shorten our abstract, we removed the respective sentence, but added grain metamorphism as cause for snow melting:

*"Global patterns of snow darkening and melting, induced by grain metamorphism and the accumulation of small light-absorbing particles (LAPs), such as mineral dust, black carbon, volcanic ash, or algae cells, lead to an intensified radiative forcing and retreat of Earth's snow cover."*

l.6: maybe specify LAPs on/in snow?

Yes, we added a list of more specific LAPs:

*Global patterns of snow darkening and melting, induced by grain metamorphism and the accumulation of small light-absorbing particles (LAPs), such as mineral dust, black carbon, volcanic ash, or algae cells, lead to an intensified radiative forcing and retreat of Earth's snow cover."*

l.11: not clear what dust properties EMIT measures

EMIT does not measure dust properties directly, but solar radiation reflected from Earth's surface. By removing atmospheric effects, we obtain characteristic surface reflectance, which is then used as input to a retrieval of mineralogy and relative abundance for different mineral types. We revised by removing the information about the prime mission focus as we believe that it is not relevant in the abstract and clarified which type of measurement imaging spectrometers such as EMIT provide:

*"This technology provides measurements of reflected solar radiation in continuous spectral channels throughout the solar spectrum, allowing to detect narrow LAP absorption bands."*

l.12: what is a "target mask"?

EMIT's target mask indicates the coverage of observations. EMIT is not collecting data for every location on its orbit, but only if it is within the predefined target mask. We removed this term in order to avoid any confusion:

*"EMIT observations include snow cover in low to mid-latitude mountainous regions, such as the Western US, the Andes in South America, or high-mountain Asia."*

l.15: anisotropy of what?

We mean the anisotropy of the snow surface and clarified the phrase:

*"Accurate retrievals of snow surface properties, including grain size, liquid water content, as well as concentration of mineral dust and algae, require a precise, ideally joint accounting for atmospheric, topographic, and anisotropic effects in the reflected radiance."*

l.16: why "forward scattering"? Not clear

We removed this term for clarity.

l.18-20: quite difficult to understand in an abstract

We agree and revised our statement for more clarity:

*"In this contribution, we present a novel surface-atmosphere radiative transfer model that couples the MODTRAN code with a physics-based snow surface reflectance model that utilizes the multistream DISORT program. Our model allows to omit the intermediate retrieval of surface reflectance, and to estimate snow surface and atmosphere properties directly from measured radiance."*

l.21: it would have been helpful to detail earlier (e.g. l.15) what are the snow properties to be retrieved

Fully agreed! We added this detail:

*"Accurate retrievals of snow surface properties, including grain size, liquid water content, as well as concentration of mineral dust and algae, require a precise, ideally joint accounting for atmospheric, topographic, and anisotropic effects in the reflected radiance."*

l.22: use µg g-1 instead, as well as for all units

Revised throughout the manuscript!

l.23: such a forcing seems huge! It's because it's instantaneous.

Yes, that's right. We added the word "instantaneous":

*"Furthermore, we demonstrate differences in instantaneous LAP radiative forcing of up to 400 Wm$^{-2}$ in cases of LAP concentration inaccurately quantified from surface reflectance"*

l.25: is the "blue hook" something sufficiently well known (it is not to me) to appear as is in an abstract?

That's a fair question. We removed the sentence, also to support shortening the abstract.

l.25-26: the link with runoff and climate models is definitely not obvious. Either to be removed, or expanded

Agreed. We decided to remove this sentence as well.

l.30: I don't see why having the highest albedo of all natural surfaces is a reason for playing a key role...

We restructured the sentence to clarify our statement:

*"Snow surfaces play a key role in Earth's radiation budget as their high albedo reflects most of the incoming solar radiation, steering important feedback mechanisms in climate change (Lemke et al., 2007)."*

l.31: "cooling effect" is a bit surprising to read. Snow does not cool the Earth, or at least it depends with respect to what? Ok if you say "more snow-covered surfaces will tend to cool the Earth"

That's another fair point. Lemke et al. (2007) actually state that because of the high albedo, changes in snow and ice cover are important feedback mechanisms in climate change, mainly manifesting in a strong correlation between snow cover and air temperature. We removed the term "cooling effect":

*"Snow surfaces play a key role in Earth's radiation budget as their high albedo reflects most of the incoming solar radiation, steering important feedback mechanisms in climate change (Lemke et al., 2007). Changes in global snow cover are very sensitive to small variations in both air temperatures and the amount of absorbed solar radiation (Di Mauro et al., 2015)."*

l.37: LAPs are not the only reason for snow darkening (or at least albedo decrease). Metamorphism has a similar effect

That's right. We revised the sentence accordingly:

*"One of the main drivers of the decrease in snow cover and albedo is the presence of small light-absorbing particles (LAPs) on snow and ice surfaces (Di Mauro, 2020)."*

l.39: Sun's energy is unclear → where the solar spectrum peaks? Where the sun irradiance is maximum?

Revised for clarification:

*"These particles are mainly absorptive in the visible part of the solar spectrum where the Sun's irradiance is highest, and when present lead to a considerable amount of additionally absorbed radiation."*

l.59: not clear what "not tied to physical units" means

Agreed, this is not very clear. We wanted to express that these algorithms do not report physical units in terms of quantifying LAP concentration. However, we decided to remove this expression anyhow.

l.61, 62: what are EnMAP, PRISMA?

EnMAP and PRISMA are European spaceborne imaging spectroscopy missions, being in operation for a few years now. We added a few more details:

*"Measurements from the recently launched orbital imaging spectroscopy missions EnMAP and PRISMA have been utilized to conduct preliminary sensitivity analyses and first attempts to estimate dust concentration on snow and ice (Bohn et al., 2021, 2022; Kokhanovsky et al., 2022). These studies concluded though that low amounts of inorganic LAP deposition cannot be detected with remote sensing measurements, which is in line with findings from other studies (e.g., Warren (2013))."*

l.68: not clear what these references correspond to

They correspond to the two mentioned missions SBG and CHIME. We moved the references for clarity:

*"Future orbital imaging spectroscopy missions, such as NASA's Surface Biology and Geology (SBG) (National Academies of Sciences, Engineering, and Medicine, 2018) and ESA's Copernicus Hyperspectral Imaging Mission for the Environment (CHIME) (Rast et al., 2019) will address this problem by providing high signal-to-noise ratios (SNR) of more than 400 in the visible-to-near-infrared (VNIR) and more than 250 in the shortwave-infrared (SWIR), as well as high spectral and spatial resolution."*

l.69: absorption spectral features?

We incorporated this suggestion:

*"This will enable the detection and quantification of LAPs by resolving their subtle spectral absorption features even for low concentrations."*

l.73: not clear why mapping arid surfaces informs about transport and radiative forcing. EMIT should be more clearly introduced

Thanks for the comment. We agree that a proper introduction of EMIT was missing and added the following phrases:

*"EMIT is a high performance VNIR-SWIR imaging spectrometer whose prime mission focus is to deliver maps of surface mineralogy and relative abundance of different mineral types from arid dust source regions. These maps will provide improved input to Earth System Models of atmospheric transport and radiative forcing by constraining the composition of regional dust emissions (Connelly et al., 2021)."*

l.83: the transition from the Lambertian assumption issue to the topography issue is too fast. Is there a link between both?

Topography causes the surface to behave non-Lambertian as a function of slope and aspect. This adds to the intrinsic directional effects of snow reflectance. We revised this paragraph and provide a more reasonable connection between anisotropy and topography:

*"However, the approach utilizes the surface reflectance as an intermediate non-physical retrieval quantity assuming Lambertian behavior. It is obtained from statistical modeling using constrained priors, impeding a proper consideration of surface topography and anisotropy. This could lead to significant biases in downstream estimates of LAP concentration, and propagate to erroneous calculations of LAP radiative forcing as these physical effects influence both magnitude and shape of measured spectral radiance as a function of local view and solar geometry (Carmon et al., 2023). Specifically, Picard et al. (2020) demonstrated the sensitivity of snow albedo measurements to surface slope based on spectral data taken in the field, and proposed a correction approach to retrieve the intrinsic albedo."*

l.89: what does "rapidly shifting terrain" mean?

We revised this sentence and removed the term "rapidly shifting terrain":

*"However, given the complex terrain of mountainous regions, and the current unavailability of coincident radar/lidar and imaging spectroscopy data in orbit, reliance on fixed DEMs may introduce additional retrieval errors, not only due to variability in local snow depth, but also because of uncertainties in the DEM product itself (Dozier et al., 2022)."*

l.91: at first order and satellite footprint scale the mountain topography probably dominates snow depth variability, nope?

We agree in general that on the satellite footprint scale (> 30 m pixel size) the local topography is the dominant effect rather than variability in snow depth. However, we do not fully understand the intention of this comment and its connection to the respective line in the manuscript. We mean that 'reliance on fixed DEMs may introduce additional retrieval errors' not only due to variability in local snow depth, but also because of uncertainties in the DEM product itself. We added this to the text to provide some clarification:

*"However, given the complex terrain of mountainous regions, and the current unavailability of coincident radar/lidar and imaging spectroscopy data in orbit, reliance on fixed DEMs may introduce additional retrieval errors, not only due to variability in local snow depth, but also because of uncertainties in the DEM product itself (Dozier et al., 2022)."*

l.95: could you explain what does this algorithm

Sure. We added the following:

*"It simultaneously retrieves atmosphere and surface properties from imaging spectrometer measurements by inverting a wavelength-dependent top-of-atmosphere (TOA) radiance model."*

l.96-97: not clear what is the atmospheric radiative code and the snow one. Also it suggests that 3D effects (reillumination by neighboring slopes) are not accounted for by such a model. Do you confirm?

We use MODTRAN to simulate atmosphere radiative transfer (it actually uses DISORT internally for the calculation of multiple scattering), and DISORT itself for modeling directional snow reflectance. To clarify, we revised:

*"In this work, we introduce a full physics-based characterization of atmosphere and surface by coupling the MODTRAN atmosphere radiative transfer code with the multistream DISORT program. The latter is utilized to simulate directional snow reflectance as a function of biogeophysical properties as well as view and illumination geometry."*

We confirm that our model does not include adjacency effects from neighboring pixels and slopes. We added this information:

*"We currently exclude the effects of adjacent pixels and slopes both to limit the complexity of our forward model and because their impact on modeled radiance is less critical than the separation of downward direct and diffuse transmittance (Guanter et al., 2009; Picard et al., 2020)."*

l.123: what tool was used to compute these HDRF?

We used DISORT to compute snow HDRF. We updated the captions of Figures 1 and 2 and revised:

*"Figure 1 shows HDRF simulated with DISORT as a function of $\theta_i$ for snow grain radii of 100 $\mu$m (panel a) and 1000 $\mu$m (panel b)."*

l.138: it's not obvious to me why the backward reflectance decreases with less direct irradiance (it means comparing backward and side scattering). Some explanation detailing the equivalent incidence angle of diffuse illumination would be helpful

The scenario shown in Figure 2 does not use a specific incidence angle for the diffuse illumination. We show HDRF for isotropic, hemispheric diffuse illumination and direct irradiance at a specific solar zenith angle. We refer Dr. Libois to Schaepman-Strub et al. (2006), Figures 8-10, which show that HDRF at 30° solar zenith and 0° view zenith angle decreases at

550 nm with increasing diffuse component of illumination. At the same time, Figure 10 in their paper confirms the opposite behavior for illumination angles greater than ~50°, which is shown in Figure 2b of our manuscript as well. The minimum values of HDRF in the visible wavelengths under fully diffuse illumination conditions and nadir view angle is caused by the angular intersection of the strong forward scattering phase function with the surface. We slightly revised the respective paragraph as follows:

*"Figure 2 highlights the sensitivity of HDRF to different fractions of direct solar irradiance and a complementing isotropic, hemispheric diffuse illumination for solar zenith angles of 0° (panel a) and 80° (panel b). Snow surfaces are properly forward scattering, i.e., have a significantly higher reflectance factor in the forward direction than in the backward direction, though only for direct irradiance fractions of >80% (Schaepman-Strub et al., 2006). As a consequence, we observe an inverse behavior of HDRF magnitude for the two illumination conditions, particularly in the VIS wavelengths. With smaller amounts of direct irradiance, HDRF decreases at $\theta_i = 0°$ due to the angular intersection of the forward scattering phase function with the surface (Schaepman-Strub et al., 2006), whereas its values significantly increase at $\theta_i = 80°$."*

l.141: could you clarify whether "scattering by surrounding objects" can actually be modeled.

Yes, there exist some attempts to model the "scattering by surrounding objects". For instance, Picard et al. (2020) developed physical equations to account for both dark and bright reflectance from neighboring pixels. However, they conclude that this theory is 'complex […] and requires information or assumptions on the neighbouring slope, which limits its interest in practice'." For clarification, we added the following sentence to the description of our forward model:

*"We currently exclude the effects of adjacent pixels and slopes both to limit the complexity of our forward model and because their impact on modeled radiance is less critical than the separation of downward direct and diffuse transmittance (Guanter et al., 2009; Picard et al., 2020)."*

l.144: I regret that EMIT has not been introduced before in more details. In particular its spatial resolution seems to be a critical quantity if it is meant to see independently distinct mountain slopes instead of a mixture of various slopes.

Yes, we agree that this information is critical and was missing before. We added a more proper description of the EMIT instrument to the introduction:

*"Extending over a wavelength range of 380-2500 nm with a spectral resolution of approximately 7.5 nm, EMIT images provide a pixel size of around 60 m on a 74 km wide swath. The temporal revisit time is variable depending on the orbital cycle of the ISS, and ranges between one day and more than a week (Thompson et al., 2024). After more than a year in operation, EMIT provides data products from many different regions of the Earth, including snow covered high mountains, and experiments have confirmed a remarkably high SNR of more than 500 on average and above 750 in the VNIR wavelengths (Thompson et al., 2024)."*

l.163: I think units (here and elsewhere) should not be italic

Revised.

l.172: can you clarify whether you invert independently the individual pixels, or not.

Yes, we invert each individual pixel independently. We added this information:

*"Following Thompson et al. (2018), we combine an atmosphere and surface state in $\boldsymbol{x}$, so that $\boldsymbol{x} = [\boldsymbol{x}_{ATM}, \boldsymbol{x}_{SURF}]^T$, and invert each pixel of the image, i.e., each measured radiance spectrum, independently."*

Eq. (2): I think it could be better explained in terms of the various contributions. Also, transmittance is a physical property (of the atmosphere for instance). Here it seems that it includes the partition between direct and diffuse irradiance. I'd recommend to explicitly mention the direct/diffuse partition. Also, I don't know what "atmospheric path radiance" is. Is it related to the spherical albedo of the atmosphere? By the way spherical albedo has not been defined before.

Thanks for this comment. Yes, transmittance is a physical property of the atmosphere. That's how we use it in our forward model to simulate radiance measured at the sensor. And yes, direct and diffuse parts of the downwelling transmittance are scaled differently, so that those two terms represent the partition into direct and diffuse irradiance at the surface. Atmospheric path radiance describes the number of solar photons that are scattered by the atmosphere into the line-of-sight of the sensor without interaction with the surface. It is not related to the spherical albedo of the atmosphere, which is the multiple scattering of photons between the target pixel and the surrounding atmosphere before they enter the line-of-sight of the sensor. We clarified our explanation of the various terms:

*"$\boldsymbol{l}_0$ is the atmospheric path radiance, i.e., the number of solar photons that are scattered by the atmosphere into the line-of-sight of the sensor without interaction with the surface. $\boldsymbol{t}^{\downarrow}_{dir}$, $\boldsymbol{t}^{\downarrow}_{dif}$, and $\boldsymbol{t}^{\uparrow}$ are direct downwelling, diffuse downwelling, and total upwelling transmittance of the atmosphere. Scaled by two different angles, $\boldsymbol{t}^{\downarrow}_{dir}$ and $\boldsymbol{t}^{\downarrow}_{dif}$ represent the partition into direct and diffuse irradiance at the surface. Finally, $\boldsymbol{s}$ is the spherical albedo of the atmosphere, which describes the multiple scattering of photons between the target pixel and the surrounding atmosphere before they enter the line-of-sight of the sensor."*

l.185: I'd expect the HDRF to depend also on the direct/diffuse partition. Regarding the incidence angle what is the motivation to have it in the state vector instead of using a DEM? How would the results with fixed vs retrieved incidence angle compare?

Yes, that's absolutely right. The HDRF is also a function of the direct/diffuse partition. We added this for clarification:

*"$\boldsymbol{\rho}_s$ is the HDRF and a function of $\boldsymbol{x}_{SURF}$ that holds snow grain size, liquid water content, algae concentration, and dust mass mixing ratio, as well as of the partition into direct and diffuse irradiance."*

The motivation of having the incidence angle in the state vector instead of using a DEM is driven by findings of a recent publication from Carmon et al. (2023): using an external DEM is "error-prone since static global digital elevation models do not generally achieve the accuracy required, and even minor mismatches in spatial resolution can introduce significant artifacts in downstream processing. Here we demonstrate that it is possible to estimate topographic parameters directly from spectral data, ensuring perfect physical consistency,

temporal coincidence, and spatial alignment". We refer Dr. Libois to this manuscript for more details, also regarding the comparison between results with fixed vs. retrieved incidence angle. Our study is focused on including the direct/diffuse irradiance partition and snow reflectance anisotropy in the forward model, rather than investigating the effect of having the incidence angle as a free parameter in the state vector. However, we added a short rationale:

*"In addition, we remove dependency from digital elevation models by adding $\theta_i$ to $\boldsymbol{x_{SURF}}$, which ensures "physical consistency, temporal coincidence, and spatial alignment" (Carmon et al., 2023)."*

l.186-187: Not very clear. Do you mean that previously the HDRF was in the surface state vector? Also I'm afraid to read that you assume spherical particles for snow (confirmed l. 198), which are very inappropriate, in particular when it comes to computing HDRF. Database exist for more realistic snow BRDF data (from either measurements of models). If not detailed elsewhere, could you clarify how many snow layers you use in the model.

Yes, that's correct. The HDRF was previously part of the state vector. In fact, the reflectance values of all instrument channels were being optimized in the former approach. We revised for more clarity:

*"Our approach breaks from previous implementations as we calculate all wavelength-dependent values of $\rho_s$ by running a combination of Mie scattering theory and the multistream DISORT program (Stamnes et al., 1988) instead of having them as free parameters in the surface state vector."*

We understand and agree that assuming spherical particles for snow might be inappropriate. However, we actually follow the approach of Grenfell and Warren (1999) and model non-spherical snow particles by a collection of independent spheres that has the same volume-to-surface-area ratio as the non-spherical particles. Several studies have shown that this approach provides an accurate representation of extinction efficiency and single-scattering albedo, while only the scattering asymmetry factor is usually overestimated (Grenfell and Warren, 1999; Neshyba et al., 2003; Grenfell et al., 2005; Warren, 2019). Its effect on bulk optical properties can be compensated though by reducing the grain size of the model (Dang et al., 2016). Nevertheless, more realistic snow BRDF datasets certainly exist, but we would like to defer the testing of various representations of snow and ice optical properties in our retrieval framework to a subsequent study. We consider the current manuscript rather as a concept study for the simultaneous retrieval of atmosphere and physical surface properties directly from at-sensor radiance. In fact, we changed and adjusted both title and narrative of the manuscript, following the suggestions of reviewer #2. In any case, we extended the discussion in Section 5.2, also to accommodate with one of the main comments of Dr. Libois:

*"We apply Mie scattering theory to obtain the ice OPs, and follow the approach of Grenfell and Warren (1999) to model non-spherical snow particles by a collection of independent spheres that has the same volume-to-surface-area ratio as the non-spherical particles. This assumption might be inappropriate, but studies have shown that this approach provides an accurate representation of extinction efficiency and single-scattering albedo, while only the scattering asymmetry factor is usually overestimated (Grenfell and Warren, 1999; Neshyba et al., 2003; Grenfell et al., 2005; Warren, 2019). Its effect on bulk optical properties can be compensated though by reducing the grain size of the model (Dang et al., 2016). Nevertheless, more accurate representations of the optical shape of snow as well as more realistic snow*

*BRDF datasets certainly exist (e.g., see Malinka (2014), Dumont et al. (2021), Malinka (2023)), but we would like to defer the testing of various representations of snow and ice optical properties in our retrieval framework to a subsequent study. We consider the current manuscript rather as a concept study for the simultaneous retrieval of atmosphere and physical surface properties directly from at-sensor radiance. However, we have to be aware of potential misrepresentations in many cases."*

We're using 3 horizontal layers for modeling snow reflectance. A small near-surface layer that contains impurities, and two semi-infinite LAP-free snow layers. We added this:

*"We run DISORT with 16 streams and use three horizontal layers for modeling snow HDRF: a small near-surface layer that contains impurities, and two semi-infinite LAP-free snow layers."*

l.187: how many streams are used for the DISORT simulations?

We used 16 streams for our DISORT simulations. Added:

*"We run DISORT with 16 streams and use three horizontal layers for modeling snow HDRF: a small near-surface layer that contains impurities, and two semi-infinite LAP-free snow layers."*

l.190: do you mean that the dimension was larger previously due to the multispectral dimension?

In the previous implementation, the reflectance values of each instrument channel were part of the state vector of free parameters. In case of EMIT, $x$ contained 285 surface and 3 atmosphere parameters. The new approach reduces this to 5 and 3, respectively. We added this clarification:

*"While $x_{SURF}$ held the reflectance values of each of the 285 EMIT channels in the previous implementation, our approach reduces this number to only 5 parameters (Table 1)."*

l.197: not only the asymmetry parameter matters, but also the detailed phase function for that kind of applications

Thanks for the comment! That's of course right. We added the respective information:

*"Next, we calculate the single-scattering phase functions by decomposing the Henyey–Greenstein phase function, which better captures the actual phase function of snow than the phase function for spheres, into Legendre coefficients for 20 moments (Aoki et al., 2000; Painter and Dozier, 2004a)."*

l.199: how is then defined the snow grain size? Including the liquid water coating? What about the size distribution of snow particles?

We define the snow grain size as the diameter or radius of the collection of spheres, excluding the liquid water coating (for details, we refer Dr. Libois to Green et al. (2002)). We don't include specific assumptions about the size distribution of snow particles, although we agree that it would be important to be considered. This will certainly be part of our subsequent study,

which will be focused on the modeling of snow and ice particles in our retrieval framework. We revised:

*"We assume the snow grains to be shaped as a collection of spheres characterized by a specified radius that equals three times the volume-to-area ratio of the real non-spherical snowpack (Warren, 2019), and apply traditional Mie Theory to obtain their OPs (Grenfell and Warren, 1999). [...] We add the influence of liquid water by modeling OPs of coated spheres for the case of wet snow by adding the width of a circular layer of liquid water to the grain radius (Green et al., 2002)."*

l.204: would you have any reference to support that algae are similar in Greenland and Patagonia? Otherwise why would you believe this? Also could you clarify what optical property is defined. Only a mass absorption coefficient? The same question holds for dust.

These are very important questions, thanks for raising them! We revised the discussion about the choice of LAP optical properties in Section 5.2:

*"To model biological LAPs, we utilize a set of algae OPs for the species Ancylonema (glacier algae) as well as Sanguina nivaloides and Chloromonas nivalis (snow algae), derived from samples collected on the Greenland Ice Sheet (Chevrollier et al., 2022). Despite being characterized at a different geographic location far away from our study site, we assume that these OPs adequately represent algae cells found on ice sheets, glaciers, and snow worldwide. This is corroborated by previous studies that identified those three species as being responsible for the darkening of snow and ice surfaces in various regions, including the Greenland Ice Sheet, Svalbard, the European Alps, and the Sierra Nevada in California (Yallop et al., 2012; Remias et al., 2012; Di Mauro et al., 2020a; Painter et al., 2001). Moreover, Takeuchi & Kohshima (2004) and Kohshima et al. (2007) identified Ancylonema and Chloromonas algae as among the most frequently encountered species on the Patagonian Ice Sheet.*

*The use of dust OPs poses a different challenge, as they strongly depend on mineralogy and source area (Di Biagio et al., 2019). Several sets of dust OPs from different geographic regions, derived using diverse techniques and data, are publicly available. They have been obtained from any combination of field samples, spectral measurements, and linear mixing modeling, with Sahara, Colorado, Greenland, and Mars being the most prominent regional types (Polashenski et al., 2015; Skiles et al., 2017b; Balkanski et al., 2007, Singh et al., 2016). However, only a few studies have considered specific dust minerals when assessing their impact on snow melt (Lawrence et al., 2010; Kaspari et al., 2014; Reynolds et al., 2014). For our study, we selected only one type of dust OPs representing rather large particles, measured from samples that were collected in the San Juan Mountains of southwestern Colorado (Skiles et al., 2017b). In the lack of dust OP characterization in South America, we believe that the Colorado type is closest to the dust type found in Patagonia. This is especially supported by the finding that very large dust particles are often present in patchy snow of arid environments (Skiles et al., 2017b). Moreover, studies of the geochemical composition and mineralogy suggest that both the San Juan and the Patagonian dust are significantly dominated by quartz with 30-50 % of the total mineral mass (Lawrence et al., 2010; Demasy et al., 2024). Such analyses will be facilitated on even larger geographical scales by the EMIT mission objective, which is providing an improved understanding of the mineralogy of dust particle source regions, and enabling an enhanced identification and classification of dust OPs and their distribution around the Earth's snow-covered areas."*

We use single-scattering albedo, mass extinction coefficient, and asymmetry parameter as optical properties for LAPs. Essentially, they are the same as for the snow particles and we model the LAP-contaminated snow as a linear mixture of all OPs, as we've mentioned in lines 197-198 and 208 of the initial manuscript submission.

l.213: what do you mean by "atmospheric aerosols"? Those assumed in MODTRAN?

We mean atmospheric aerosols in general, including those assumed in MODTRAN. To our knowledge, almost all atmosphere radiative transfer models treat aerosols as absorptive and/or scattering particles that cause a smooth decrease in reflected photons in the visible part of the solar spectrum.

l.222: it should be clear what measurements are included here. Multi spectral or also multi-pixels?

Good point! Equation 3 is applied on a per-pixel basis, so $y$ is always one single radiance spectrum measured at the sensor. We clarified this:

*"Following Thompson et al. (2018), we combine an atmosphere and surface state in $x$, so that $x = [x_{ATM}, x_{SURF}]^T$, and invert each pixel of the image, i.e., each measured radiance spectrum, independently."*

l.226: then why using a prior at all if in the end it does not constrain the cost function?

The priors are actually not completely flat since we apply light constraints to the parameters in $x_{ATM}$, i.e., water vapor, aod, and pressure elevation. We updated with this important information:

*"However, we use only light constraints on the parameters in $x_{ATM}$, and uninformative, flat priors for $x_{SURF}$ by adding large values to the diagonal of $S_a$."*

l.227: for the model to be well-posed it should be proved that measurements at distinct wavelengths are actually independent, and the number of spectral channels should be mentioned (to be compared to the number of parameters to be retrieved).

Yes, good point! We've already mentioned in the initial manuscript that instrument noise is commonly assumed to be uncorrelated between channels (Thompson et al., 2018) (lines 222-223), which underlines the independence of measurements at distinct wavelengths. We revised to add the number of EMIT's spectral channels:

*"Due to the reduced number of only eight state vector parameters and 285 independent measurements from EMIT's spectral channels, the problem is well-posed, in contrast to using previous surface models (Thompson et al., 2018; Bohn et al., 2021; Bohn et al., 2022)."*

Eq. (4): here again this factor is very dependent on the actual phase function of snow, which is likely to be different than that of spheres.

Fully agreed! For this reason, we don't use the phase function for spheres, but the Henyey-Greenstein (HG) phase function. Aoki et al. (2000) have shown that modelled snow HDRF agrees better with measurements in this case. The HG phase function is very smooth, while

that of spheres features ice bow and glory peaks not seen for real snow along with very low sideward scattering (Räisänen et al., 2015). As mentioned before, we added the following:

*"Next, we calculate the single-scattering phase functions by decomposing the Henyey–Greenstein phase function, which better captures the actual phase function of snow than the phase function for spheres, into Legendre coefficients for 20 moments (Aoki et al., 2000; Painter and Dozier, 2004a)."*

l.230: the main advantage of optimal estimation is to provide an estimation of the posterior error, which is not discussed at all. It would be worth adding this uncertainty range for the retrieved parameters of interest.

We absolutely agree that providing posterior uncertainties is one of the main advantages of optimal estimation. Our initial submission was more focused on the dependency of the retrieved parameters on topographic and anisotropic effects of the surface than on error estimates given by our OE inversion setup. However, we added the posterior uncertainties, i.e., the square-root of the variance, to each retrieved parameter value in Table 2, and briefly discuss the numbers in a new paragraph:

*"Table 2 also shows posterior uncertainties for each retrieved parameter in standard deviation as reported by the OE framework. Errors are generally low, in particular for quantities exhibiting distinct absorption features in the reflectance spectrum, such as water vapor, grain size, or algae. Higher uncertainties are reported especially for AOD, but also for retrieved dust concentration. It has to be noted though that the linearized posterior error predictions by OE are likely optimistic. OE presumes a linearized version of the forward model, with a local multivariate Gaussian error prediction. This ignores other local minima solutions, if they exist, and even the local estimate may under predict errors (Hobbs et al., 2017; Cressie, 2018). However, the uncertainty estimates provide at least a useful hint whether the variance of retrieved values is reasonable or not."*

l.248: I think the units should be like W m-2.

Fixed.

l.264: how do EMIT spatial resolution and SRTM match (or not)? Is SRTM averaged somehow to find incidence angles comparable to EMIT retrievals?

The spatial resolution of SRTM data is 30 m, so finer than EMIT measurements, which have 60 m pixel size. We averaged four SRTM pixels to get an EMIT-equivalent surface elevation and respective slope and aspect angles. We added this information to the manuscript:

*"The SRTM DEM has a spatial resolution of 30 m, so that we averaged four pixels to get surface elevation, slope, and aspect values that match EMIT's ground sampling distance of 60 m."*

l.268: the correlation between snow grain size and slope is tricky. You could either argue that the retrieval is homogeneous for snow grain size in a mountainous terrain with various slopes, suggesting that accounting for slope corrects for an apparent heterogeneity of snow grain size when assuming flat terrain. Or you give a physical reason why snow grain size can differ

depending on the slope… Looking at the correlations between retrieved parameters may help clarify this point. A too strong correlation may indicate compensation between both variables.

Thanks for this comment! We agree that this correlation is tricky. We discuss it in a few paragraphs of the initial submission of the manuscript. We mention that a higher amount of direct illumination on sun-facing slopes could potentially induce melting processes leading to larger grain sizes due to clustering, but doesn't necessarily need to (lines 266-268). Figure 8a gives an indication for a potential correlation between slope and grain size by plotting the difference in assumed incident angles to the difference in retrieved grain size. It's not a strong correlation, but it's somewhat present. In addition, we point to Figures 1, 2, and 5, which confirm that the anisotropy of snow reflectance causes changing shape and magnitude of HDRF as a function of increasing grain size all along the solar spectrum. Hence, varying ratios of direct to diffuse illumination as a consequence of erroneously assumed local solar zenith angles could lead to errors in derived snow grain size (lines 333-336).

l.271: I would not necessarily say that snow grains of 200 microns (at least if it is the radius) are small.

We agree and removed the respective words.

Fig.7: it should be clear somewhere that EMIT L2A is the standard EMIT product ignoring topography

Absolutely right! We added the following phrases:

*"We also show corresponding spectra from the EMIT L2A product. They were retrieved by applying the same OE technique, but without considering topography in the forward model, and by obtaining HDRF from statistical modeling using constrained priors instead of utilizing a snow surface radiative transfer model."*

l.299: it is not clear what the single-transmittance model is (what wavelength?).

Thanks for raising this point. We understand that we haven't defined this term. We updated the caption of Figure 7:

*"For comparison, panels (a) and (c) are complemented by results from the EMIT L2A product, which assumes $\theta_i = \theta_0$, i.e., uses a single downward transmittance term (direct + diffuse) in the forward model (called single-transmittance model hereinafter)."*

l.303: I think this "hook" behavior should be better identified in the figure. Is it the too strong decrease in the blue visible in the HDRF? I believe the direct/diffuse partition, that greatly changes in this spectral range, if not properly accounted for can also contribute to this hook.

Yes, the "hook" is the strong decrease in HDRF in the visible blue wavelengths. We added some more explanation:

*"The assumed direct illumination and direct to diffuse ratio are consequently too small, leading to higher reflectance due to more photons reaching the instrument, and causing a red-shift in HDRF and the formation of a downward hook, which is identified as a strong decrease in HDRF in the VIS blue wavelengths below 500 nm."*

Absolutely, the direct/diffuse partition significantly contributes to the formation of the hook, if not properly accounted for. This is exactly what we mean by saying 'The assumed direct illumination and direct to diffuse ratio are consequently too small, …' in lines 301-302 of the initial manuscript submission. We can only change this ratio by assuming different incident angles for direct and diffuse illumination.

l.311: I don't see in this paragraph the sensitivity to assuming a Lambertian snow surface. Unless both impacts are combined altogether. In this case it would be worth separating both to disentangle the impacts, and point what assumptions is most critical.

Good point! The term 'assumption of a Lambertian surface' doesn't really fit here. We removed it.

l.325: only here is the direct/diffuse partition explicitly mentioned, while I think it would be valuable to clarify its treatment and impact earlier on.

Agreed and revised by adding some clarification to the description of our forward model:

*"$t^{\downarrow}_{dir}$, $t^{\downarrow}_{dif}$, and $t^{\uparrow}$ are direct downwelling, diffuse downwelling, and total upwelling transmittance of the atmosphere. Scaled by two different angles, $t^{\downarrow}_{dir}$ and $t^{\downarrow}_{dif}$ represent the partition into direct and diffuse irradiance at the surface."*

l.333: the 3 digits may be a lot for such an estimation.

Agreed and reduced to one digit.

Table 2: any comment on the fact that algae can be zero somewhere, and present elsewhere?

To estimate algae concentration, our retrieval is sensitive to their distinct absorption features caused by, e.g., carotenoid and chlorophyll. If those features are not present or only very weakly expressed in the HDRF, the framework reports very close to zero numbers. This happened in cases S1-S3 and I1, so that we decided to put a zero number in Table 2. However, we addressed this issue by adding the posterior uncertainties to each retrieved state vector parameter. The variances in the posterior covariance matrix are never exactly equal to zero, so that the estimated posterior mean isn't actually zero either if we enclose it by its variance.

l.375: "small" is awkward. Preliminary?

Right, we recognize that this is indeed a strange formulation. Revised accordingly.

l.376: this point suggests that their could be correlations between the retrieved parameters. You could look at these correlations to inform about the independence (or not) of the retrieved parameters, which is trivial with optimal estimation. The underlying question being for instance:can the retrieval algorithm return stronger AOD and lower LAP in snow, which may result in more or less the same apparent radiance at TOA?

Thanks for this comment! That's a very good suggestion. We calculated the error covariance matrix for the retrieved parameters from the image of the Chilean ice field following Govaerts et al. (2010), and added this result to Section 4.2.2 as Figure 9, including a new paragraph:

*"Another way of assessing the sensitivity of retrieved parameters is to look at their error correlation. OE provides a measure of retrieval uncertainty for each state vector element in terms of a posterior covariance matrix, which can be normalized to an error correlation matrix (Rodgers, 2000; Govaerts et al., 2010). We calculated this matrix from the retrieval maps for the Chilean ice field for both atmosphere and surface properties (Fig. 9). The coefficients confirm the findings from Fig. 8 by showing a slight negative correlation between $\theta_i$ and grain size, a slight positive correlation between $\theta_i$ and dust, and no correlations between the assumed incident angle and both liquid water and algae. We observe expected anticorrelation between pressure elevation and both AOD and water vapor, and identify an overall disentanglement between surface and atmosphere parameters. Only exceptions are $\theta_i$, being an input to both atmosphere and surface model, and AOD, which features a negative correlation with dust LAPs. $\theta_i$, or more precisely, the assumed ratio of direct to diffuse irradiance, dust, and atmospheric aerosols cause a very similar shape of reflectance in the visible wavelengths, leading to potential ambiguities between them."*

In addition, we added some of these findings to Section 4.2.4 about the blue wavelengths:

*"Our selection of AOD for the comparison is supported by the findings from Fig. 9, which show a negative correlation between AOD and both $\theta_i$ and dust. In other words, the retrieval could bare the risk of compensating for a biased AOD by altering the other two parameters and vice versa."*

And:

*"In contrast, variations in AOD can not compensate for erroneously assumed $\theta_i$, which supports the conclusion that the anticorrelation between the two properties is rather one-sided, and a correct assumption or retrieval of the incident angle is more important than a proper characterization of atmospheric aerosols."*

The calculated error correlation coefficients indeed show a negative correlation between AOD and dust as well as AOD and the local incident angle, so that there's a certain risk that the retrieval returns stronger AOD and lower dust, resulting in the same modeled apparent radiance.

l.382: this suggests that AOD cannot be accurately retrieved, unless it is the blue end of the spectrum that puts most constraint on AOD (rather than the longer wavelengths). Could you expand on that?

Yes, sure. The estimation of AOD has traditionally been a particular challenge since most algorithms use strong assumptions about scene content, or shadowed pixels that may be absent at coarse ground sampling. Thompson et al. (2018) suggest that jointly estimating atmospheric state and the surface reflectance spectrum can make better use of the information in the VSWIR interval, i.e., facilitate a more accurate retrieval of AOD. However, it's still heavily dependent on the aerosol optical properties as defined within the utilized atmosphere radiative transfer model. We might circumvent this at least to a certain degree by being able to accurately retrieve the local incident angle directly from the measured radiance, as shown by Carmon et al. (2023) and confirmed by our manuscript. This adds one constraint to the shape of the reflectance in the blue wavelengths, potentially facilitating a more reliable retrieval of AOD. In any case, we need to investigate this in more detail in subsequent studies. We added this discussion to Section 4.2.4:

*"In contrast, variations in AOD can not compensate for erroneously assumed $\theta_i$, which supports the conclusion that the anticorrelation between the two properties is rather one-sided, and a correct assumption or retrieval of the incident angle is more important than a proper characterization of atmospheric aerosols. The latter has traditionally been a particular challenge since most algorithms use strong assumptions about specific scene content that may be absent at coarse ground sampling distance. Thompson et al. (2018) suggest that jointly estimating atmospheric state and the surface reflectance spectrum can make better use of the information in the VSWIR interval, i.e., facilitate a more accurate retrieval of AOD. However, it is still heavily dependent on the aerosol optical properties as defined within the utilized atmosphere radiative transfer model. Being able to accurately retrieve the local incident angle directly from the measured radiance, as shown by Carmon et al. (2023) and confirmed by our manuscript, adds one constraint to the shape of the reflectance in the blue wavelengths, potentially facilitating a more reliable retrieval of AOD."*

l.395: AOT or AOD?

AOD. That was a typo. Fixed.

l.399: what would be the impact of not considering blue wavelenghts in the retrievals? What variables would be most affected, and to which extent?

Not considering blue wavelengths in the retrieval would impede the estimation of aerosol properties, but especially of inorganic LAP in snow, such as dust or black carbon. The slope of HDRF in the shortest wavelengths provides an important hint about the concentration of dust. The retrieval of AOD is difficult anyhow, but we show in our manuscript that we can leverage the blue wavelengths to accurately estimate the local incident angle, which facilitates a more reliable retrieval of AOD. Without the blue part of the spectrum, the difference in the ratio of direct to diffuse irradiance would be less prominent, impeding a correct estimation of $\theta_i$. We added this discussion to Section 5.1:

*"Alternatively, one could exclude blue wavelengths from the retrieval, but it would impede the estimation of aerosol properties and inorganic LAP in snow. The slope of HDRF in the shortest wavelengths provides an important hint about the concentration of dust. The retrieval of AOD is difficult anyhow, but we show in our manuscript that we can leverage the blue wavelengths to accurately estimate the local incident angle, which facilitates a more reliable retrieval of AOD. Without the blue part of the spectrum, the difference in the ratio of direct to diffuse irradiance would be less prominent, impeding a correct estimation of $\theta_i$."*

l.401: this physical explanation for the blue hook could have been given earlier on, and a bit more detailed.

Thanks for this comment! We recognized that our statement is actually incorrect. The blue-shift in reflectance is actually induced by a too high assumed direct portion of incoming light (red-shift in irradiance) and is expressed by the formation of an upward hook in the shortest wavelengths. In contrast, the downward hook in the blue wavelengths is caused by a too high assumed diffuse portion of incoming light (blue-shift in irradiance), leading to a red-shift in reflectance. More details are already given in Section 4.2.1 of the initial manuscript submission (lines 292-295 and lines 301-305). We now provide a more comprehensive physical explanation earlier in the manuscript:

*"Likewise, the assumed ratio of direct to diffuse irradiance is too large, which compensates for present LAP absorption in the VIS wavelengths. Overall, this leads to a blue-shift in HDRF and is expressed by an upward hook in the shortest wavelengths."*

And:

*"The assumed direct illumination and direct to diffuse ratio are consequently too small, leading to higher reflectance due to more photons reaching the instrument, and causing a red-shift in HDRF and the formation of a downward hook, which is identified as a strong decrease in HDRF in the VIS blue wavelengths below 500 nm."*

l.402: can you expand on these laboratory measurements?

We recognized that we actually don't have any sound prove of this statement and thus, removed it.

l.411: where does this assumption come from?

The assumption about the value of peak radiance originates from various tests we ran during the preparation of the manuscript. However, as we didn't include a more detailed derivation, we removed it and added a reference that illustrates the difficulties of calibrating the blue wavelength range in imaging spectroscopy:

*"Saturation and non-linearity effects in sensor radiometry, which occur over very bright surfaces such as snow (Helmlinger et al., 2016)."*

l.412: you could also refer to Picard et al. (2016) who suggest absorption is in between Warren and Brandt (2008) and Warren (1984).

Added accordingly:

*"For instance, the imaginary part of the refractive index as presented by Warren (1984) indicates more ice absorption in the shortest blue wavelengths than the updated index in Warren and Brandt (2008), while Picard et al. (2016) suggest that the magnitude of absorption lies in-between the two."*

l.423: on which basis do you argue that the spherical assumption is the best general shape? As a suggestion, Malinka (2014, 2023) has developed a general mixture model that works very well to estimate the "optical shape" of snow. Maybe it's relevant as well for snow BRDF. See also Dumont et al. (2021).

Thanks for this comment! We recognize that arguing that the spherical assumption provides the best general shape is tricky and doesn't have a sound basis. We removed this statement and significantly revised the paragraph presenting the discussion of snow optical properties:

*"We apply Mie scattering theory to obtain the ice OPs, and follow the approach of Grenfell and Warren (1999) to model non-spherical snow particles by a collection of independent spheres that has the same volume-to-surface-area ratio as the non-spherical particles. This assumption might be inappropriate, but studies have shown that this approach provides an accurate representation of extinction efficiency and single-scattering albedo, while only the*

*scattering asymmetry factor is usually overestimated (Grenfell and Warren, 1999; Neshyba et al., 2003; Grenfell et al., 2005; Warren, 2019). Its effect on bulk optical properties can be compensated though by reducing the grain size of the model (Dang et al., 2016). Nevertheless, more accurate representations of the optical shape of snow as well as more realistic snow BRDF datasets certainly exist (e.g., see Malinka (2014), Dumont et al. (2021), Malinka (2023)), but we would like to defer the testing of various representations of snow and ice optical properties in our retrieval framework to a subsequent study. We consider the current manuscript rather as a concept study for the simultaneous retrieval of atmosphere and physical surface properties directly from at-sensor radiance. However, we have to be aware of potential misrepresentations in many cases.”*

l.426: much larger than what? Why couldn't it be large spherical particles?

Fair point. We decided to remove this phrase and provide a more plausible rationale:

*"The grains on bare ice surfaces typically appear to be arbitrarily shaped with irregular dimensions, so that, e.g., a Geometric Optics approach based on ray-tracing would be more appropriate to model their OPs (Kokhanovsky and Zege, 2004; Cook et al., 2020; Bohn et al., 2022).”*

l.427: It's definitely a good idea, and I would strongly suggest to further investigate this in the present paper.

We absolutely agree! However, we believe that this would go beyond the scope of our manuscript, which we rather consider to be a concept study for the simultaneous retrieval of atmosphere and physical surface properties directly from at-sensor radiance. As mentioned before, we changed and adjusted both title and narrative of the manuscript, following the suggestions of reviewer #2. At least, we now clearly point out that future work needs to and will follow:

*"In future work, we need to compare retrieval results from assuming different grain shapes, including spheres, spheroids, hexagonal plates, and Koch snowflakes (He et al., 2017; Hao et al., 2023). We will also apply different approaches to model ice layers, e.g., with enclosed air bubbles and a Fresnel layer between the ice and a thin snow cover (Whicker et al., 2022).”*

l.430: I guess one of the EMIT objectives is to map this variability in dust optical properties, so it might be worth referring to this and directly related studies.

That's correct. We expanded a little bit on that:

*"Such analyses will be facilitated on even larger geographical scales by the EMIT mission objective, which is providing an improved understanding of the mineralogy of dust particle source regions, and enabling an enhanced identification and classification of dust OPs and their distribution around the Earth's snow-covered areas (Connelly et al., 2021, Goncalves Ageitos et al., 2023).”*

l.442: as the spatial resolution has never been discussed it's hard to guess how critical are these mixed pixels.

Fair point. Information about EMIT's spatial resolution is now added:

*"Extending over a wavelength range of 380-2500 nm with a spectral resolution of approximately 7.5 nm, EMIT images provide a pixel size of around 60 m on a 74 km wide swath. The temporal revisit time is variable depending on the orbital cycle of the ISS, and ranges between one day and more than a week (Thompson et al., 2024)."*

l.453: how would you calculate snow fractional cover? By including it in the state vector?

Yes, that would be the optimal way of doing it. However, while extremely important, this is a non-trivial problem and would require some extensive future work. We slightly revised our phrase:

*"A more sophisticated alternative would be to calculate snow fractional cover by including it in the state vector, and use a respective minimum value as constraint."*

l.471: the link between this work and melt runoff and climate model input are not clear, but this might be clarified if it sounds important to the authors.

We agree and removed the mention of this link.

References:

[revised manuscript text omitted]

---

## Author Response (AR2)

Niklas Bohn
Jet Propulsion Laboratory
California Institute of Technology
4800 Oak Grove Dr
Pasadena, CA 91109
urs.n.bohn@jpl.nasa.gov

Dear Nora,

This letter accompanies our manuscript "Do we still need reflectance? From radiance to snow properties in mountainous terrain: a case study with EMIT" (egusphere-2024-1020), which we resubmit for consideration by *The Cryosphere*. We appreciate your time and effort in dealing with our manuscript and for encouraging us to resubmit it. We hope that we have addressed all remaining minor concerns raised by you and the reviewers in this revised version of the paper and in our point-by-point answers, which follow below. In the revised version of the manuscript, the changes are highlighted in magenta.

We hope that this new version of the manuscript meets the quality criteria necessary for publication in The Cryosphere and look forward to future correspondence. Many thanks for your help again.

Sincerely,

Niklas Bohn
*Jet Propulsion Laboratory, California Institute of Technology*
*Pasadena, CA, USA*

Edward H. Bair
*Civil Group, Leidos, Inc.*
*Reston, VA, USA*

Philip G. Brodrick, Nimrod Carmon, Robert O. Green, David R. Thompson
*Jet Propulsion Laboratory, California Institute of Technology*
*Pasadena, CA, USA*

Thomas H. Painter
*Joint Institute for Regional Earth System Science and Engineering, UCLA*
*Los Angeles, CA, USA*
* * *
Ms. Ref. No: egusphere-2024-1020
Do we still need reflectance? From radiance to snow properties in mountainous terrain: a case study with EMIT

**Anonymous Referee #1**

I thank the authors for addressing my comments and I apologize for my late re-review. I'm satisfied with their answers. I have just two minor comments on the revised version, detailed below.

We thank the referee for re-reviewing our manuscript and for the positive feedback. Below are our responses to the two minor comments.

1- I see that the title has been now changed according to a specific request from R#2. I'm ok with that, but I suggest that the authors should try to answer to the rhetoric question in the title. In fact, at the moment I can't find a specific section in the discussion part of the manuscript. I suggest also to include one sentence in the abstract.

That is a very good point, and we agree that we have not really answered the question in none of the different sections. We updated the second half of the abstract (lines 14-22):

*"Moreover, the term 'surface reflectance' is often used with ambiguity in the literature, which instantly raises the question if we still need this quantity as a retrieval product. In this contribution, we present a novel forward model that couples the MODTRAN atmosphere radiative transfer code with a physics-based snow reflectance model that utilizes the multistream DISORT program. Our model allows to estimate snow surface and atmosphere properties directly from measured radiance. We apply the approach to EMIT images from Patagonia, South America, and compare our results to the EMIT L2A products that retrieve surface reflectance as a free parameter. We find discrepancies in snow grain size of up to 200 μm and in dust mass mixing ratio of up to 75 μg g$^{-1}$. Furthermore, we demonstrate differences in instantaneous LAP radiative forcing of up to 400 W m$^{-2}$. We conclude that we still need reflectance, but only if clearly defined and preferably as a modeled quantity within the forward model."*

In addition, we added subsection 5.5 "Do we still need reflectance" to the discussion (lines 569 - 577):

*"Remote sensing retrievals that utilize measurements of reflected sunlight will always need a reflectance term within the forward model as this is the essential quantity to summarize the radiative transfer of photons through a material. However, our findings confirm that this term needs to be treated carefully and should either refer to the intrinsic reflectivity of the target material or even more accurately, be defined as a bi-directional reflectance distribution function (BRDF) providing coupled, angle- and illumination-dependent reflectance terms (Vermote et al., 1997; Schaepman-Strub et al., 2006; Verhoef and Bach, 2007; Guanter et al., 2009). Furthermore, our study shows that using surface reflectance as a modeled quantity within the forward model rather than retrieving it as a free parameter improves the accuracy of estimated biophysical snow properties. In the absence of well-parametrized surface radiative transfer models, this can be balanced by a clear definition and consistent use of the surface reflectance term."*

Finally, we added the following sentence to the conclusion (lines 591-592):

*"Surface reflectance is still needed as a modeled quantity within the forward model but must be consistent and clearly defined."*

By the way, I saw also a new paper just published on TC on this same topic (Wilder et al. 2024). Probably it's worthwhile add this reference in the bibliography of this manuscript.

Absolutely! We added this reference in lines 98-100:

*"Only recently, Wilder et al. (2024) introduced a more mature and comprehensive modeling of topography for spaceborne imaging spectroscopy data over mountain snow, but most studies apply a limited post-hoc correction at the airborne scale (Painter et al., 2013a; Seidel et al., 2016)."*

And in lines 109-110:

*"This facilitates the consideration of local surface anisotropy and topography in the forward model and removes dependency from external DEMs (Carmon et al., 2023; Wilder et al., 2024)."*

And in lines 201-203:

*"Aligning with Wilder et al. (2024), we remove dependency from digital elevation models by adding $\theta_i$ to $x_{SURF}$, which ensures "physical consistency, temporal coincidence, and spatial alignment" (Carmon et al., 2023)."*

And in lines 477-479:

*"Being able to accurately retrieve the local incident angle directly from the measured radiance though, as shown by Carmon et al. (2023) and Wilder et al. (2024), and confirmed by our manuscript, adds one constraint to the shape of the reflectance in the blue wavelengths, facilitating a more reliable retrieval of AOD."*

2- I see that the blue hook section has been removed, since a more detailed manuscript (also submitted to TC) focuses on this issue. I'm ok with that. I suggest to improve cross-citation between the manuscripts, in order to make clear the distinction of those two studies.

Thanks for this comment. We further improved the cross-citation between the two manuscripts by adding lines 96-98:

*"Overall, there can be a handful of factors biasing snow reflectance shape and magnitude, especially in the blue wavelengths, as demonstrated in a complementing study by Bair et al. (2024)."*

References:

Wilder, B. A., Meyer, J., Enterkine, J., and Glenn, N. F.: Improved snow property retrievals by solving for topography in the inversion of at-sensor radiance measurements, The Cryosphere, 18, 5015–5029, https://doi.org/10.5194/tc-18-5015-2024, 2024.

**Anonymous Referee #2**

In this paper, a combined optimal estimation technique for simultaneous retrieval of retrieval of snow and atmopsheric parameters from EMIT data is presented. The method is new and specifically the inclusion of diffuse-to direct irradiance estimation makes it worth publishing. Also, the authors did a good job in augmenting the paper in comparison to the first version. But still, I think there should be some mandatory changes done in the text to avoid ambiguities.

We thank the referee for the positive feedback. Below are our responses to their comments.

Some specific comments:

- The title asks if one still needs reflectance, but no answer to this question is given int the text - and one should be careful answering this as the confusion between AOD and snow properties shows the risks if omitting reflectance as an intermediate product.

That is a very good point, and we agree that we have not really answered the question in none of the different sections. We updated the second half of the abstract (lines 14-22):

*"Moreover, the term 'surface reflectance' is often used with ambiguity in the literature, which instantly raises the question if we still need this quantity as a retrieval product. In this contribution, we present a novel forward model that couples the MODTRAN atmosphere radiative transfer code with a physics-based snow reflectance model that utilizes the multistream DISORT program. Our model allows to estimate snow surface and atmosphere properties directly from measured radiance. We apply the approach to EMIT images from Patagonia, South America, and compare our results to the EMIT L2A products that retrieve surface reflectance as a free parameter. We find discrepancies in snow grain size of up to 200 $\mu m$ and in dust mass mixing ratio of up to 75 $\mu g\ g^{-1}$. Furthermore, we demonstrate differences in instantaneous LAP radiative forcing of up to 400 $W\ m^{-2}$. We conclude that we still need reflectance, but only if clearly defined and preferably as a modeled quantity within the forward model."*

In addition, we added subsection 5.5 "Do we still need reflectance" to the discussion (lines 569 - 577):

*"Remote sensing retrievals that utilize measurements of reflected sunlight will always need a reflectance term within the forward model as this is the essential quantity to summarize the radiative transfer of photons through a material. However, our findings confirm that this term needs to be treated carefully and should either refer to the intrinsic reflectivity of the target material or even more accurately, be defined as a bi-directional reflectance distribution function (BRDF) providing coupled, angle- and illumination-dependent reflectance terms (Vermote et al., 1997; Schaepman-Strub et al., 2006; Verhoef and Bach, 2007; Guanter et al., 2009). Furthermore, our study shows that using surface reflectance as a modeled quantity within the forward model rather than retrieving it as a free parameter improves the accuracy of estimated biophysical snow properties. In the absence of well-parametrized surface radiative transfer models, this can be balanced by a clear definition and consistent use of the surface reflectance term."*

Finally, we added the following sentence to the conclusion (lines 591-592):

*Surface reflectance is still needed as a modeled quantity within the forward model but must be consistent and clearly defined.*

- You have apparently decided to use the definition of HDRF as of G Schaepman-Strub. This should be clearly stated, as HDRF may be defined that way, but the purely physical definition of Nicodemus is different: HDRF is the BRDF integrated over the incidence hemisphere (ie. white sky). Please change (at least) the wording: instead of 'the HDRF is defined as' use 'in the this paper, the defnition of 'HDRF' as proposed by Schaepman-Strub is used. Even better would be to call this 'bottom-of atmosphere reflectance'.

Thanks for this comment. We agree and implemented the referee's suggestion (lines 121-123):

*"In this study, we follow the definition of HDRF as given by Schaepman-Strub et al. (2006), which is based on the nomenclature of Nicodemus et al. (1977) but incorporates the adaptations to the remote sensing case from Martonchik et al. (2000)."*

- The sentence in the abstract 'previous methods..' is wrong and should be deleted. There are many methods around which use physical models for well defined BOA reflectance retrieval (what you call HDRF) under consideration and also BRDF models are somtimes linked directly. (and the definition of reflectance used in this paper is also physcially shaky).

We thank the referee for this comment. We do not fully agree with this statement. In our opinion, the sentence is not 'wrong' but probably too inclusive. We absolutely acknowledge that methods exist that incorporate physical models for BOA reflectance/HDRF retrievals. However, it is also correct that methods exist that either neglect physical effects of the surface or utilize the BOA reflectance/HDRF as an intermediate non-physical quantity, without proper error propagation from the atmospheric modeling and obtained from statistical modeling. Furthermore, we specifically refer to the case of accurate retrievals of snow surface properties. We therefore softened our expression, but did not remove it (lines 12-14):

*"However, some methods still either neglect physical effects of the surface or utilize the surface reflectance as an intermediate non-physical quantity, in part without proper error propagation from the atmospheric modeling and obtained from statistical modeling."*

- page8: the omission of adjacency may have a strong impact on the results and the given reference does not corroborate that this is not a problem. By nature, snow covered areas are in mountainous terrain and neighbouring slopes as well as the aerosol scattering both lead to strong cross talk to the neighbourhood. This also hampers the validity of the 'HDRF' you have been using as it assumes a flat terrain. A discussion about adjacency should be added.

Thanks for this comment. We agree that Guanter et al. (2009) do not corroborate this finding, and thus, removed this reference. Also, the term 'less critical' that we used in this context is not appropriate. However, Picard et al. (2020) conclude that '…the upward- and downward-looking sensors are affected by additional illumination coming from the slope itself and the neighbouring slopes. [This effect] becomes significant for slopes larger than about 15°. The theory for large slopes is analytically tractable in several particular cases but is more complex than for the small slopes and requires information or assumptions on the neighbouring slope, which limits its interest in practice.' Also, snow-covered areas are not 'by nature' in mountainous terrain. Yes, most of the Earth's mountains are snow-covered, but high-latitude landscapes, including vast plains and ice sheets, are also snow-covered, particularly in the

winter season. Finally, the HDRF we are using does not assume a flat terrain. In fact, we understand it as the ratio of reflected spectral radiance at a local solar and view geometry to the radiant flux that would be reflected from an ideal Lambertian surface, illuminated and observed under the same conditions. We slightly modified our justification of not including adjacency effects to accommodate the referee's suggestion (lines 187-190):

*"While acknowledging their impact on reflectance of larger slopes, we currently exclude adjacency effects of neighboring pixels as their modeling is complex and requires assumptions on the neighboring terrain which may introduce additional uncertainty themselves (Picard et al., 2020)."*

- Snowy areas are often in northern latitudes with small solar zenith angles; excluding extrem solar angles and low apparent reflectances may exclude too much (maybe add a remark about that).

Yes, absolutely right, snowy areas are often to be found in high latitudes. However, we believe the referee means large solar zenith angles, i.e., small solar elevation angles. If yes, we totally agree. However, we do not fully understand what this comment refers to. Why and how did we exclude extreme solar angles and low apparent reflectance? The EMIT sensor does not cover latitudes beyond ~55° due to its orbit on the ISS. Thus, we are not able to utilize EMIT images from those regions the referee is alluding to. Moreover, we actually included examples of large solar zenith angles and low apparent reflectance by applying our retrieval approach to pixels facing away from the sun (see Figure 7a and Figure 10, panels S4 and I2).

- The uncertainties in Table 2 are error propagation, but no errors towards real measurements yet; the AOD uncertainty is somwhat strange as it allows values below zero (?).

Yes, correct, these uncertainties are propagated through the Optimal Estimation based retrieval framework and are taken from the diagonal of the posterior covariance matrix. And yes again, we do not have error estimates yet obtained from comparison to in-situ measurements. We are aware of this and already included this topic as Section 5.4 in the discussion. In fact, all posterior uncertainties theoretically allow values below zero by nature as they represent the standard deviation around the mean value of the posterior distribution of each state vector element. We agree that, in practice, negative values are of course unrealistic. We added a sentence for clarification to lines 407-409:

*"This can not only lead to error distributions extending into the negative value range, but also ignores other local minima solutions, if they exist, and even the local estimate may under predict errors (Hobbs et al., 2017; Cressie, 2018)."*

- processing time seems to be critical - please add the current processing times.

Good point! However, it is difficult to give exact numbers as these highly vary depending on the computing system. At least, we added a comparison to the traditional processing as presented in Thompson et al. (2018) to lines 269-271:

*"It usually finds a solution after five iterations on average with an overall processing speed-up of about four orders of magnitude compared to the traditional approach as presented in Thompson et al. (2018)."*